# One-shot entorhinal maps enable flexible navigation in novel environments

John H. Wen[1,4], Ben Sorscher[2,4], Emily A. Aery Jones[1], Surya Ganguli[1,2] & Lisa M. Giocomo[1,3 ✉]

Animals must navigate changing environments to find food, shelter or mates. In mammals, grid cells in the medial entorhinal cortex construct a neural spatial map of the external environment[1–5]. However, how grid cell firing patterns rapidly adapt to novel or changing environmental features on a timescale relevant to behaviour remains unknown. Here, by recording over 15,000 grid cells in mice navigating virtual environments, we tracked the real-time state of the grid cell network. This allowed us to observe and predict how altering environmental features influenced grid cell firing patterns on a nearly instantaneous timescale. We found evidence that visual landmarks provide inputs to fixed points in the grid cell network. This resulted in stable grid cell firing patterns in novel and altered environments after a single exposure. Fixed visual landmark inputs also influenced the grid cell network such that altering landmarks induced distortions in grid cell firing patterns. Such distortions could be predicted by a computational model with a fixed landmark to grid cell network architecture. Finally, a medial entorhinal cortex-dependent task revealed that although grid cell firing patterns are distorted by landmark changes, behaviour can adapt via a downstream region implementing behavioural timescale synaptic plasticity[6]. Overall, our findings reveal how the navigational system of the brain constructs spatial maps that balance rapidity and accuracy. Fixed connections between landmarks and grid cells enable the brain to quickly generate stable spatial maps, essential for navigation in novel or changing environments. Conversely, plasticity in regions downstream from grid cells allows the spatial maps of the brain to more accurately mirror the external spatial environment. More generally, these findings raise the possibility of a broader neural principle: by allocating fixed and plastic connectivity across different networks, the brain can solve problems requiring both rapidity and representational accuracy.

To navigate, the brain builds a map of space that integrates landmark information with idiothetic cues derived from self-motion[1,7,8]. A challenge faced by navigating animals is the need to incorporate changing environmental features into their existing neural maps of space. Such changes can be caused by weather patterns, natural forces or resource competition, over long (for example, seasonal changes) and short (for example, destruction of a nest site) timescales. How the brain rapidly incorporates such changing environmental features into the neural map of space of an animal to support navigation remains incompletely understood. One solution is a rigid system for calculating distance travelled using idiothetic cues, such as path integration[9], which can operate instantly regardless of changes in environmental features. However, path integration that does not reference landmarks is prone to error accumulation[10,11]. Alternatively, a slower, less error-prone system integrates changes in environmental features through Hebbian synaptic mechanisms[12–14]. This allows neural maps of space to adjust flexibly, enabling accurate matching between distances and directions

travelled in real space with those represented by trajectories in neural space[15]. Although this system requires time and repeated experience, studies in *Drosophila* provide evidence consistent with this mechanism: Hebbian plasticity operates on the timescale of minutes to integrate the location of novel landmarks in the neural map of heading orientation in the fly[12,13].

Neither solution entirely aligns with previous works on medial entorhinal cortex (MEC) grid cells, which are active at multiple, regularly spaced spatial locations and support building a neural map of space[1,7]. Although grid cells use idiothetic cues to encode distance travelled in the dark[16], grid firing pattern regularity is sensitive to environmental changes. Boundaries[17–21] and reward locations[22,23] can distort the grid firing pattern regularity, a finding inconsistent with a rigid system based solely on idiothetic cues. In addition, geometric perturbations to familiar environments[19,24] and novel environmental geometries[25] result in distortions in grid cell patterns that persist for days[24], contrary to the rapid Hebbian plasticity solution observed in

[1]Department of Neurobiology, Stanford University School of Medicine, Stanford, CA, USA. [2]Department of Applied Physics, Stanford University, Stanford, CA, USA. [3]Howard Hughes Medical Institute, Stanford University School of Medicine, CA, Stanford, USA. [4]These authors contributed equally: John H. Wen, Ben Sorscher. ✉e-mail: giocomo@stanford.edu

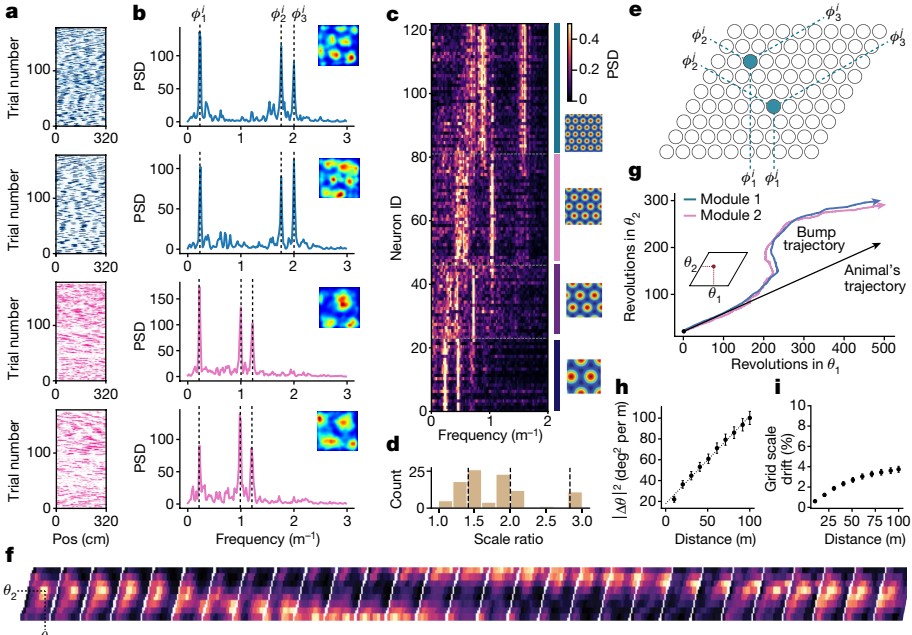

**Fig. 1 | Tracking the moment-by-moment grid cell attractor state. a**, Spike rasters for four grid cells from two simultaneously recorded modules in the dark. The dots denote spikes and are coded to match modules in panel **c**. One lap refers to 320 cm travelled. Pos, position along the track. **b**, Power spectral densities (PSDs) for cells in panel **a**, computed over 10 laps. The inset shows a 2D open field firing pattern of the same neuron, coded for minimum (blue) and maximum (red) values. **c**, Power spectra for all grid cells in a session, clustered into modules. Power spectra coded for maximum (white) and minimum (black) power, sorted dorsal (top left) to ventral (bottom left). The coloured bars indicate modules, with an inferred grid scale in a 2D open field (coded as in panel **b**; right). **d**, Histogram of inferred 2D grid scale ratios between modules ($n = 4$ animals, $n = 19$ sessions, $n = 51$ modules and $n = 1,178$ cells). The dashed lines denote powers of $\sqrt{2}$. **e**, Schematic of sorting, for each module, grid cells based on their 2D phase, $\phi_1$ and $\phi_2$, onto a neural sheet. **f**, This sorting revealed an activity bump that translated with the movement of the animal in the dark. Each frame indicates the activity of 64 co-modular grid cells within a 2-cm track bin, coded for maximum (white) and minimum (black) firing rates, averaged over 20 trials. **g**, Inferred (unwrapped) bump trajectory from two simultaneously recorded modules: blue module (37 cells) and pink module (36 cells). **h**, The 1D slice angle through the 2D lattice drifts in the dark, approximated by a diffusion constant $D = 1.16 \pm 0.07$ deg$^2$ m$^{-1}$ ($n = 4$ animals, $n = 6$ sessions, $n = 10$ modules and $n = 260$ cells). The dots denote the mean and the bars indicate the s.e.m. **i**, Grid scale remained similar in the dark ($n = 4$ animals, $n = 6$ sessions, $n = 10$ modules and $n = 260$ cells). The dots denote the mean and the bars refer to the s.e.m.

*Drosophila*. On a longer timescale (many days), the regularity of grid firing patterns can eventually return[18,26]. However, such slow plasticity cannot explain rapid and accurate navigation in novel or rapidly changing environments[27].

Here we investigated a third possibility for how mammalian spatial maps rapidly adapt to novel or changing environmental features. Using Neuropixels probes, we recorded from over 15,000 grid cells as mice navigated virtual reality environments with systematically manipulated landmarks. We then examined how grid firing patterns incorporated novel environmental features on a moment-by-moment basis in the context of attractor networks, a computational framework consistent with the emergence of grid cell firing patterns[7,28–35]. This approach revealed that the brain constructs spatial maps for navigation through a finely tuned balance of fixed and plastic connections.

## The real-time grid cell attractor state

Using Neuropixels probes, we recorded tens of thousands of MEC neurons ($n = 68,484$ neurons; Extended Data Fig. 1 and Supplementary Table 1) in head-fixed mice navigating virtual reality one-dimensional (1D) tracks. For all experiments, mice ran at least one block of trials in complete darkness. In darkness, single-cell spike trains exhibited little apparent spatial structure (Fig. 1a). However, applying the Fourier transform to the spatially binned spike train of each neuron revealed a three-peaked structure in a subset of neurons (Fig. 1b). We classified these neurons as grid cells for several reasons ($n = 15,342$; Methods). First, virtual reality-identified grid cells exhibited features consistent

with previous grid cell observations. The three-peaked structure suggested grid cell activity in 1D virtual reality was well described by a 1D slice through a 2D hexagonal lattice over short distances (approximately 10 laps)[36]. Grid cells clustered into discrete modules by their power spectra (Fig. 1c) and, in sessions with multiple modules, the inferred distance between grid firing fields (that is, grid scale) increased with a scale ratio between 1.4 and 1.7 (ref. 19; Fig. 1d). Grid cells from the same module preserved their correlation structures and phase relationships across long distances[28,30,32] (Extended Data Fig. 2). Moreover, the grid cell population in 1D virtual reality exhibited low-dimensional attractor dynamics[28,34] (Extended Data Fig. 2). Second, in a subset of mice, we recorded the same MEC neurons in virtual reality and in a 2D open-field environment ($n = 3$ animals, $n = 504$ cells). Neurons identified as grid cells in the 2D open field using established methods[34] were highly overlapping with those identified as grid cells in dark recordings (Fig. 1b and Extended Data Fig. 3).

To examine the population dynamics of grid cells in 1D virtual reality environments, we sorted grid cells from the same module (that is, co-modular) onto a 2D toroidal neural sheet (Fig. 1e). For each module, we sorted grid cells according to the phases of each cell: $\phi_{1,2,3}^i$ (note that these three phases represent only 2 d.f.; Extended Data Fig. 2). These phases were determined from the peaks of the power spectra of a given cell, as measured over 10–20 laps (320 cm of running per lap) during recordings in the dark (Methods; Fig. 1a,b and Extended Data Fig. 2). When sorted in this manner, the population activity of the grid cells revealed a 'bump' of activity that translated across the 2D toroidal neural sheet in concert with the forwards movement of

the animal (Fig. 1f). To visualize the trajectory of the activity bump over the course of a recording session, we computed the centre of mass of the activity bump at each spatial bin along the track (320-cm track binned into 2-cm bins). The moment-by-moment grid cell attractor state for a given grid module can then be calculated as the centre of mass of the activity bump. Note that the 2D sorting of co-modular grid cells was based on the first 10–20 trials of the session but the extracted phases were consistent across blocks of 10 trials (Extended Data Fig. 2). This allowed us to follow the subsequent trajectory of the activity bump over the course of the session.

First, we considered the trajectory of the activity bump in darkness, observing that it deviated from a straight-line path over long distances (Fig. 1g). As a consequence, the 1D slice angle through the 2D hexagonal lattice drifted and rotated (Fig. 1h and Extended Data Fig. 2). This drift in angle grew as the square root of the distance travelled, closely matching an angular diffusion process with diffusion constant $D = 1.16 \pm 0.07$ deg$^2$ m$^{-1}$ ($n = 4$ animals, $n = 6$ sessions, $n = 10$ modules and $n = 260$ cells). These observations are consistent with the accumulation of error in path integration estimates without sensory landmarks[16,37]. The overall distance between grid fields remained stable and drifted less than 5% over 100 m (Fig. 1i; $n = 4$ animals, $n = 6$ sessions, $n = 10$ modules and $n = 260$ cells). In recordings with multiple grid modules, activity bumps from different modules drifted together (Fig. 1g and Extended Data Fig. 4), consistent with previous work examining grid cells in the dark in a 2D open field[32]. Together, these results reveal a robust method for classifying grid cells and observing the moment-by-moment grid cell attractor state in 1D head-fixed virtual reality conditions.

## Landmarks induce one-shot grid remapping

We then extracted the moment-by-moment grid cell attractor state (that is, the bump centre of mass) to assess how environmental features influenced grid cell firing patterns. We designed a virtual reality track in which features were systematically added (the 'build-up track'; Fig. 2a and Extended Data Fig. 1). After an initial dark block, animals ran 40 trials in each of blocks 2–7, where features were cumulatively added (Fig. 2a). In block 2, automatic water rewards were dispensed. In block 3, optic flow was added to the track floor. In blocks 4–7, distinct landmark towers were added. In block 8 (20 trials), we removed the most recently added tower. In block 9, the animal again ran a dark block.

First, we considered the responses of individual grid cells to the addition of environmental features (Fig. 2b). We observed that as visual landmarks were added, spatial structure emerged within grid cell spike trains. To then assess the grid cell population response, we reconstructed the 2D toroidal neural sheet from data collected during dark running (as in Fig. 1) and plotted the trajectory of the activity bump (Fig. 2c). Plotting each coordinate on the neural sheet separately ($\theta_1$ or $\theta_2$) revealed that additional visual landmarks entrained the activity bump to whole number revolutions around the torus for each trial (Methods; Fig. 2c and Extended Data Fig. 5). Consistent with this entrainment, the activity bump took increasingly straight-line paths as landmarks were added (Fig. 2d,e and Extended Data Fig. 5). When the mouse stopped running, the activity bump remained roughly at the same 2D neural sheet location, moving once the animal started running again (Extended Data Fig. 6).

However, we also observed that the introduction of a single landmark was sufficient to induce remapping, such that the shape of grid cell tuning curves changed (Fig. 2b,f). Quantitatively, grid cells maintained high spatial correlation within, but not across, blocks containing at least one tower (Fig. 2f; Wilcoxon rank sum test, $P < 1 \times 10^{-10}$; $n = 126$ blocks, $n = 14$ sessions, $n = 5$ animals, $n = 35$ modules and $n = 747$ cells). Moreover, a moving window population vector correlation revealed that remapping occurred rapidly between trial blocks (for example, it took an average of $1.29 \pm 0.07$ trials for grid cell tuning curves to more closely resemble the average tuning curve in block 7 than in block 6;

Pearson $r$; $n = 35$ modules, $n = 14$ sessions, $n = 4$ animals and $n = 747$ cells; Fig. 2f). This number was similar on the first exposures of the animals to the track ($1.42 \pm 0.23$ trials, $n = 14$ modules, $n = 7$ sessions, $n = 7$ animals and $n = 328$ cells). Such rapid changes to grid cell tuning curves suggests that the introduction of a single visual landmark can induce one-shot (that is, single trial) remapping.

To understand the remapping of grid cells between trial blocks, we again considered grid cells as a population. First, we noted that despite remapping across trial blocks, grid cells maintained low-dimensional continuous attractor dynamics, as temporal correlations between grid cells were preserved (Pearson $r$; $n = 14$ sessions, $n = 4$ animals, $n = 35$ modules and $n = 747$ cells; Fig. 2g and Extended Data Fig. 2), consistent with previous observations of grid cells during 2D open-field remapping[26]. When we examined the trajectories of the activity bump, we noted a change in its angle or length across trial blocks (Fig. 2h,i). Together with previous observations of grid cell remapping[26], these results support the idea that grid cell remapping corresponds to the activity bump taking a new trajectory on a toroidal attractor[38]. This finding thereby provides a transparent population framework for understanding the complex reorganization of individual grid cells during a remapping event.

## Landmarks induce grid distortions

How the addition of visual landmarks to the build-up track (Fig. 2a) causes the trajectory of the activity bump to rapidly change yet then remain stable on subsequent trials (Fig. 2j–l) remains unexplored. One possibility is that landmark inputs do not shift where they project onto the neural sheet (Fig. 2j,k). In this case, landmarks influence the activity bump by causing slight deflections in its trajectory as an animal approaches a landmark. This would result in stable but distorted maps, meaning that the distance travelled by the mouse in physical space would not retain a fixed proportion to the distance travelled by the activity bump on the neural sheet (Fig. 2j,k). Another possibility is that new landmark inputs rapidly shift to project to the appropriate location of the activity bump on the neural sheet (Fig. 2l). This would result in alignment between where landmark inputs project onto the neural sheet and neural coordinate estimates, as indicated by the location of the activity bump on the neural sheet. Although such learning between landmarks and neural coordinate estimates have been reported in the fly compass system[12,13] and the mammalian head direction cell system[39], it can take minutes to accomplish. To consider these two possibilities, we designed an experiment in which mice navigated 12 virtual reality environments, each consisting of the same set of 5 visual landmarks (Fig. 3a). For a given block of 20 trials (that is, a 'trial block'), these 5 landmarks were placed at random locations, and thus the locations and order of the same landmarks differed across the 12 environments (the 'random environment track'; $n = 11$ animals and $n = 4,226$ grid cells; Fig. 3a).

Similar to the build-up track, visual landmarks on the random environment track induced distortions to the trajectory of the bump (Fig. 3b–d). This was quantified in two ways. First, we measured the anisometry of the trajectory of the activity bump, a measure of how stretched or compressed distances on the neural sheet become relative to distances travelled in physical space (Methods; Fig. 3b,c). We found that the anisometry of the bump trajectory was significantly higher in the presence of landmarks than in the dark (Fig. 3c; Kolmogorov–Smirnov test $P < 1 \times 10^{-10}$; $n = 1,680$ trials, $n = 6$ animals and $n = 7$ sessions), indicating that the trajectory of the activity bump was distorted in the presence of visual landmarks. Second, as the animal was constrained to 1D trajectories, the activity bump should follow straight lines on the 2D toroidal surface (a geodesic; Methods). However, over short distances (one to a few trials), the bump deviated substantially from straight trajectories in the presence of landmarks (Fig. 3d; Kolmogorov–Smirnov test $P < 1 \times 10^{-10}$; $n = 1,680$ trials, $n = 6$

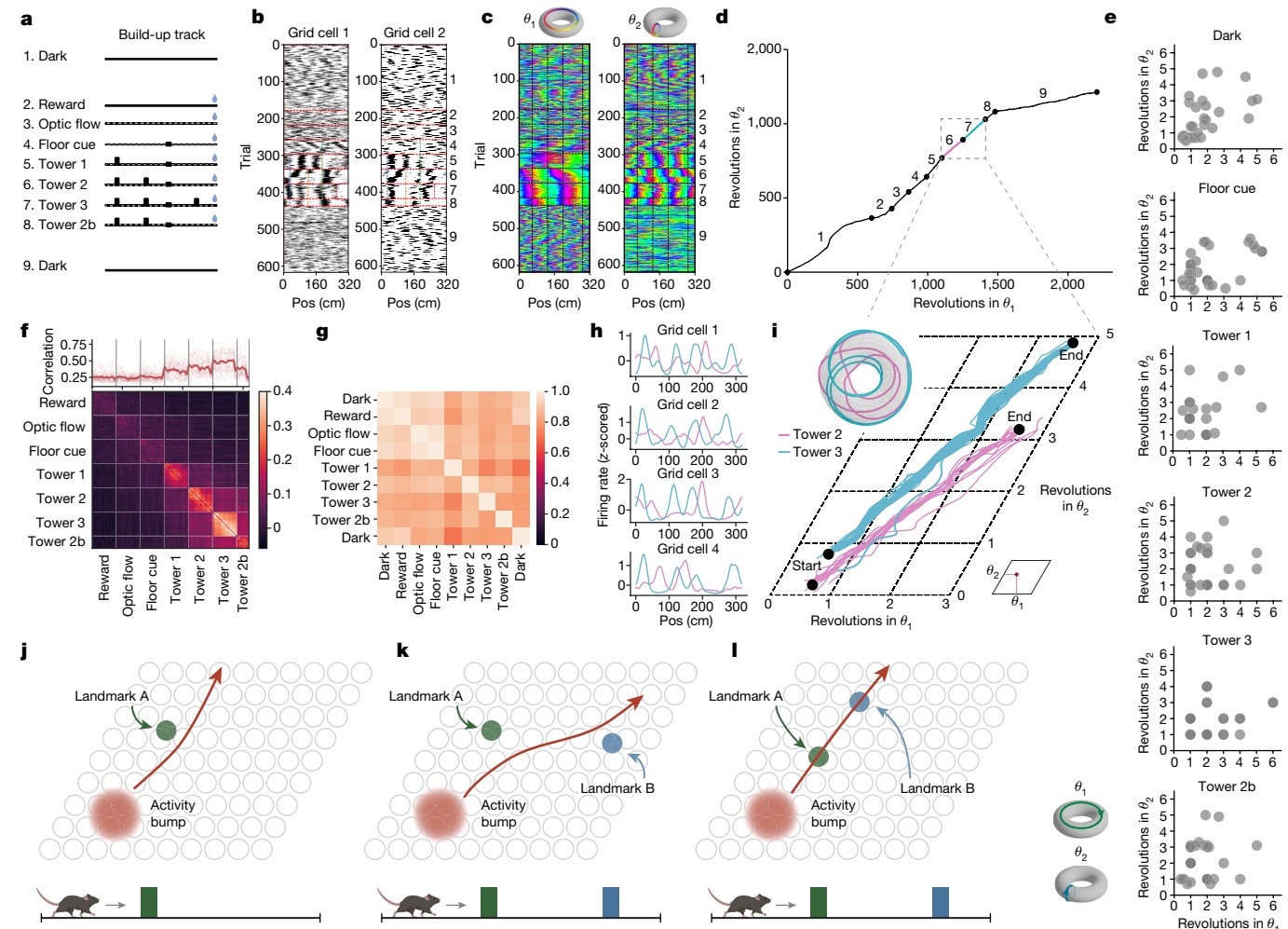

**Fig. 2 | Landmarks entrain grid cells to periodic trajectories on the torus.**
**a**, The build-up track. **b**, Two grid cells, one animal (mouse N2) and one session.
The black dots denote spikes; the red dotted lines refer to block transitions;
and the vertical coloured lines indicate landmarks. The trial block numbers
(right) are as in panel **a**. **c**, Bump trajectory for the session as in panel **b** as 2D
coordinates on the torus: $\theta_1$ (left) and $\theta_2$ (right). $n = 37$ cells. The colours
indicate the angle along each toroidal axis. **d**, Bump trajectory (unwrapped)
for one session ($n = 37$ cells). The black dots indicate block transitions, as in
panel **a**. **e**, The dots denote the number of activity bump revolutions along
each toroidal axis for one lap ($n = 4$ animals, $n = 14$ sessions, $n = 35$ modules and
$n = 747$ cells). **f**, Moving-window correlation (Pearson $r$) of population activity
(top; window size of 20 m; the thick line denotes the mean and the thin lines
indicate individual blocks; $n = 14$ sessions, $n = 4$ animals, $n = 35$ modules and
$n = 747$ cells; top). Correlation (Pearson $r$) of grid cells across trials ($n = 14$

sessions, $n = 4$ animals, $n = 35$ modules and $n = 747$ cells) is also shown (bottom).
**g**, Grid–grid pairwise temporal correlation matrices for spike trains within
each block, the same session as panel **b**. Colour denotes correlation (Pearson $r$).
**h**,**i**, Same session as panel **b**. Tuning curves for four grid cells (**h**) and single-trial bump trajectories (unwrapped; **i**) are shown. The
lines indicate bump trajectories on individual trials. The bold blue line denotes
the first tower 3 trial. Average bump trajectories on the torus is also shown
(inset). **j**–**l**, Cartoon of the influence of landmarks on the grid cell attractor
state, represented as a bump of activity (red) on a sheet of grid cells (grey
circles), which follows the red trajectory as the animal (bottom) traverses the
track. A landmark (green) weakly pins the attractor bump (**j**). A new landmark
(blue) deflects the bump onto a new trajectory (**k**). Plasticity could allow
landmarks to update their pinning phases (**l**). The mouse in panels **j**–**l** was
created using BioRender (https://biorender.com).

animals and $n = 7$ sessions). This suggests that visual landmarks 'tug'
the activity bump off straight-line trajectories, consistent with the
finding that environmental boundaries locally distort grid cell firing
patterns[17,20,21,23–25].

We refer to this tugging as 'weak pinning', because landmarks could
entrain the bump to periodic trajectories (Fig. 2e and Extended Data
Fig. 7) but were not strong enough to fully 'pin' the bump to a par-
ticular position on the neural sheet each time the animal passed the
corresponding visual landmark across the 12 environments (Fig. 3e).
This results from the fact that the trajectory of the bump in any given
environment was influenced by both the landmarks and the history
of the bump. In other words, the bump could not deviate very far
from the trajectory it took in the previous environment. This history
dependence was further evidenced in a subset of sessions in which we

repeated blocks of trials in the same virtual reality environment twice,
separated by several blocks of visually distinct environments ($n = 14$
build-up track sessions, $n = 4$ animals, $n = 35$ modules and $n = 747$ cells).
We found that grid cells developed different maps in the two repeated
environments, despite them being visually identical (approximately
50% of the time; Extended Data Fig. 5). These results together point to a
model in which landmarks have a weak-pinning influence on the bump
trajectory, resulting in distorted but stable grid cell representations
for a given environment.

## Fixed, weak pinning predicts remapping

Whether weak landmark pinning provides enough structure to the
trajectory of the bump to predict the grid cell firing pattern in a given

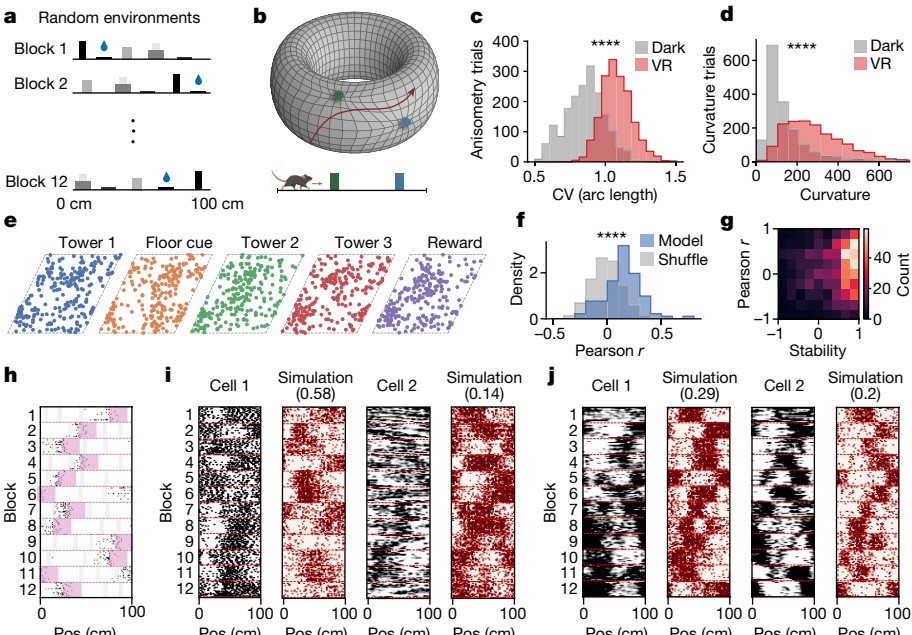

**Fig. 3 | Distortions underlying grid cell remapping. a**, The random environment track. **b**, Schematic of anisotropy and curvature (red line) for bump trajectories. The grid lines indicate unit distance intervals in physical space. Rectangles (below) represent two different landmarks, and circles (above) represent their corresponding pinning phases. The mouse was created using BioRender (https://biorender.com). **c**, Bump trajectories have significantly higher anisometry with landmarks than with darkness (two-sided Kolmogorov–Smirnov test $P < 1 \times 10^{-10}$; $n = 1,680$ trials, $n = 6$ animals and $n = 7$ sessions), as quantified by the coefficient of variation (CV) in the arc length of neural trajectories corresponding to 2-cm intervals in real space, computed over windows of 16 m. VR, virtual reality. **d**, Bump trajectories have significantly higher curvature with landmarks than with darkness (two-sided Kolmogorov–Smirnov test $P < 1 \times 10^{-10}$; $n = 1,680$ trials, $n = 6$ animals and $n = 7$ sessions). **e**, Location of the bump on the neural sheet each time the animal passed a landmark across all trial blocks in panel **a**, for a single grid cell module and a single session. **f**, Histogram of the model predictivity (Pearson $r$ between the predicted and true tuning curves in each block, averaged over blocks) across all cells, compared with a random shuffle (two-sided Kolmogorov–Smirnov test $P = 3.58 \times 10^{-18}$; $n = 91$ cells, $n = 3$ animals, $n = 3$ sessions and $n = 3$ modules). **g**, Heatmap of the model predictivity (average Pearson $r$ between predicted and true tuning curves on held out blocks) and grid cell stability, plotted for each block and each cell (average Pearson $r$ correlation of tuning curves between the first and second half of a block of trials; $n = 3$ animals, $n = 3$ sessions, $n = 3$ modules and $n = 91$ cells). **h**, Licking behaviour during a random environment track session. The black dots denote licks; the pink shaded areas indicate the reward zone; and the coloured bars refer to landmarks, coloured as in panel **e**. **i,j**, Raster plots of real (black) and simulated (red) neurons from two recordings across two mice (two neurons per mouse). Pearson $r$ between simulated and true tuning curves are also shown (top).

environment is yet to explored. To answer this, we considered grid cells in 11 of the random environments and asked whether we could predict their firing in a twelfth held-out environment. First, we noted that animals remained task engaged, even though the landmarks moved across environments (Fig. 3h and Extended Data Fig. 1). Second, we modelled grid cell activity using a simple model of an activity bump moving on a 2D toroidal sheet (Methods; Fig. 3f,g). In the absence of landmarks, the model bump took trajectories whose angle drifted according to the diffusion constant $D$, extracted from neural data (Fig. 1h). When the simulated animal came within 50 cm of a landmark, the model bump was tugged towards the 'pinning phase' of the landmark on the 2D toroidal sheet (Fig. 3f,g). To estimate the pinning phase for each landmark from the neural data, we computed the centroid position of the activity bump on the neural sheet, conditioned on the animal being at that landmark in 11 of the 12 environments.

After identifying a single fixed pinning phase for each landmark, we simulated the bump trajectory on the held-out environment. We used this bump trajectory to simulate spike trains for each model grid cell by computing a firing rate based on the proximity of the 2D phase of the grid cell to the location of the bump at each timepoint. The model grid cell spike trains showed a remarkable resemblance to experimentally observed grid cell spike trains (Fig. 3i,j and Extended Data Fig. 8). To quantify model performance, we measured the average correlation between model grid cell spike trains and neural grid cell spike trains. We found that this simple model was able to predict the firing patterns of many grid cells with a Pearson correlation higher than chance

across held-out environments, and that the best-predicted cells were the most stable (Fig. 3g). We compared these correlations with two shuffled conditions: randomly shifting the simulated tuning curves (Fig. 3f), and scrambling the landmark locations used to train the model (Extended Data Fig. 8). The simple model consistently outperformed both null models.

Motivated by previous theoretical work and evidence from *Drosophila*[12–15], we incorporated a Hebbian learning mechanism with a tuneable learning rate into our model. We found that the learning rate that best explained the neural data was zero (Extended Data Fig. 8). This suggests that grid cell firing across novel environments was best explained by a fixed circuit that exhibits little plasticity for changes in the location of landmarks. In this framework, the rapid changes in the bump trajectory reflect the weak, fixed pinning of landmark inputs to the grid cell network.

## Rapid, stable maps and flexible behaviour

A fixed landmark-to-grid cell network allows grid cells to form rapid, stable firing patterns when new landmarks are observed, which could support short timescale behaviours. However, such fixed networks also distort the regularity of grid cell firing patterns, raising the question of how well animals can navigate with distorted grid cell firing patterns. To investigate this, we developed a virtual reality task in which animals had to use both distance run and landmark positions to find a hidden reward zone (the 'hidden reward task').

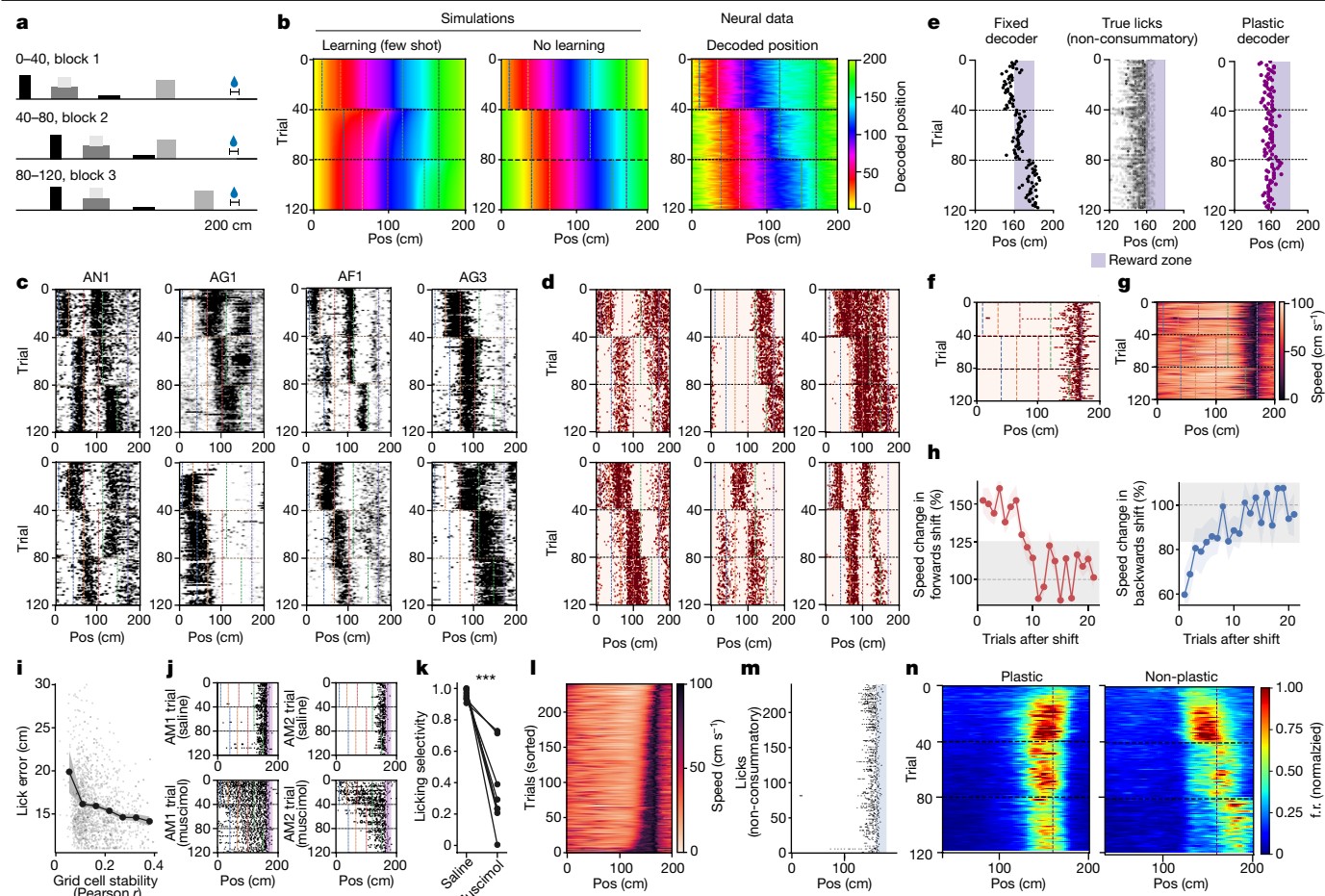

**Fig. 4 | Rapid learning with fixed maps. a**, Hidden reward task, forwards shift condition. The water drop denotes the reward zone. **b**, Decoded position from simulated grid cells: few-shot learning (left) and no learning (middle). Decoded position (linear decoder trained on grid cells), averaged over 29 modules, 10 sessions, 5 animals and 1,325 cells (right). The landmark shift trial is indicated by the horizontal lines. Landmark is denoted by the vertical lines. **c**, Grid cell rasters. The animal ID (top) is labelled as in panel **b**. **d**, Model simulations with random landmark pinning. **e**, Predicted lick locations, fixed decoder trained on all blocks (left). The dots denote licks. Non-consummatory licks for all sessions and mice (middle). Predicted lick locations, decoder trained with BTSP (right). **f,g**, Lick (**f**) and running speed (**g**) rasters for one session. **h**, Running speed change in the 10-cm window preceding the reward following a tower shift, averaged over 5 animals and 16 sessions (the line denotes the mean and the coloured shaded areas indicate s.e.m.). The grey shaded area indicates the 95% confidence interval of trial-to-trial speed variability preceding the tower shift.

**i**, Licking error (Methods) decreased with more stable grid cells (trial-by-trial Pearson $r$ correlation of population activity within a block; the grey points indicate trials; Pearson $r = -0.20$, $P < 1 \times 10^{-10}$; the black dots denote binned average licking error, and the shaded area indicates bootstrapped 95% confidence interval; $n = 1,668$ trials, $n = 7$ animals, $n = 16$ sessions, $n = 50$ modules and $n = 2,099$ cells). **j**, Non-consummatory lick rasters for saline (top) and muscimol (bottom) sessions. Dashed coloured lines indicate the positions of landmarks and reward. **k**, Licking selectivity. Connected dots indicate pairs of sessions from individual mice ($P = 0.0004$, two-sided permutation test with 1,000 shuffles; $n = 7$ session pairs and $n = 6$ mice). **l,m**, Speed raster (**l**), as in panel **g**, sorted by average speed preceding the reward zone. The speed varied but the licking centre of mass (**m**) was consistent. **n**, Simulated CA1 place cell using BTSP. The place cell adapts (left), and when plasticity is knocked out, the place field shifts (right).

Animals were trained on a 200-cm track, with 4 visual landmarks and an unmarked 10-cm wide reward zone 15 cm past the last tower (baseline block). To prevent animals using time elapsed or distance run since the last reward, the distance from the track start to the first tower varied randomly on each trial. After training, animals licked before the reward zone and slowed down after the last tower, indicating that they had learned the location of the hidden reward zone (Extended Data Fig. 9).

After training, we recorded neural activity during the hidden reward task on the same track. In the first block (40 trials), the track had the same 4 landmarks as in training, with an unmarked 10-cm wide reward zone starting 50 cm past the last tower. In the second block (40 trials), the first 3 landmarks moved forwards (towards the reward) by 35 cm. In the final block (40 trials), the last landmark was shifted forwards by 35 cm. We refer to this as the 'forwards shift condition' (Fig. 4a). To ensure symmetric behavioural effects, we performed

these three blocks in reverse ('backwards shift condition'; Extended Data Fig. 9).

On the basis of our previous observations (Figs. 2 and 3), landmark shifts should distort grid cell firing patterns. However, if the animal estimates its spatial position from the regularity of these grid patterns, the grid cells must recover from landmark-induced distortions, or the animal will lick too early or late when the towers are shifted forwards or backwards and miss the reward zone. We simulated the possible outcomes of this experiment with (Fig. 2l) and without (Fig. 2k) learning of landmark inputs to grid cells, using the same model as in Fig. 3. With learning (Fig. 4b, left), the decoded position of the simulated animal was distorted by the landmark shift, but returned to its original position after a few trials (that is, few-shot learning condition). Without learning (Fig. 4b, middle), the decoded position of the simulated animal remained distorted. We then repeated this procedure for our experimental data by training a linear decoder on grid cell

population activity recorded during the hidden reward experiment, before landmark shifts. Consistent with the random environment track results, the neural data (Fig. 4b, right) closely resembled the simulated no-learning outcome (Fig. 4b, middle): the estimated position of the animal based on the decoded grid cell population activity distorted with the landmark shift and never recovered. This distortion was evident in single-grid cell activity across animals and sessions (Fig. 4c) and in simulated model cells (Fig. 4d). Including all recorded MEC neurons ($n$ = 2,630 neurons and $n$ = 10 mice) did not change the observed distortion (Extended Data Fig. 10). Indeed, no fixed linear decoder could correctly decode the position of the animal across blocks, including a decoder trained with neural activity from all three blocks (Fig. 4e, left). Moreover, the decoder predicted that the animal would lick in different spots before and after the landmark shift (Fig. 4e, left). However, animals rapidly learned to adapt their behaviour after the shift (Fig. 4e, middle). Unlike the neural activity, animals returned to their previous performance accuracy after fewer than 10 trials in the new environment (Fig. 4e–h). This was the case for both forwards and backwards shift conditions.

What could explain this difference between the long-term (more than 10 trials) distortion of grid cell firing patterns and the short-term (fewer than 10 trials) disruption of animal behaviour remains unknown. One possibility is that grid cells do not support task performance. However, three pieces of evidence support a role for MEC and grid cells in task performance. First, grid cell spatial stability correlated with behavioural performance (Fig. 4i). Second, bilateral muscimol injection into the MEC significantly silenced neural activity (Extended Data Fig. 9) and impaired behavioural performance compared with saline injection (Fig. 4j,k and Extended Data Fig. 9). In a recording with incomplete silencing of the MEC, behavioural performance was also less impaired (Extended Data Fig. 9). Third, distance-based strategies better predicted the behaviour of the animal than time-based strategies (Fig. 4l,m). If animals used a time-based strategy, they should lick late on trials in which they run quickly and vice versa. However, we did not observe this type of relationship.

Another possibility is that although landmark connectivity to the grid cell network is fixed, grid cell connectivity to a downstream decoder supporting behaviour might be plastic. Such a decoder would have to adapt to a distorted grid cell code after one or few trials. Recent studies have identified a rapid, behavioural timescale synaptic plasticity (BTSP) mechanism that governs place field formation in the CA1 and is driven by a signal from the MEC[6,40,41]. To test the feasibility of this mechanism, we trained a linear decoder using the BTSP learning rule on recorded grid cell activity (Methods). Unlike the fixed decoder, this BTSP decoder quickly adapted to landmark shifts and reliably predicted the reward location across environments (Fig. 4e, right). This results from the ability of the BTSP to act on the timescale of seconds, enabling larger temporal offsets between MEC input regarding spatial position and input regarding environmental features such as reward location. Moreover, the BTSP decoder predicts how the firing field of a hippocampal place cell would respond to a tower shift in the hidden reward experiment if it adapted according to a BTSP learning rule (Fig. 4n), and how place cells would respond if plasticity between the MEC and the CA1 were absent.

## Discussion

Utilizing Neuropixels recordings from thousands of grid cells, we tracked the real-time evolution of the grid cell attractor state as landmark features were added or changed. MEC grid cells rapidly exhibited new, stable spatial firing patterns in novel or changed environments after a single trial. Visual landmarks entrained grid cells to periodic firing patterns. Consequently, these patterns were distorted such that distances travelled in real space did not mirror distances travelled in neural space. These distortions followed dynamics consistent with a model in

which landmark input to fixed points on the neural sheet do not change over short timescales (that is, minutes and hours). Together, these results reveal how grid cells, combined with a downstream decoder using a biologically plausible learning rule[6,41], provide a spatial map that is both rapid and stable enough to enable one-shot or few-shot learning in novel or changing environments.

Although our model can account for various remapping phenomena, the mechanistic details upstream and downstream of grid cells remain unclear. Upstream, the specific nature of landmark inputs to the grid cell neural sheet remains unknown. This input could be provided by any neuron encoding landmark information, for example, MEC non-grid spatial[42], border[2,5] and object vector cells[4], lateral entorhinal cortex neurons that encode object locations[43], V1 neurons active near visual landmarks[44,45], or subicular boundary vector[46] and corner cells[47]. The model here assumes simple connectivity between landmarks and grid cells, with each landmark sending output to a single preferred 'phase' on the neural sheet. Future work could consider broader fields of connectivity between landmarks and grid cells. Similarly, although our results indicate that rapid plasticity to support flexible behaviour probably occurs downstream of the MEC, the exact plasticity mechanism will require future investigation. One possible mechanism is BTSP[6,40,41], which induces the formation of new hippocampal place fields on timescales of one or a few trials, and is directed by a spatial template provided by the MEC[6,40,41]. Our experiments show that the MEC code provides a sufficiently stable template for adaptive behaviour when paired with downstream BTSP. Another potential mechanism is Hebbian learning in an associative network[8,48], in which a downstream recurrent circuit associates the grid cell spatial code with a sensory cue, such as a landmark or reward. Future work recording both MEC and hippocampal activity simultaneously could potentially offer additional insight. Finally, regions such as the lateral entorhinal cortex, which contains information about rewarded locations[49,50], could also support flexible behaviour over timescales in which there are distortions in MEC grid cell firing patterns.

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

## Methods

### Experimental model and mouse details

All techniques were approved by the Institutional Animal Care and Use Committee at Stanford University School of Medicine. A total of 19 female and 3 male C57BL/6 mice were used for this paper. Sample size was determined by what is conventionally required. No blinding or randomization was performed. Acute electrophysiological recordings were conducted on 15 female C57BL/6 mice 12–24 weeks of age at the time of the first surgery (headbar implantation). See Supplementary Table 1 for a breakdown of which mice were used in which experiments. Mice were housed with littermates on a 12-h light–dark cycle and had ad libitum access to food and water before the headbar implantation surgery. Mice were housed in an animal facility with a temperature range of 20–26 °C. Humidity ranged between 30% and 70%. After the craniotomy surgery, mice were housed singly to prevent disturbance of the surgical site. The remaining four female mice were used for fluorescent muscimol histology only, and three male mice were used for chronic electrophysiological recordings. All animal experiments were conducted in accordance with the guidelines and regulations of the National Institutes of Health and Stanford University under APLAC protocol 27694.

### Virtual reality environments

The virtual reality recording setup was nearly identical to that in ref. 51. Mice were head-fixed and ran on a 15.2-cm diameter foam roller (ethylene vinyl acetate) constrained to rotate about one axis. Rotation of the cylinder was tracked with a rotary encoder (Yumo 1024P/R) and relayed to a microcontroller (Arduino UNO) using custom software. The virtual reality environment was developed using custom code in Unity 3D. The virtual scene was displayed on three 24-inch monitors surrounding the mouse. Water rewards were delivered through Tygon tubing attached to a custom-built lickport. The lickport consisted of a lick spout and an infrared light beam to detect licks. Water was delivered via opening of a solenoid valve. Phase 3B Neuropixels 1.0 silicon probes[52] were mounted on a custom-built rotatable mount and positioned using a micromanipulator (Sensapex uMp). Virtual tracks specific to each task are detailed below.

**Build-up track.** The build-up track was 320 cm in length with a 300-cm visual lookahead. Upon reaching the end of a track, mice were seamlessly teleported to the beginning of the track without any visual discontinuity. The experiment consisted of nine blocks. The first and last blocks occurred in the dark for a minimum of 10 min each, where no water rewards were given and no visual features were displayed. Blocks 2–7 consisted of 40 trials each, and block 8 consisted of 20 trials. Starting in the second block, water rewards were automatically dispensed at the end of the track. In the third block, a repeating checkerboard optic flow pattern was displayed on the ground. In blocks 4–7, visually distinct tower landmarks were cumulatively added. In the eighth block, the tower landmark added in the previous block (block 7) was removed. All tower landmarks were displayed on either side of the animal.

**Random environment track.** The random environment track was 100 cm in length with a 100-cm visual lookahead. Unlike the build-up track, this and the following track did not feature a seamless teleport. The experiment consisted of 14 blocks. The first and last blocks occurred in the dark and consisted of at least 100 trials where no water rewards were given and no visual features were displayed. In the blocks 2–13, consisting of 20 trials each, a set of 5 familiar visual landmarks were pseudo-randomly rearranged. All visual landmarks were visually distinct from one another. Three of the landmarks were tower landmarks, and two were ground landmarks. One of the ground landmarks was associated with a water reward. For each random rearrangement, the distance between any two landmarks had to be at least 15 cm apart and the position of the first landmark had to be at least 15 cm from the start of the track.

**Hidden reward track.** The hidden reward track was 200 cm in length with a 100-cm lookahead. The track consisted of the same visual landmarks in the random environment track except for the absence of the reward cue. Thus, there were four visual landmarks on the track (three tower landmarks and one ground landmark). The positions (in cm) of the landmarks were 40, 65, 100 and 145 in the baseline condition. Animals could lick anywhere between 160 cm and 170 cm to trigger a water reward. If an animal failed to lick within the zone, they received an automatic reward at 170 cm. Note that for one animal, the hidden reward task was conducted on a 240-cm track instead of a 200-cm track. On this track, all landmarks and the reward zone were pushed back (closer to the end of the track) by 40 cm. To encourage animals to pay attention to their position relative to landmarks, and not simply path integrate a fixed distance between rewards, an offset value was drawn from a uniform distribution between 0 cm and 30 cm on each trial. This offset value was added to the positions of each of the landmarks and the reward zone, resulting in a variable inter-trial distance before encountering the first landmark. This offset value is not shown in Fig. 4 for visualization clarity. The experiment consisted of seven blocks: A, B, C, D, C, B, A. Block A was the same landmark arrangement as the training track and served as the baseline condition. Block B was identical to A, except the last visual landmark was moved 35 cm closer to the start of the track. Block C was identical to block B except the first three landmarks were also moved 35 cm closer to the start of the track. Block D occurred in the dark and consisted of at least 100 trials where no water rewards were given and no visual features were displayed. For the second set of shifts (C, B, A), the landmark arrangement in block C was used as the baseline condition. All blocks except block D consisted of 40 trials each.

### Freely moving environment

Chronically implanted mice were trained and recorded in a separate room kept at 5 lux, the same illumination as the virtual reality environment. The data acquisition apparatus consisted of a 165 × 225-cm frame surrounded by two black curtains and two walls with large distal visual cues. A 75-cm square open field with 30-cm walls and a single blue cue on the top wall was placed in the centre of the frame 75 cm off the ground.

### Behavioural training and handling

Before headbar implantation, mice were given an in-cage running wheel (K3570 and K3251, Bio-Serv). Mice were handled for at least a day before headbar implantation. Handling included holding each mouse for at least 30 s before setting them on the running wheel to encourage their use of it.

After headbar implantation, mice were injected with Rimadyl (Covetrus, 5 mg kg$^{-1}$) and Baytril (VetOne, 10 mg kg$^{-1}$) daily for 3 days. Then, they restricted to 0.8 ml of water per day. Mice were weighed daily to ensure they stayed above 85% of their baseline weight. After at least 1 day of water deprivation, mice began training on the virtual reality setup. All mice received the same initial basic training. The first stage of training acclimated mice to head fixation and running on the virtual reality wheel. No virtual reality visual cues were presented on the monitors. The virtual reality wheel was manually moved to encourage walking on the setup. As the wheel was moved, a 1-ml syringe with water was manually brought closer to the mouse. After a half revolution of the wheel, the syringe dispensed water.

After mice could run on their own (more than half a revolution of the virtual reality wheel), the second stage of training began. This stage acclimated mice to a lickport, which dispensed water (approximately 1.5–2 µl at a time) and detected licks with an infrared beam sensor. The dispensation of water was associated with the sound of a solenoid valve

opening. After mice reliably licked upon hearing the solenoid click, they proceeded to task-specific training, as described below.

**Build-up track.** Mice continued training in the absence of any virtual reality visual features. They were trained on a training track that was completely dark. The track contained an automatic water reward that was given after a certain distance was run. This distance was increased adaptively within and across training sessions to encourage consistent running in the dark. To avoid training mice to encode any particular distance, the average distance required to receive a water reward was different from session to session, ranging between 400 cm and 1,000 cm between rewards. Mice were considered ready for recordings when their average speed exceeded 250 virtual cm s$^{-1}$ for several days.

**Random environment track.** Mice were introduced to the 5 visual landmarks that would be used in the experiment by running on a virtual reality track with a specific arrangement of the landmarks for 100 trials per day for 3 consecutive days. Then, the day before recording, mice were first trained with the same arrangement of landmarks for 20 trials followed by 11 blocks of 20 trials each where the landmark locations were pseudo-randomly shuffled. This shuffling of landmarks was done to ensure mice still licked at the reward cue. If a mouse did not complete 100 trials in less than 1 h, the original training was completed with the same initial arrangement of landmarks. Mice were ready for recordings when they could run 100 trials in less than 1 h.

**Hidden reward track.** Upon completion of recordings in the random rearrangement experiment, seven mice were used for the hidden reward experiment. The visual landmarks in both tracks were identical except for the absence of a visible reward cue in the hidden reward experiment. Training proceeded by giving mice an automatic water reward at the end of a hidden reward zone. Initially, the hidden reward zone was 30 cm in width so that any licks within 30 cm of the end of the zone would trigger a water reward to be dispensed. The reward zone was gradually decreased in size over training sessions until it was 10 cm in width. To maintain running behaviour, animals received an automatic reward at the end of the reward zone if they failed to lick within the reward zone. Mice were ready for recordings when they triggered water rewards with licks within the reward zone before the automatic reward on at least 70% of trials.

**Open field.** Mice used in chronic electrophysiology recordings experienced the build-up track training and also open-field exposure each day. Mice were first habituated to the data acquisition room and experimenter, and were encouraged to explore a 50-cm open field containing several objects for 30 min with their littermates for 2 days. Then, they explored a 75-cm open field with vanilla sprinkles scattered throughout for 30 min daily for the remainder of the virtual reality training (7–9 days).

## Surgeries

For all surgeries, a mixture of oxygen and isoflurane was used for induction (4%) and maintenance (0.5–1.5%) of anaesthesia. After induction, buprenorphine (0.05–0.1 mg kg$^{-1}$) was injected subcutaneously. Upon completion of surgery and for 3 additional days after, Baytril (10 mg kg$^{-1}$) and Rimadyl (5 mg kg$^{-1}$) were subcutaneously injected.

**Headbar surgery.** The first surgery consisted of attaching a stainless steel headbar, implanting a gold ground pin and making fiducial marks on the skull. The headbar was cemented to the skull with Metabond (S380, Parkell). The ground pin was implanted roughly 2.5 mm anterior and 1.5 mm lateral of bregma. Fiducial marks were made at ±3.3 mm relative to midline and 3.7 mm posterior of bregma. These marks served to guide insertion of the Neuropixels probes and microsyringes for muscimol infusion.

**Bilateral craniotomies.** Following completion of training, bilateral craniotomy surgeries were conducted. Craniotomies removed a small portion (approximately 200 μm along the mediolateral axis and approximately 300 μm along the anteroposterior axis) of the skull posterior to the fiducial marks and exposed the transverse sinus. Craniotomies were covered with a silicone elastomer (KWIK-SIL, World Precision Instruments). Then, a small plastic well was implanted over each craniotomy and affixed with dental cement. Immediately following surgery, dexamethasone (2 mg kg$^{-1}$) was injected subcutaneously.

**Microsyringe infusions.** For the hidden reward task, bilateral injections of muscimol (M15223, Sigma-Aldrich) or saline control in the MEC were performed in seven mice. In addition, two of the seven mice also received BODIPY TMR-X fluorescent muscimol (M23400, Thermo Fisher) injections on a separate day to confirm the neural and behavioural effects of fluorescent muscimol compared with untagged muscimol. An additional four mice received bilateral injections of fluorescent muscimol using identical coordinates to the seven mice in the MEC without behaviour or Neuropixels recording, which showed consistent MEC microinfusion targeting with minimal spread to the parasubiculum. See Supplementary Table 1 for a detailed list of all the mice in this study, which sessions they ran and which drugs they received.

Of muscimol or BODIPY TMR-X muscimol conjugate, 1 mg was diluted in 4.3 ml and 0.82 ml, respectively, of filtered 1X PBS (BP3991, Fisher Scientific), resulting in a final concentration of 2.0 mM.

Before infusion of muscimol or saline control, craniotomies were inspected and cleaned with sterile saline if needed. Bilateral infusions were achieved using a 35-gauge needle (NF35BV-2, World Precision Instruments) and a 10-μl Hamilton syringe (NANOFIL, World Precision Instrument). The syringe was angled 14° from vertical. Then, the tip of the needle was positioned ±3.3 mm mediolaterally relative to the midline, 100–150 μm in front of the transverse sinus and touching the dura surface. Then, the syringe was advanced into the brain at a rate of 8.3 μm s$^{-1}$ using a Neurostar robot stereotaxic. Of either form of muscimol or saline control, 220 pmol or 110 nl was injected at 334 nl min$^{-1}$ using an ultra-micropump (UMP3, World Precision Instruments) at each of the four sites along the dorsoventral axis in the MEC. These sites were 1,200 μm, 1,950 μm, 2,700 μm and 3,450 μm below the dura. Following injection at each site, the syringe was left in place for 1 min to allow for sufficient diffusion of the drug.

**Chronic implant surgery.** A detailed protocol is available[53] and described briefly here. Before surgery, a Neuropixels 2.0 (Imec)[54] was sharpened (Narishige). Ground and reference pads were shorted and connected to a male gold pin. The probe assembly (body piece, wing, front and back flex holders, and dome) was 3D printed (Formlabs). The body piece was affixed to the wing by screws and glued to the back of the probe. A skull screw was connected to a female gold pin via a silver wire.

Following completion of training, implantation surgeries were conducted for the four mice used in the chronic electrophysiology studies. A craniotomy was made over the left or right MEC, as described above. The probe was sterilized in isopropyl alcohol and dipped in DiD (V22889, Thermo Fisher). The probe was dipped 10–15 times with 10 s in between dips. The probe was attached to a stereotactic robot (Neurostar), body piece oriented rostrally and parallel to the headbar, and inserted at 10° from vertical, tip pointed rostrally, as close to the sinus as possible. The probe was advanced at 0.5 mm s$^{-1}$ to 800-μm angled depth below the brain surface, then at 3.3 μm s$^{-1}$ to an angled depth of 3,200–3,400 μm. The craniotomy was filled with silicone gel (Dow Corning). The male pin of the ground-reference wire from the probe was attached to the female pin of the ground screw and secured by UV-cured acrylic (Pearson Dental). The wing was securely attached to the headbar and skull, creating a ring from the side of the well, across the wing and around to the ground screw, by dental cement (Lang Dental). The dome was attached to the posterior border of the skull, attached by UV-cured acrylic and

secured by dental cement. The stereotactic holder was released from the probe and removed. The flex cable holders were placed just above the body piece with the tab slot caudal, closed with electrical tape and attached to the antennae of the dome with dental cement. The entire implant was covered with copper tape and the flex cable was folded over into the tab slot, which was closed with electrical tape.

## Acute electrophysiological data collection

All recordings took place after the mice had recovered from bilateral craniotomy surgeries (a minimum of 16 h later). Immediately before recording, the Neuropixels probe was dipped in one of three dyes (DiD, DiI or DiO; V22889, Thermo Fisher). The probe was dipped 10–15 times with 10 s in between dips. If three recordings had already been performed in one hemisphere of the brain, no dye was used for subsequent recordings in that hemisphere. Dyes were used to reconstruct the probe location for undyed recordings, as described below. Then, mice were head-fixed on the virtual reality setup, and the craniotomy sites were exposed and rinsed with saline. The Neuropixels probe was angled 12–14° from vertical and positioned using the fiducial marks made during the headbar surgery and 50–200 μm anterior of the transverse sinus. The mediolateral position of the probe relative to the medial edge of the fiducial mark, the anteroposterior position relative to the transverse sinus and the angle relative to vertical were all recorded for replicability and probe localization (see 'Histology and probe localization' below). The 384 active recording sites on the probe were the sites occupying the 4 mm closest to the tip of the probe. The reference and ground were shorted together and the reference electrode was connected to the gold pin implanted in the skull. Then, the probe was advanced into the brain using a Sensapex micromanipulator, typically at 5 μm s$^{-1}$ but no faster than 10 μm s$^{-1}$. Insertion of the probe continued until it reached the skull or until several channels near the tip of the probe were quiet. Then, the probe was retracted 100–150 μm and allowed to settle for at least 15 min before recording. Finally, the craniotomy site was covered in silicon oil (378429, Sigma-Aldrich).

Raw signals were filtered, amplified, multiplexed and digitized on-probe. SpikeGLX (https://billkarsh.github.io/SpikeGLX/) was used to sample voltage traces at 30 kHz and filter between 300 Hz and 10 kHz for the action potential (AP) band. SpikeGLX was also used to acquire auxiliary signals. Each Unity virtual reality frame emitted a transistor–transistor logic (TTL) pulse, which was first relayed to an Arduino UNO, which was then relayed to an auxiliary National Instruments data acquisition card (NI PXIe-6341). These auxiliary signals were used to synchronize virtual reality data with electrophysiological traces.

At the end of each recording, both craniotomies were inspected, rinsed with saline and then covered in Kwik-Sil. The probe was rinsed and soaked for at least 10 min in a 2% solution of Tergazyme (16-000-116, Fisher Scientific) in deionized water and then rinsed with deionized water alone.

## Chronic electrophysiological data collection

Mice were food restricted to 85–90% of baseline weight. During each chronic electrophysiology session, mice ran head-fixed on a dark, featureless 1D track with no water rewards for at least 140 trials at 200 cm per trial. Immediately afterwards, they explored a 75-cm open field with vanilla sprinkles scattered throughout for 60–70 min. RGB video was captured at 50 fps with 10 pixels cm$^{-1}$ resolution (BFS-U3-23S3C-C, FLIR Blackfly) using a custom Python script using FLIR Spinnaker API.

## Histology and probe localization

After the last recording session, mice were euthanized with an overdose of pentobarbital and transcardially perfused with PBS followed by 4% paraformaldehyde. Then, brains were extracted and stored in 4% paraformaldehyde for at least 24 h and then transferred to 30% sucrose solution for at least 24 h. Finally, brains were frozen and cut into 100-μm sagittal sections with a cryostat, stained with DAPI, and imaged using widefield microscopy (Zeiss Axio Imager 2). All sections that contained dye fluorescence were aligned to the Allen Mouse Brain Atlas using software written in MATLAB[55]. Then, probe tracks were reconstructed by localizing dye fluorescence in each section. Probe locations for recordings without dyes were estimated relative to previous recordings, using two common reference locations: mediolateral distance relative to the medial edge of the fiducial mark, and distance anterior of the anterior edge of the transverse sinus.

## Quantification of virtual reality behaviour

Anticipatory licking (as in Fig. 4k, for example) was calculated using a standard method for behavioural selectivity[56,57]. This metric is calculated as the following:

$$\text{lick}_{\text{ant}} = \frac{\text{lick}_{\text{near}} - \text{lick}_{\text{far}}}{\text{lick}_{\text{near}} + \text{lick}_{\text{far}}}$$

where lick$_{\text{near}}$ represents the average lick rate in a zone starting one-tenth of the length of the track before the reward zone and ending at the start of the reward zone, and lick$_{\text{far}}$ represents the average lick rate in a zone starting one-tenth of the length of the track before the centre of the track and ending at the centre of the track. This metric ranges from −1 to +1, with perfect anticipatory licking behaviour yielding a value of +1 and chance level behaviour of 0. We computed a similar metric for anticipatory slowing behaviour as follows, using the same zones but substituting lick rate with time elapsed:

$$\text{slowing}_{\text{ant}} = \frac{\text{slowing}_{\text{near}} - \text{slowing}_{\text{far}}}{\text{slowing}_{\text{near}} + \text{slowing}_{\text{far}}}$$

For the random rearrangement track, we also calculated an accuracy score for each block of 20 trials where rewards were present. This score represents the fraction of trials out of all the trials within the block that the animal successfully triggered reward delivery before an automatic reward was delivered. The animal could trigger a reward by licking in the first half of the reward zone (first 10 cm within the reward zone). If an animal failed to lick within the first half of the reward zone, an automatic reward was delivered at 10 cm into the zone, but the trial counted as an incorrect trial.

## Position tracking in freely moving recordings

Position and head direction were calculated from tracking green and red LEDs on the headstage holder. First, for each arena, we manually labelled 20 frames each from 4 videos and trained a DeepLabCut (v2.2.0.6)[58,59] ResNet-50-based neural network for up to 100,000 iterations, as test error plateaued after this. We selected the model with the lowest test error, which was always less than 2 pixels. Models with high numbers of outliers (training videos had more than 100 frames with labels that jumped more than 2.5 cm) had outlier frames manually labelled and were retrained with these additional frames. These models were then used to analyse all videos collected from the same arena. Second, labels were used if they met all the following criteria: more than 90% likelihood, inside arena boundaries and distance between temporally adjacent labels of less than 2.5 cm. Missing frames were interpolated over unless they spanned more than 10 contiguous frames and more than 5 cm, in which case they were left blank. Animal coordinates and head direction were extracted from the final LED labels and smoothed with a Hanning filter length of 15 frames.

## Offline spike sorting

Spike sorting was performed offline using the open source spike sorting algorithm Kilosort2 (ref. 60). Clusters were manually inspected and curated in Phy (https://github.com/cortex-lab/phy). Clusters containing fewer than 500 spikes were discarded. Clusters were also discarded if their peak-to-peak amplitude over noise ratio was less than 3, in which

noise was estimated by calculating the standard deviation of a 10-ms window preceding each spike. All remaining clusters were manually examined and were discarded if they did not appear to be from single well-isolated units. For chronic recordings, neural data streams from the virtual reality and freely moving sessions were concatenated using CatGT (https://github.com/billkarsh/CatGT) before spike sorting.

Virtual reality data were then synchronized to spiking data using the TTL pulse times from Unity and the recorded TTL pulses in SpikeGLX. Custom code written in MATLAB was used to synchronize the data. Confirmation of appropriate syncing was accomplished by noting a high correlation between virtual reality time differences and the corresponding TTL time differences recorded in SpikeGLX. Finally, because the virtual reality frame rate was not constant, we used interpolation to resample behavioural data at a constant 50 Hz, similar to ref. 51. Data were not speed filtered. To calculate spatial firing rates, spikes were first binned into 2-cm spatial bins and divided by time occupancy per spatial bin. Then, the firing rates were smoothed with a 2-bin Gaussian kernel.

## Identification of grid cells in freely moving conditions

Spatial tuning curves in freely moving conditions were computed by binning spikes into 2.5 × 2.5-cm bins. Cells with average firing rates greater than 40 Hz were excluded. To identify putative grid cells, we considered two approaches. First, we computed a 'grid score' for each neuron[61] (Extended Data Fig. 3). Grid score was computed by a circular sample of 2D spatial autocorrelograms, correlated with the same sample rotated 30°, 60°, 90°, 120° and 150°. Grid score was defined as the minimum difference between the 60° and 120°, and the 30°, 90° and 150° rotations[61]. Grid cells were identified as those cells whose grid score exceeded the 99th percentile of a shuffled distribution of grid scores, computed by time-shifting individual spike trains 1,000 times. Using this approach, we observed a fair number of false positives (cells that did not clearly have hexagonally spaced firing fields but were classified as grid cells) and false negatives (cells with clear hexagonal firing fields that were not classified as grid cells). Thus, we used a second approach, which was recently applied ref. 34, clustering the spatial autocorrelograms of the 2D tuning curves. As in ref. 34, we used uniform manifold approximation and projection (UMAP) to project the spatial autocorrelograms of individual neurons into two dimensions (Extended Data Fig. 3), after removing the central peaks from the autocorrelograms and z-scoring them. We matched the UMAP settings in ref. 34. We then clustered the resulting 2D projections. We found that this approach more reliably separated grid cells and non-grid cells. However, we found it less successful than reported in ref. 34 at separating distinct modules, perhaps because we had fewer cells recorded from each module, or because our 2D tuning curves were slightly less clean than those recorded from rats in ref. 34.

## Identification of non-grid spatial cells in head-fixed virtual reality conditions

We identified non-grid spatial cells as those cells not identified as grid cells whose tuning curves in the first half of a virtual reality block had a Pearson correlation of more than 0.2 with their tuning curves in the second half of the virtual reality block, averaged across virtual reality blocks.

## Identification of grid cells in head-fixed complete darkness conditions

We identified putative grid cells by computing spectrograms for the spatial firing rate of each neuron in the dark. We first z-scored the spatial firing rates of each cell. Spectrograms were computed using scipy.signal.spectrogram with nperseg = 1,600 and noverlap = 1,400, corresponding to segments of width 32 m (or 10 laps on the build-up track) with measurements every 4 m (example single-cell spectrograms in Extended Data Fig. 4). The segment width and overlap were chosen to balance resolution in the frequency domain (to give good accuracy in inferring peak locations) with resolution in the time domain (as the

location of the three peaks drifts over many trials in the dark, consistent with previous work showing grid cells accumulate error and drift in complete darkness)[16,32,62] (Fig. 1h and Extended Data Figs. 2 and 4). We chose slightly different values for nperseg depending on the number of dark trials in a given experiment; values for all conditions are reported in the data reporting Excel sheet.

We then extracted co-modular grid cells, neurons that shared the same grid spacing and orientation but differed in phase, by applying k-means clustering to the spectrograms (using k = 5 or 6). We identified a cluster as a putative grid cell module if the average spectrogram correlation (Pearson r) was greater than 0.3 across windows in the first block of trials (n = 15,342 out of 68,484 neurons). We found that these correlations were preserved over much longer timescales, from the first block of dark trials at the beginning of the session to the final block of dark trials at the end of the session (Extended Data Fig. 2).

## Sorting grid cells onto a neural sheet

The hexagonal spatial map of an idealized grid cell can be written as the sum of three plane waves oriented at 0°, 60° and 120°,

$$r_i(\vec{x}) = \sum_{a=1}^{3} \cos\left(\vec{k}_a \cdot \vec{x} - \phi_a^i\right)$$

Where $r_i(\vec{x})$ is the activity of neuron $i$ at 2D location $\vec{x}$ and $\vec{k}_a$, $a = 1, 2, 3$, are wave vectors oriented at 0°, 60° and 120°, which set the overall grid scale of the module. Hence, each grid cell in a module is uniquely determined by its phases $\phi_a^i$. Our goal was to infer these phases from recordings while the animal is running on the 1D virtual reality track. In 1D, the position of the animal is restricted to lie along a line at some angle $\gamma$, $\vec{x} = (x\cos(\gamma), x\sin(\gamma))$ and hence the 2D wave vectors are transformed to 1D scalar frequencies $f_a = \vec{k}_a \cdot (\cos(\gamma), \sin(\gamma))$[36],

$$r_i(x) = \sum_{a=1}^{3} \cos(f_a x - \phi_a^i)$$

These three frequencies $f_a$ are the peaks that we observed in the PSDs in Fig. 1. To estimate $f_a$, we used a standard peak-finding algorithm (scipy.signal.find_peaks) to identify the largest three peaks in the PSD within each window, $\hat{f}_a$. For each neuron $i$ of $N$ neurons, we then extracted the phases $\phi_a^i$, $a = 1, 2, 3$, of the Fourier transform at the location of each peak (Fig. 1b):

$$\phi_a^i = \arg\left[\int_{x_a}^{x_b} dx \, e^{-i2\pi x \hat{f}_a} r_i(x)\right]$$

Where $x_a$ and $x_b$ represent the start and end of a window, with the same size as above (32 m). We found that estimated phases were highly consistent across independent windows in the dark (Extended Data Fig. 2e). To obtain refined estimates, we averaged estimated phases across windows in the dark. We used these estimated phases to extract the instantaneous attractor state of the grid cell population activity, and to sort grid cells onto a fictitious neural sheet (Fig. 1e) so that nearby cells have similar phases (described below).

To extract the moment-by-moment attractor state $\vec{\theta}(x)$ of the grid cell population activity, we estimated the centre of mass of the bump of activity on the sheet as follows. We computed the average of the phases of the cells on the neural sheet weighted by their (filtered) firing rates $r_i(\vec{x})$. Because the phases are periodic, we computed a circular average via the following formula,

$$\psi_a(x) = \text{atan2}\left(\sum_{i=1}^{N} \sin(\phi_a^i) r_i(x), \sum_{i=1}^{N} \cos(\phi_a^i) r_i(x)\right)$$

Where $\psi_a(x)$, $a = 1, 2, 3$ represent the firing-rate-weighted average of each phase. If the population activity of grid cells is well described by

the motions of a bump on a 2D sheet, then the three coordinates $\psi_a(x)$ represent only 2 d.f. In Extended Data Fig. 2d, we showed that $\psi_a(x)$ does not take on all possible values, but are indeed restricted to a 2D subspace. We can therefore extract these 2 d.f., $\theta_1$, $\theta_2$, which represent the centre of mass of the bump of activity on the sheet,

$$\theta_1 = \psi_1 - \frac{\psi_2}{2}, \quad \theta_2 = \frac{\sqrt{3}}{2}\psi_2$$

This transformation captures the hexagonal geometry of the grid cell lattice[63], and converts the firing-rate-weighted average of each phase $\psi_a(x)$ to 2D coordinates $\theta_1$, $\theta_2$ on a unit cell of the neural sheet, as illustrated in Fig. 1e. We can then visualize the trajectory of the bump of activity on the sheet by plotting the two coordinates $\theta_1$ and $\theta_2$ as the animal traverses the virtual reality environment. We plot $\theta_1$ and $\theta_2$ for one full session as a heatmap over spatial bins and trials in Fig. 2c. As the neural sheet is periodic, we also occasionally unwrapped $\theta_1$, $\theta_2$ to plot them as continuous trajectories across multiple copies of the neural sheet (Figs. 1g and 2d,i). Unwrapping was performed using np.unwrap on $\psi_a(x)$ before converting to $\theta_1$, $\theta_2$. Note that this unwrapping is used only for visualization and not used in data analyses. Unwrapping is sensitive to noise and often leads to unwrapping errors in which the unwrapped trajectory becomes dislocated by one neural sheet length. To mitigate this issue, in Extended Data Fig. 5, we adopted the following approach. Owing to the discontinuous nature of the unwrapping procedure, it is sensitive to noise only at the edge of the neural sheet. However, the location of the edge of the neural sheet is arbitrary (we can experimentally measure only the relative phases between grid cells; the location of the origin is arbitrarily chosen). Moreover, different choices of the origin will lead to more or fewer unwrapping errors. Taking advantage of this freedom, we performed a grid search over 100 different choices of the origin for each window of 5 trials by applying a global shift to all the phases of a cell $\phi_a^i \to \phi_a^i + \eta_a \bmod 2\pi, \eta_a \in [0, 2\pi]$, and kept the choice of origin, which leads to the most consistent trajectories over the 5 trial window (the underlying assumption is that trajectories do not drift much over windows of 5 trials, and that unwrapping errors will produce inconsistent trajectories). This procedure is somewhat cumbersome but leads to cleaner visualizations of the unwrapped bump trajectory for noisy trajectories.

The above procedure for extracting the moment-by-moment centre of mass of the bump on the neural sheet has a geometrically equivalent interpretation as identifying the instantaneous location of the high-dimensional population activity vector on a toroidal attractor. To see this, note that computing the arguments of the circular average above,

$$\psi_a(x) = \mathrm{atan2}\left(\sum_{i=1}^{N} \sin(\phi_a^i) r_i(x), \; \sum_{i=1}^{N} \cos(\phi_a^i) r_i(x)\right)$$

involves amounts to projecting the $N$-dimensional neural activity vector $r_i(x)$ onto three pairs of axis spaces spanned by the vectors $u_a^i = \cos(\phi_a^i)$ and $v_a^i = \sin(\phi_a^i)$. Now recall that the firing rate of an idealized grid cell in 1D can be written as,

$$r_i(x) = \sum_{a=1}^{3} \cos(f_a x - \phi_a^i)$$

This can be rearrange to find,

$$r_i(x) = \sum_{a=1}^{3} \cos(\phi_a^i)\cos(f_a x) + \sin(\phi_a^i)\sin(f_a x)$$
$$= \sum_{a=1}^{3} u_a^i \cos(f_a x) + v_a^i \sin(f_a x)$$

Hence, the idealized grid cell population activity lives in a subspace spanned by the same three pairs of axes as those that we projected onto to compute the centre of mass of the bump on the neural sheet. Within each pair of axes, the activity lies along a ring, and in the full subspace, lies along a torus (Extended Data Fig. 2a). Therefore, if we have reliably inferred the phases of our recorded grid cells, and they are well approximated by the idealized grid cells above, then projecting their population activity onto these three pairs of axes should reveal three rings. Indeed this is what we found for our recordings on the 1D virtual reality track (Extended Data Fig. 2b,c), indicating that the population activity lies close to a toroidal attractor. Note that estimating the firing-rate-weighted average of each phase on the neural sheet $\psi_a(x)$ is mathematically equivalent to extracting the angle on each of the three rings. Hence, the moment-by-moment centre of mass of the bump of activity on the neural sheet is in one-to-one correspondence with the instantaneous location of the high-dimensional population activity on the toroidal attractor.

To visualize the population activity of co-modular grid cells as a pattern of activity across the fictitious neural sheet, we identified the estimated phases $\phi_{1,2,3}^i$ with locations on the 2D neural sheet $s_1^i$, $s_2^i$ using the same transformation used to compute $\theta_1$ and $\theta_2$ above,

$$s_1^i = \phi_1^i - \frac{\phi_2^i}{2}, \quad s_2^i = \frac{\sqrt{3}}{2}\phi_2^i$$

This transformation is schematized in Fig. 1e. Because this transformation requires only 2 d.f., we selected the two phases that were best correlated across windows in the first block of dark trials. To visualize population activity at an instant in time in this sort (Fig. 1e,f), we binned the two sheet coordinates $s_1^i$, $s_2^i$ into $\sqrt{N} \times \sqrt{N}$ quantiles and plotted (z-scored and Gaussian filtered) activity as a heatmap on this grid. Because this analysis requires good coverage of the 2D phase space, we restricted this analysis to sessions in which 36 or more cleanly recorded co-modular grid cells were recorded simultaneously within a module.

### Remapping between virtual reality blocks

To quantify remapping between blocks in the build-up track environment, we computed the spatial correlation of grid cell population activity $r_i^\alpha$, $r_i^\beta$, for all pairs of trials $\alpha$ and $\beta$. Averaged across modules, sessions and animals, the resulting correlation matrix (Fig. 2f, bottom) revealed that population activity was stable within a block, but differed between consecutive blocks. The sharp boundaries indicated that remapping happened quickly. To quantify the rate of remapping, we additionally computed the average correlation within a sliding window of five trials (Fig. 2f, top),

$$\overline{\rho}_t = \frac{1}{10} \sum_{\alpha > \beta}^{5} \rho(t + \alpha, \; t + \beta)$$

Where $\rho(\alpha, \beta)$ represents the spatial correlation between trials $a$ and $b$. Finally, to visually inspect the differences in tuning curves between consecutive environments at a single-cell level, and within a single session, we compared single-cell spatial tuning curves averaged over all 40 trials within the tower 2 environment (block 6) and the 40 trials within the tower 3 environment (block 7; Fig. 2h).

### Distortions to grid cell tuning curves in the presence of landmarks

We quantified the anisometry of neural trajectories on the attractor by computing the arc length $s$ of the trajectory of the centre of mass of the activity bump on the neural sheet over intervals corresponding to 2 cm in real space, $s(x) = \sqrt{d\theta_1(x)^2 + d\theta_2(x)^2}$, where $d\theta_1(x) = \theta_1(x + dx) - \theta_1(x)$, $d\theta_2(x) = \theta_2(x + dx) - \theta_2(x)$, and $dx = 2$ cm. To handle periodic boundary conditions, we first unwrapped the

coordinates $\theta_1$ and $\theta_2$ before computing $s$. Intuitively, $s$ quantified the distance the activity bump travelled on the neural sheet when the animal traversed 2 cm on the virtual reality track. If the mapping from the position of the animal to the position of the bump on the neural sheet is an isometry (that is, distances in real space are proportional to distances on the neural sheet), then $s(x)$ should not vary from one position on the track to the next. Hence, to quantify the anisometry of the grid cell map, we measured the variation in $s(x)$. To obtain a dimensionless measure of anisometry, we calculated the coefficient of variation in $s(x)$ over windows of length of 16 m (Fig. 3c).

The second quantity that we used to measure distortions to the grid cell spatial map in the presence of visual landmarks is geodesic curvature (Fig. 3d). Intuitively, as the animal travels in a straight line on the treadmill through the virtual reality environment, if the grid cell spatial map faithfully captured the trajectory of the animal, the bump of activity should travel in a straight line trajectory on the neural sheet, or, equivalently, the neural activity should trace out a geodesic trajectory on the torus (a geodesic being the generalization of a straight line on a curved manifold). Parameterizing the 2D torus by the coordinates $\theta_1$ and $\theta_2$, the geodesic curvature $\kappa_g$ reduces to the planar curvature:

$$\kappa_g = \frac{\theta_1' \theta_2'' - \theta_1' \theta_2''}{\left(\theta_1'^2 + \theta_2'^2\right)^{3/2}}$$

To handle periodic boundary conditions, we first unwrapped the coordinates $\theta_1$ and $\theta_2$ before computing $\kappa_g$. To extract a single scalar $D$ measuring the curvature of a single-trial trajectory, we integrated the geodesic curvature along the trajectory as the animal completes one lap down the virtual reality track:

$$D = \int_C |\kappa_g| dx$$

Where $x$ represents the position along the track, and $C$ represents a single lap down the track. We computed the single-trial geodesic curvatures for trials in the dark and trials in virtual reality, and found that trials in virtual reality have significantly higher geodesic curvature (Fig. 3d).

**Pinning strength.** To quantify the local effect of a virtual reality landmark on the bump trajectory in the random environments experiment, we computed the trial-by-trial position of the bump of activity, conditioned on the animal being at the location of one of the virtual reality landmarks $\theta_l^t \equiv \theta(x_l^t)$, where $x_l^t$ represents the position of landmark $l$ on the virtual reality track on trial $t$. $\theta_l^t$ was plotted across trials in one session in Fig. 3e for all five virtual reality landmarks. We then considered the extent to which landmarks were 'strongly pinning', in the sense that the bump of activity was pinned to the same location on the neural sheet each time the animal passed a given landmark (Fig. 3e). To quantify this, for each landmark, we measured the dispersion of $\theta_l^t$ around its mean $\overline{\theta_l}$ across trials:

$$\sigma_l^2 = \frac{1}{T} \sum_t^T ||\theta_l^t - \overline{\theta_l}||_P$$

## 2D dynamical model

We modelled the activity of a population of grid cells by a bump moving on a periodic sheet, with the topology of a torus, which is twisted such that grid cells exhibit hexagonal rather than square firing patterns[63]. In the absence of landmarks, the bump moves in straight lines with a velocity $\vec{v}$. We simulated the effect of visual landmark $i$ on the grid cell code as 'pinning' the bump of activity to a preferred location $\vec{\rho_i}$,

$$\frac{d}{dt}\vec{\theta} = \vec{v} + \sum_{i=1}^P \alpha_i d(\vec{\rho_i}, \vec{\theta}) F(p_i - x)$$

Where $x$ is the position of the animal on the track, $p_i$ is the location of landmark $i$ on the virtual reality track, $\vec{\rho_i}$ is the preferred 2D pinning location of that landmark on the grid cell sheet and $\alpha_i$ represents the pinning strength of the landmark. $d(\cdot,\cdot)$ represents the (signed) displacement between two positions on the neural sheet, taking into account the hexagonal periodic boundary conditions as in ref. 63. $F$ parameterizes the range of influence of landmarks on the grid cell code. We found that the precise functional form of $F$ did not make a crucial difference, so we chose a particularly simple form: $F(p_i - x) = \Theta(25 - |p_i - 25 - x|)$, where $\Theta$ is the Heaviside function, so that each landmark only influenced the grid cell code in the 50 cm leading up to its location on the virtual reality track $p_i$.

Finally, we modelled the velocity as a straight-line trajectory with angle $\varphi$, $\vec{v}(t) = |v|(\cos(\varphi), \sin(\varphi))t$, where the slice angle drifts with time at a rate $\delta$. The magnitude (or gain) $|v|$ governs the grid scale.

We estimated $|v|$ and $\delta$ using the responses of the grid cells in the dark, as described above. To fit $\rho_i$ and $\alpha_i$, we held out one block of the random environments experiment at a time, and used the remaining 11 blocks to estimate $\rho_i$, by calculating the average location of the bump of activity when the animal was at the location of each landmark on the virtual reality track across all 11 blocks, and $\alpha_i$, by the average stability of the grid cell code around the landmark (Extended Data Fig. 5b).

Using these estimated values, we simulated the dynamical model above and obtained estimated firing rates $\hat{r}_i(x)$ for each grid cell. We evaluated the performance of the model by computing the Pearson correlation between the estimated firing rates $\hat{r}_i(x)$ and the true firing rates $r_i(x)$. We compared with two shuffled conditions. In the first, we repeatedly shifted the predicted firing rates of each cell by a random amount, $\hat{r}_i((x + \delta) \bmod L)$, $\delta \sim Uniform(0, L)$, where $L$ is the length of the track and mod $L$ accounts for the periodic nature of the track (Fig. 3f). This shuffle is stronger than simply shuffling spikes, as it preserves the spatial structure of the predicted tuning curves and shifts only the locations of their peaks. In the second, we fed a randomly shuffled set of landmark locations to the model (Extended Data Fig. 8e). This shuffle was designed to test whether the model uses information about the locations of the landmarks in the heldout environment to predict the tuning curves of the grid cells, or simply reproduces their average firing properties.

Finally, we found that the performance of the model tended to be higher on blocks of trials where the grid cell code was stable. To illustrate this effect, we plotted a 2D histogram of grid cell code stability (defined as the spatial correlation between the average population activity in the first half of the block and the second half of the block) and model performance (Fig. 3g). To visualize the rate maps of simulated cells (Fig. 3i,j), we drew spikes from a Poisson process based on the underlying firing rate $r_i(x)$ within each spatial bin.

## Position decoding analysis

To determine whether the grid cell code provided a consistent map across all three shifted environments in the hidden reward task, we trained a linear decoder to predict the location of the animal on the track from grid cell population activity. Because single grid cell modules are periodic on length scales shorter than the length of the track, we trained the decoder on the combined population activity of all simultaneously recorded grid cell modules, $r_i(x)$, $i = 1, ..., N_g$, where $N_g$ represents the total number of grid cells in a recording, so that the position of the animal could be uniquely decoded. Owing to the periodic nature of the virtual reality track, we used circular–linear regression, and trained the decoder to predict two coordinates: $\cos(2\pi x/L)$, $\sin(2\pi x/L)$ using two sets of regression coefficients, $\beta^1$ and $\beta^2$, and a least-squares objective. The predicted position of the animal was then given by

$$\hat{x} = \arctan2\left(\sum_{i=1}^{N_g} \beta_i^1 r_i(x), \sum_{i=1}^{N_g} \beta_i^2 r_i(x)\right)$$

The decoder was trained on a random set of one-half of the trials across all three blocks in the tower-shift environment, and evaluated on the held-out half. Cross-validated predictions, averaged across sessions and animals, are shown in Fig. 4b, right.

### Linking neural activity and behaviour

For the tower-shift task, we defined the trial-by-trial stability of the grid cell code as the average spatial correlation of the grid cell population activity on a given trial with all other trials in the block, in the 50 cm preceding the hidden reward zone. Licking error was defined as the average distance between the non-consummatory licks by the animal on that trial and the start of the hidden reward zone.

### Plastic downstream decoder

To model a flexible downstream decoder that could consistently predict the hidden reward location across all three environments, we simulated a biologically plausible learning rule. The decoder was initialized with a random set of afferent weights $w_i$, $i = 1, \dots, N_g$ from all grid cells. Each time the animal licked and received a reward, the weights were updated in the direction of the grid cells active at the moment the animal licked.

$$w_i^{t+1} = w_i^t + \eta \ r_i(x)l(x)$$

Where $l(x)$ is an integer-valued function representing the number of consummatory licks in the spatial bin indexed by $x$. We set $\eta = 1$ in our simulations. We found that the outputs

$$y = f\left(\sum_{i=1}^{N_g} w_i(x)r_i(x)\right)$$

where $f$ is a sigmoidal non-linearity, rapidly adapted to deformations in the grid cell code to consistently predict the hidden reward location across tower-shift environments (Fig. 4n). We simulated the effect of a plasticity knockout by allowing the decoder to adapt normally in the first environment, but setting $\eta = 0$ in both subsequent environments, preventing the decoder from adapting to the distortions in the grid cell code induced by the tower shift (Fig. 4o). The plasticity-knockout decoder predicted that the animal would lick late in both subsequent environments.

### Reporting summary

Further information on research design is available in the Nature Portfolio Reporting Summary linked to this article.

### Data availability

A modified Allen Common Coordinate Framework 'template' volume and 'annotation' volume were used to register histological sections to a reference atlas: https://figshare.com/articles/dataset/Modified_Allen_CCF_2017_for_cortex-lab_allenCCF/25365829. Data collected for this publication are available in two parts on Mendeley Data at https://doi.org/10.17632/rgtk6jygjc.1 and https://doi.org/10.17632/2n4t9bw3xz.1. Source data are provided with this paper.

### Code availability

All custom code written for this article is available at https://github.com/GiocomoLab/mec-rapid-learning/tree/main. Kilosort2 was used for spike sorting offline: https://github.com/MouseLand/Kilosort. Phy v1 was used to manually inspect and curate clusters after spike sorting: https://github.com/cortex-lab/phy. SHARP-Track was used to reference histological slices to the Allen Brain Atlas and to infer probe locations from dyes: https://github.com/cortex-lab/allenCCF. Deep-LabCut v2.2.0.6 was used to identify the position and head direction of an animal from green and red LEDs on a headstage holder: https://github.com/DeepLabCut/DeepLabCut.

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

**Acknowledgements** We thank A. Diaz for cryosectioning and histology assistance as well as animal care; S. Ocko, G. Mel and J. Whittington for useful discussions on modelling approaches; and S. Shi for feedback on the manuscript. L.M.G. is a Howard Hughes Medical Institute Investigator. Funding was provided by the US National Institutes of Health grants 1R01MH126904-01A1 and R01MH130452 (to L.M.G.), BRAIN Initiative U19NS118284 (to L.M.G.) and K99NS134734 (to E.A.A.J.), The Vallee Foundation (to L.M.G.), The James S. McDonnell Foundation (to L.M.G. and S.G.), The Simons Foundation 542987SPI (to L.M.G. and S.G.), NSF CAREER and Schmidt Foundation (to S.G.), the Stanford Graduate Fellowship (to B.S.) and the Stanford Interdisciplinary Graduate Fellowship (to J.H.W.).

**Author contributions** J.H.W., B.S., S.G. and L.M.G. conceptualized the study. J.H.W., B.S., S.G. and L.M.G. contributed to the methodology. J.H.W., B.S. and E.A.A.J. performed the investigation. J.H.W. and B.S. performed the visualization. S.G. and L.M.G. acquired funding. S.G. and L.M.G. conducted project administration. S.G. and L.M.G. supervised the project. J.H.W., B.S. and L.M.G. wrote the original draft of the manuscript. J.H.W., B.S., E.A.A.J., S.G. and L.M.G. reviewed and edited the manuscript.

**Competing interests** The authors declare no competing interests.

**Additional information**
**Correspondence and requests for materials** should be addressed to Lisa M. Giocomo.

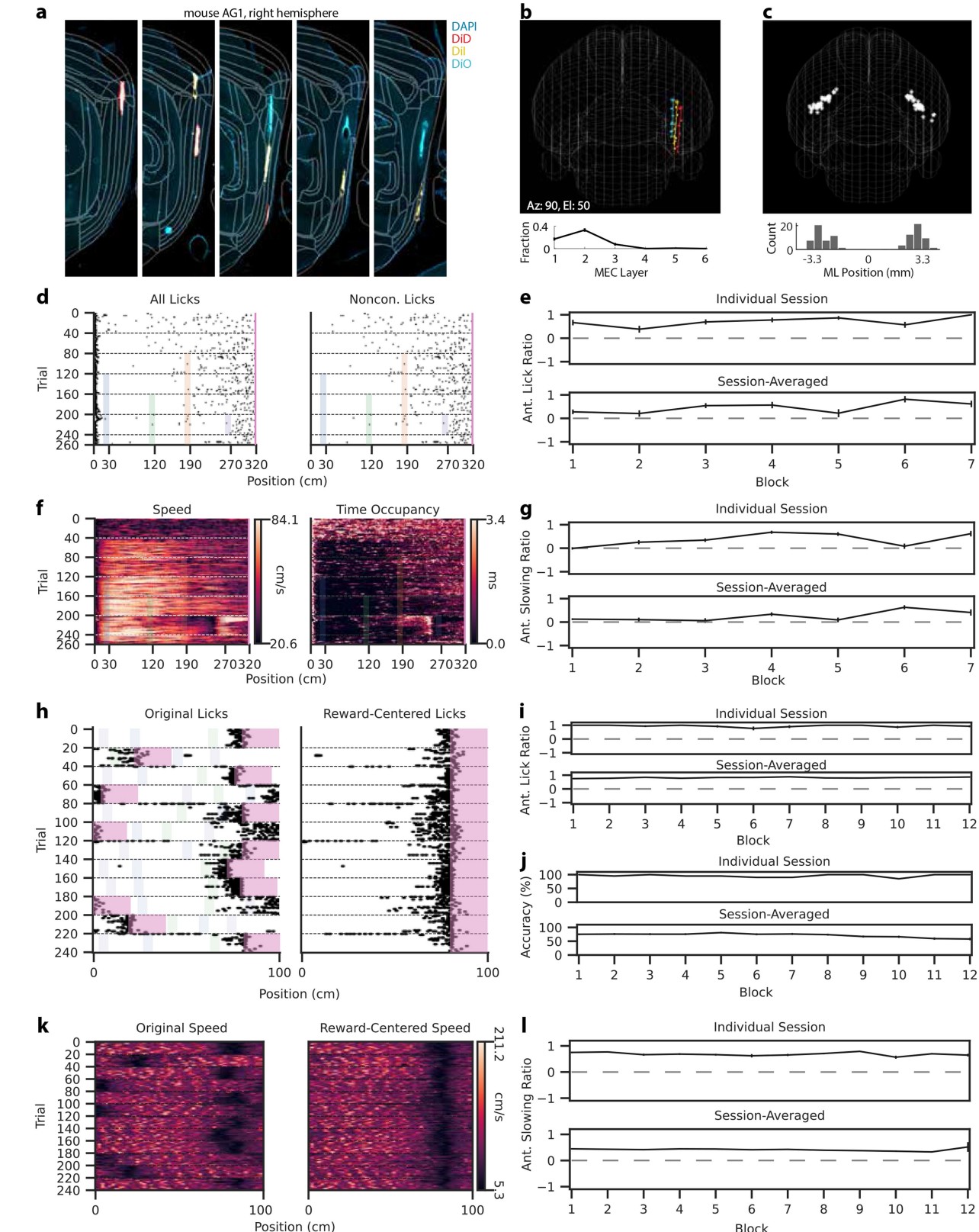

**Extended Data Fig. 1** | See next page for caption.

**Extended Data Fig. 1 | Histological confirmation of MEC targeting and additional animal behavior quantification. a**. Example histology from the right hemisphere of mouse AG1. Gray lines demarcate inferred boundaries between brain regions after sagittal slices were referenced to the Allen Brain Atlas using SHARP-Track[55]. Slices are ordered lateral to medial from left to right. Slices were stained with DAPI, rendered in blue, to label cell nuclei. DiD, DiI, and DiO are rendered red, yellow, and turquoise, respectively. **b**. *Top:* 3D reconstruction of probe locations in (a) overlaid on a mesh wire diagram of the brain. The brain is angled at azimuth 90 degrees, elevation 50 degrees for clarity of visualization. Individual colored points represented selected locations with clear dye. Solid, colored lines represent the best linear fit through the points. Starred points indicate entry points, locations where the probe first entered the brain. *Bottom:* estimate of the fraction of all Neuropixels sites in each MEC layer across all sessions (animal n = 15, session n = 92). **c**. Quantification of entry points across all recordings (animal n = 15, n = 92 sessions). *Top*: entry points for all recordings overlaid on a mesh wire diagram of the brain. Each star represents the entry point for an individual session. *Bottom*: histogram across all entry point medial-lateral positions relative to the midline (at 0 mm). Each bin is 350 μm. **d-l**. Example licking and running speed behavior for the build-up (d-g) and random rearrangement (h-l) tracks. **d**. Example licks raster plot for one session on the build-up track, using only the blocks where rewards were present. *Left:* all licks plotted. *Right:* only non-consummatory licks plotted. Note that because the build-up track is periodic and the reward is at the end of a given trial, consummatory licks occur at the beginning of the subsequent trial. **e**. *Top:* quantification of anticipatory lick ratio (Methods) across blocks for the corresponding session in (d). *Bottom:* anticipatory lick ratio averaged over all sessions (animal n = 4, session n = 19). **f**. *Left, right:* speed and time occupancy heatmaps, respectively, for the corresponding session in (d). **g**. *Top:* quantification of anticipatory slowing ratio across blocks for the corresponding session in (d). *Bottom:* anticipatory slowing ratio averaged over all sessions (animal n = 4, session n = 19). **h**. Example non-consummatory licks raster plot for one session on the random rearrangement track, using only the blocks where rewards were present. *Left:* original lick locations plotted. *Right:* reward-centered licks. **i**. *Top:* quantification of anticipatory lick ratio across blocks for the corresponding session in (h). *Bottom:* anticipatory lick ratio averaged over all sessions (animal n = 11, session n = 38). **j**. *Top:* quantification of licking accuracy (Methods) for the corresponding session in (h). *Bottom:* licking accuracy averaged over all sessions (animal n = 11, session n = 38). **k**. Speed heatmap for the corresponding session in (h). *Left:* original speeds. *Right:* reward-centered speeds **l**. *Top:* quantification of anticipatory slowing ratio for the corresponding session in (h). *Bottom:* anticipatory slowing ratio averaged over all sessions (animal n = 11, session n = 38). For all line plots, lines indicate mean, bars indicate SEM. Where appropriate, gray dashed lines indicate chance-level performance.

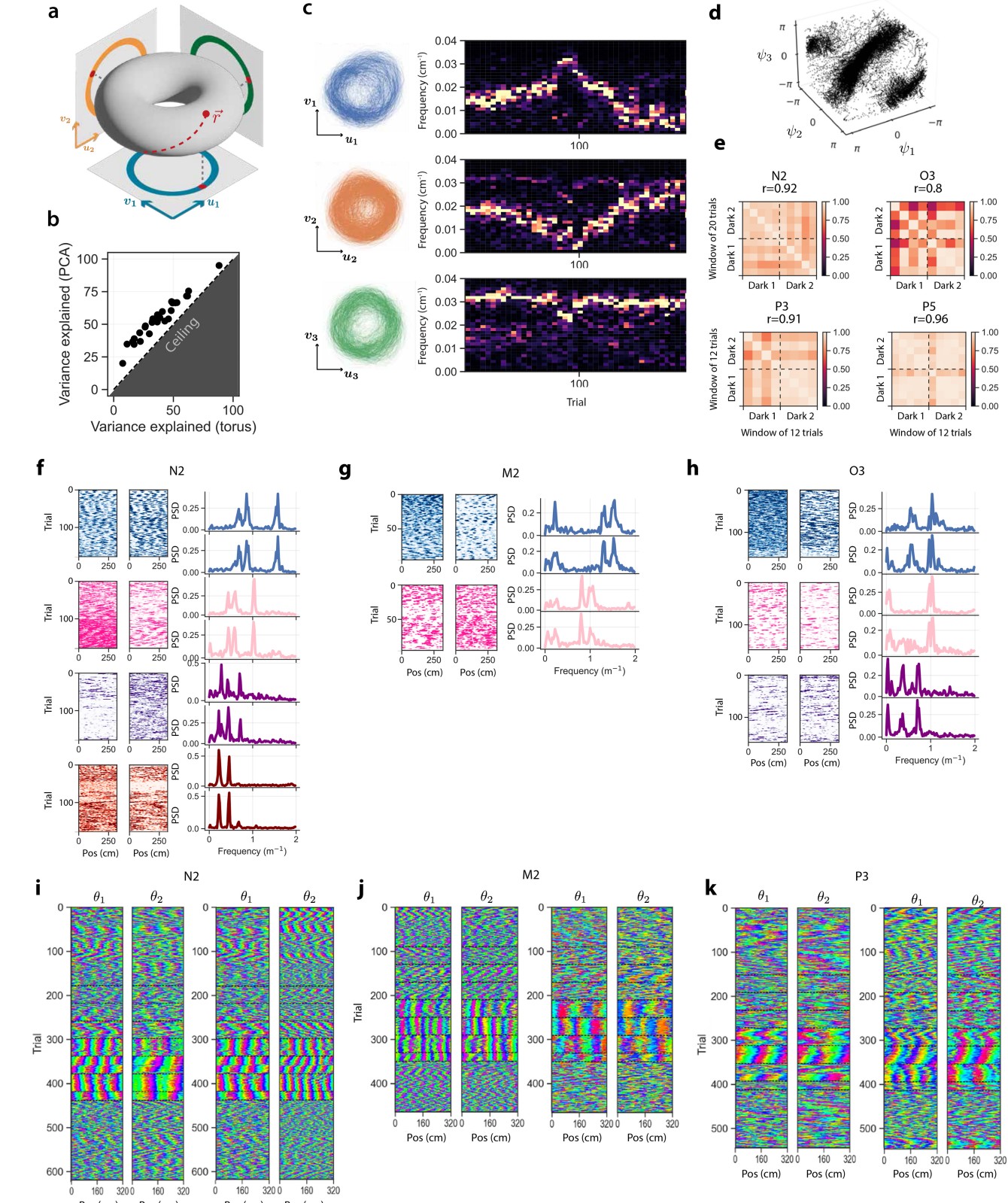

**Extended Data Fig. 2** | See next page for caption.

**Extended Data Fig. 2 | Periodic structure underlying the grid cell code in 1D environments. a**. Neural trajectory (red dashed line) in high-dimensional firing rate space lies close to a torus, identified by projecting neural activity onto the three pairs of axes $u_a = \cos(\phi_a)$ and $v_a = \sin(\phi_a)$. **b**. The 6D subspace in (a) captures nearly as much variance in the grid cell population activity as the top 6 principal components, across modules and sessions (black dots). **c**. Population activity of a set of co-modular grid cells projected onto the three pairs of axes identified in (a). Activity lies close to a ring in each 2D subspace. The three peaks in the PSD (Fig. 1b) are the frequencies at which the neural activity wraps around each of these rings. These frequencies drifted over long distance scales (*right*), color coded for maximum (white) and minimum (black) power. **d**. Scatterplot of angle $\psi_a(x)$ on each of the three rings in (c) as animal (AF3) traverses the dark VR track in a single session (Methods). The three coordinates $\psi_a(x)$ do not take on all possible values in the 3D space, but are instead restricted to lie along a 2D subspace (modulo $2\pi$) indicating that they represent only two underlying degrees of freedom: the 2D coordinates of the bump of activity on the sheet. **e**. Grid cell phase relationships are preserved over minutes and meters of running, over the course of an experiment. Phase relationships

were defined as pairwise distances between 2D phases on the neural sheet $d_{ij} = ||\vec{\phi}_i - \vec{\phi}_j||_P$. We computed the full matrix of phase relationships $d_{ij}^a$ in separate windows $a$ of 10–20 trials in the dark, including both windows at the beginning and end of the experiment. We then correlated the upper triangular part of $d_{ij}^a$ between each pair of windows $a, b$ to obtain a matrix of window-by-window correlations. This matrix of correlations is shown in (b), for four animals and sessions. Each panel is annotated with the average pairwise correlation. **f-h**. Data from three mice shown, with mouse labels above each column. *Left:* Example spike train raster plots for pairs of grid cells from two or more simultaneously recorded modules over 200 laps in the dark. Dots indicate spikes, color coded to match the modules identified on (*right*), sorted by overall grid scale $\lambda = \sqrt{2}/\sqrt{f_1^2 + f_2^2 + f_3^2}$. *Right:* Power spectral densities (PSDs) for each cell in (*left*), computed over the first 10 laps of the session (16 meters), revealed a prominent three-peaked structure within each module. PSDs are consistent for two simultaneously recorded cells within one module, but differ between modules. **i-k**. Bump trajectories, plotted as 2D coordinates on the torus: $\theta_1$ (*left*) and $\theta_2$ (*right*), for two simultaneously recorded modules from the three build-up track sessions in **f-h**.

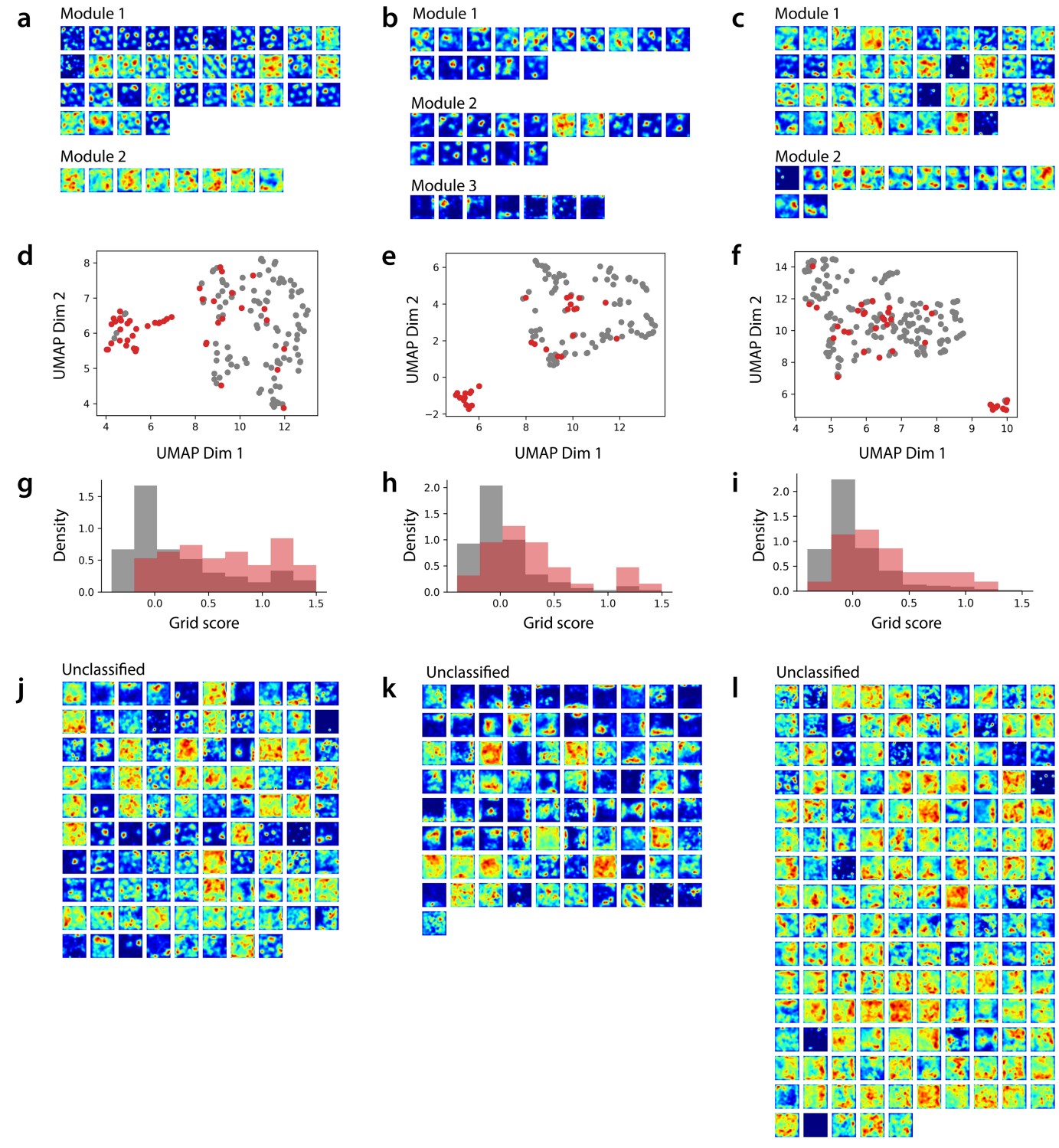

**Extended Data Fig. 3 | Grid cell identification in 2D open-field environments.**
**a-c**. 2D firing rate maps for putative grid cells identified based on their power spectra on the 1D VR track in the dark (Methods), for three different animals. Cells are separated into putative modules. We speculate that the largest modules in (a) (Module 2) and (b) (Module 3) did not show clear grid-like tuning in 2D because their inferred 2D grid scales, 76 cm and 65 cm, respectively, were too large to exhibit clear periodic firing in the 75 cm wide environment. **d-f**. Spatial autocorrelograms for all cells from each of the recordings in (a-c) are collected and projected into 2D via UMAP (Methods), as in refs. 28,34. This projection reveals a cluster of cells with periodic tuning spatial maps, and a cluster without. Putative grid cells identified based on their power spectra on

the 1D track in the absence of visual cues are shown in *red*, and all cells in *gray*. The cluster of cells with periodic tuning, identified via KMeans clustering, had high agreement with grid cells identified based on their power spectra on the 1D track in the absence of visual cues (Methods, average false negative rate: 10.3%, average false positive rate: 15.8%, cell n = 504, session n = 3, animal n = 3). **g-i**. Grid scores distributions for the recordings in (a-c) (*gray)* and for putative grid cells identified based on their power spectra on the 1D track in the absence of visual cues (*red)* (Methods). **j-l**. All cells from the recordings in (a-c) not classified as putative grid cells based on their power spectra on the 1D track in the absence of visual cues.

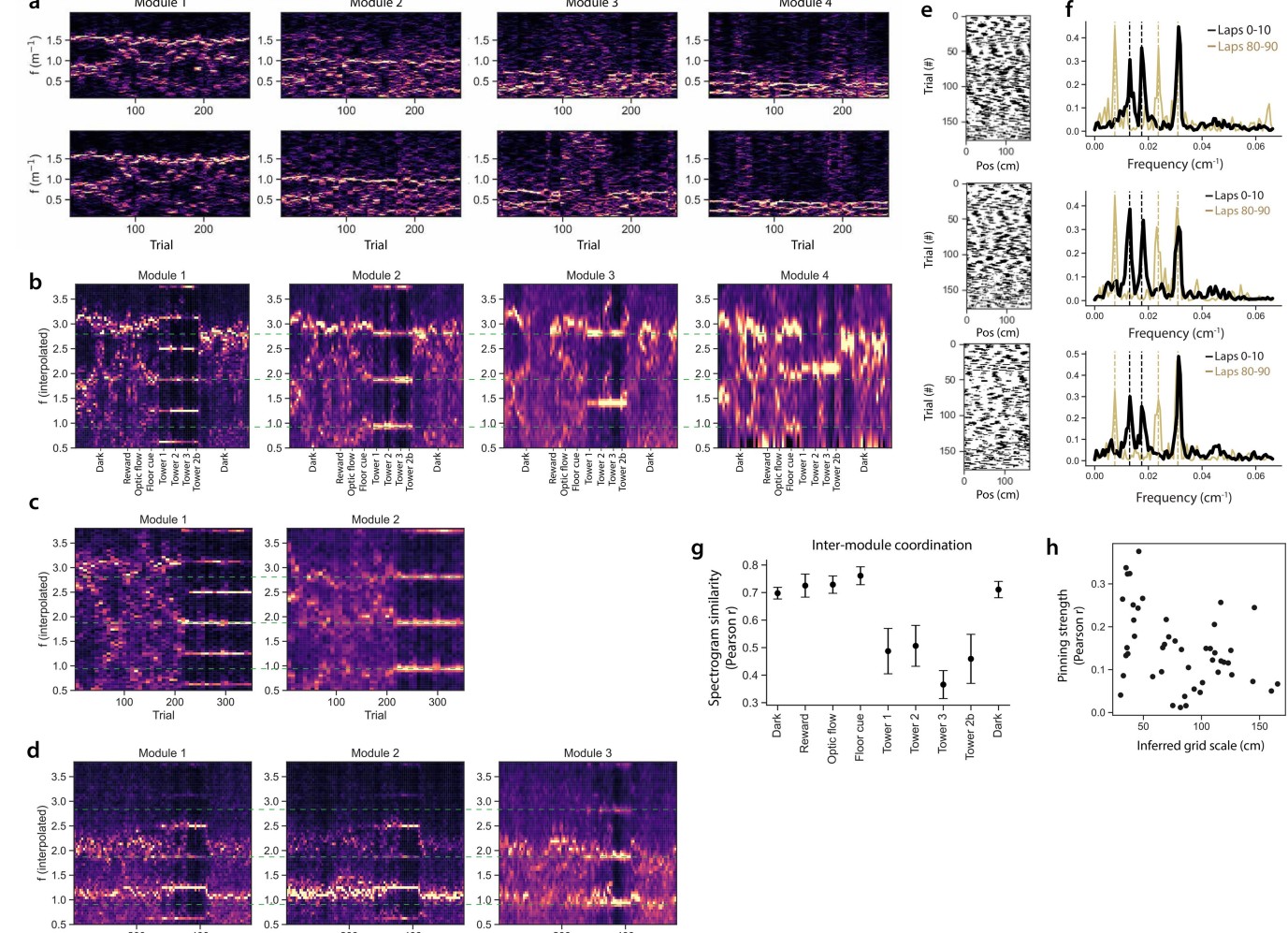

**Extended Data Fig. 4 | Grid cell modules drift coherently in the dark.**
**a**. Single-neuron spectrograms across an entire session, for pairs of
simultaneously recorded grid cells from 4 separate modules. Color (dark low,
white high) represents power in a given frequency $f$ (y-axis). **b-d**. Average
spectrograms of grid cells from simultaneously recorded modules of
increasing spatial scale, from the three sessions in Extended Data Fig. 2a. If
modules rotate coherently in the dark, we would expect different modules'
spectrograms to be identical up to a stretching of the y-axis by a factor of $\sqrt{2}$
(the approximate value of the scale ratio between successive modules). To
investigate this, we scale each successively larger module by a factor of $\sqrt{2}$, and
continuously interpolate the spectrograms so that the y-axes are aligned. The
interpolated spectrograms are shown in (a-c). Green lines indicate the peak
frequencies for Module 2 in the presence of visual landmarks (Tower 1 - Tower
2b conditions). Plots are color coded for minimum (black) and maximum
(white) values. This visualization reveals that the three peaks in the power
spectrum drift coherently between modules in the dark. However, in VR, the
three peaks for each module snap to incommensurate whole number

frequencies. Hence the presence of landmarks disrupts the coordination
between grid cell modules. We quantify this effect in panel (b). **e**. Spiketrain
rasters for three simultaneously recorded grid cells from module 1 in (a) over
180 laps in the dark. **f**. Corresponding PSDs computed over 10-lap windows.
Either laps 0–10 (black), or laps 80–90 (gold). Locations of three estimated
peaks are indicated with vertical dashed lines. **g**. Pearson correlation between
interpolated spectrograms over the course of a recording session. In the dark
and in landmark-poor settings, grid cell modules are tightly coordinated. This
coordination is disrupted with the introduction of visual landmarks. Dots
indicate the mean Pearson correlation within a block, averaged across animals
and sessions. Error bars represent standard error over windows within a block,
averaged across animals and sessions (animal n = 3, session n = 3, module n = 9,
cell n = 264). **h**. Grid scale versus estimated pinning strength in the build-up
track. Pinning strength is measured for a given module as the average stability
of the grid cell code within VR blocks (Pearson r). We observe a small effect
where estimated pinning strength is weaker for larger modules (p = 0.005,
module n = 49, animal n = 4, session n = 14).

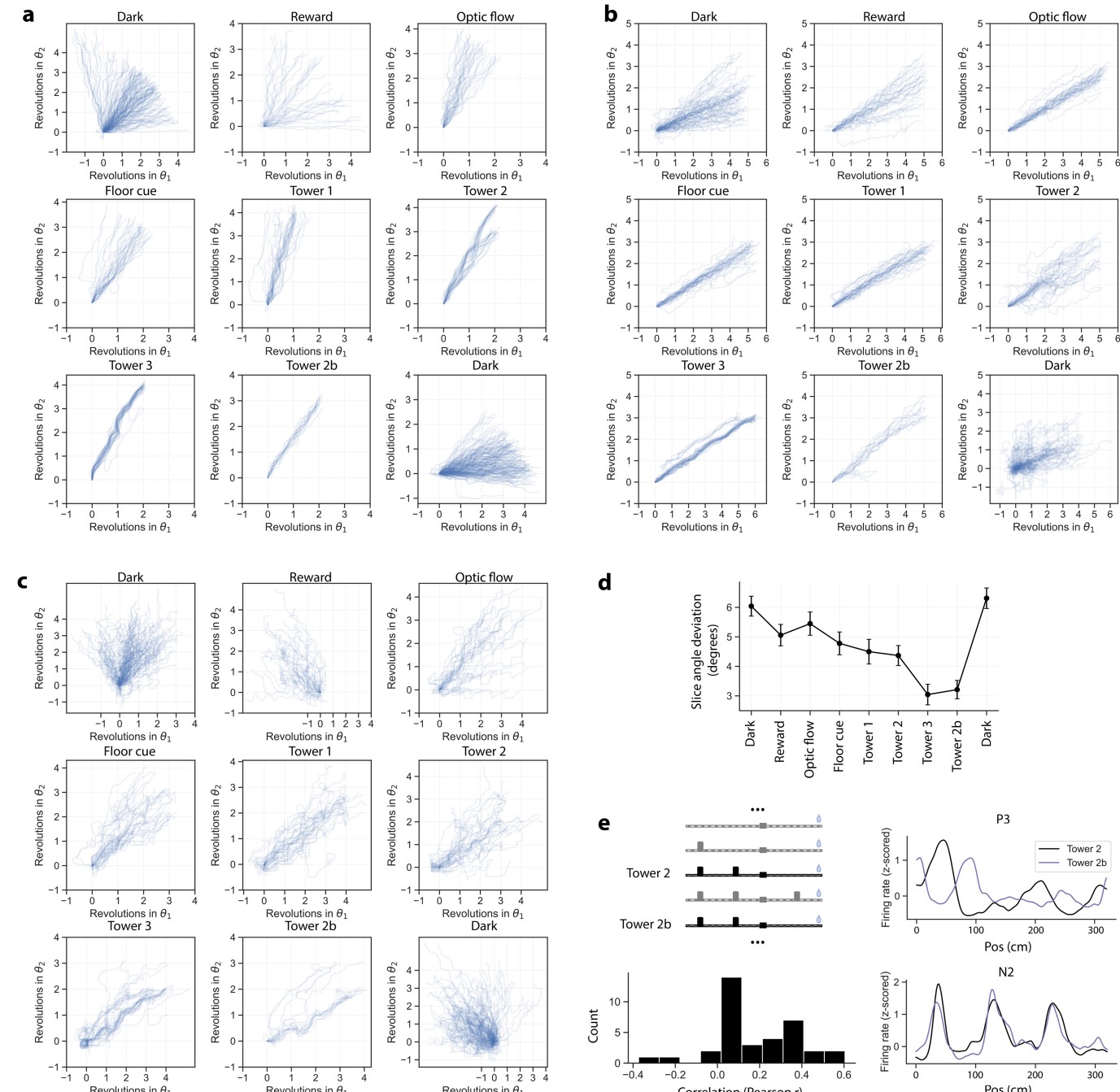

**Extended Data Fig. 5 | Additional activity bump trajectories and remapping statistics on the build-up track. a-c.** Single-trial bump trajectories across all 9 blocks of the build-up track, for three different animals (Methods). Trajectories are unwrapped and plotted across multiple copies of the neural sheet (gray grid, Methods). **d.** Average slice angle deviation across trial blocks (circular standard deviation, session n = 14, animal n = 4, module n = 35, cell n = 747). Dots represent average deviation, and error bars standard error on the mean, across modules, sessions, and animals. **e.** Grid cell maps differ between two identical environments. *Top Left*: Schematic of the build-up track, highlighting the two

visually identical environments (Tower 2 and Tower 2b blocks). *Bottom Left*: Histogram showing the median correlation of grid cell tuning curves in Tower 2 and Tower 2b blocks, across sessions and animals (animal n = 4, session n = 14, module n = 35, cell n = 747). Approximately 50% of sessions have median correlation <0.2 between the two visually identical environments. *Top Right*: Example cell with a low correlation between Tower 2 and Tower 2b environments. *Bottom Right*: Example cell with a high correlation between Tower 2 and Tower 2b environments.

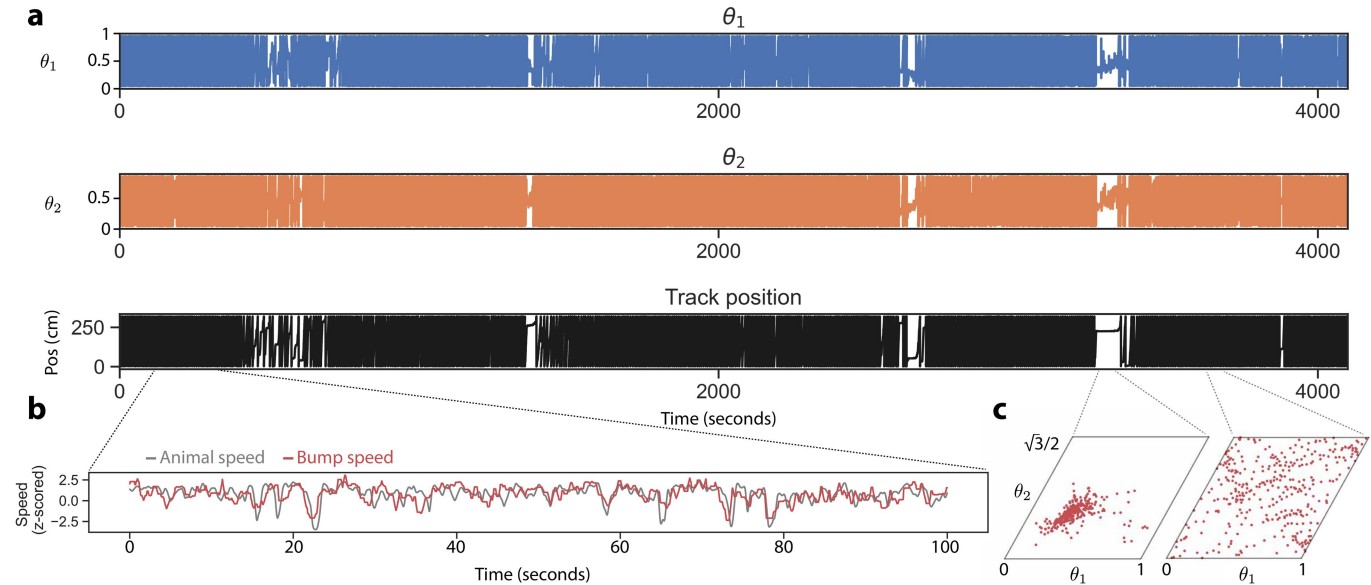

**Extended Data Fig. 6 | Bump trajectory and animal behavior on an example build-up track session (~1 hr). a**. *top, middle*, Inferred 2D bump coordinates on the toroidal attractor, with both phases plotted as a function of time (blue and orange panels) over the course of the entire session. *bottom*, Animal's position on the periodic track throughout the entire duration of the experiment. We observe a few periods of many seconds-minutes where the animal is stationary on the treadmill. During these periods, the inferred position of the activity bump on the attractor is also roughly stationary, though drifts slightly (within one revolution). Note that this drift appears magnified, since one lap of the animal on the track typically corresponds to 3-4 revolutions of the activity bump around the attractor. **b**. Zoomed in view of a period of ~90 seconds where the animal is engaged. We compare the (z-scored) speed of the animal on the treadmill (gray line) to the (z-scored) speed of the activity bump on the attractor (red line). The activity bump's speed closely tracks the animal's speed on the treadmill. **c**. The drift in the inferred position of the activity bump on the attractor during a period when the animal is stationary (*left*), compared to a period of equal duration where the animal is running (*right*). The variability in bump position during the stationary period spans roughly 1/5 of a period, which, for this module of inferred grid scale 25 cm, corresponds to a spatial error of ~5 cm.

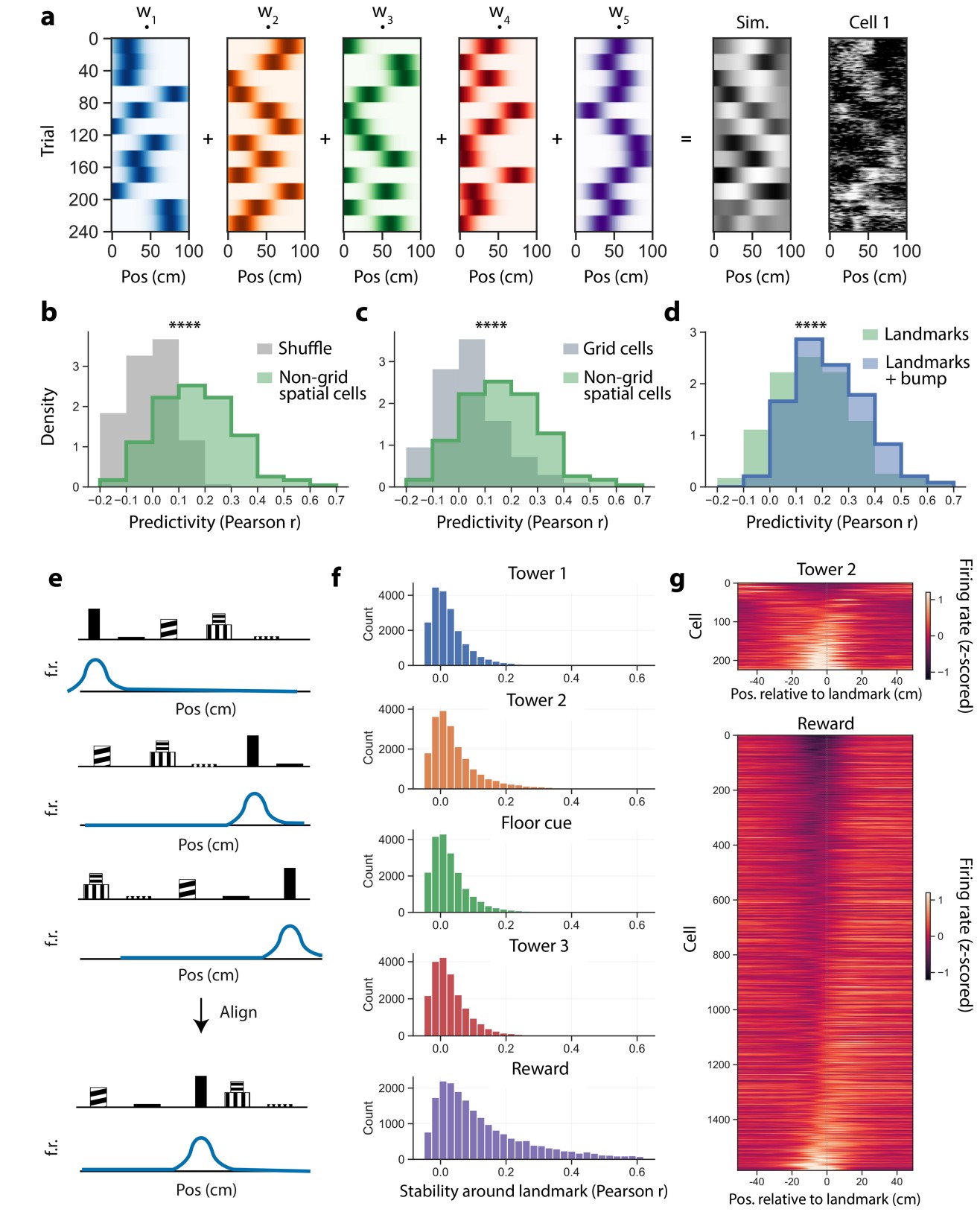

**Extended Data Fig. 7** | See next page for caption.

**Extended Data Fig. 7 | Spatial coding properties of non-grid cells. a**. In the random environment track, we identified non-grid spatial cells as cells that 1) showed no periodic structure in the dark and 2) had an average spatial Pearson correlation > 0.2 between the first and second half of each VR block (Methods). We found that many of these cells were driven by linear combinations of environmental features. To quantify this effect, we defined template maps (colored heatmaps) for each VR landmark as gaussian firing fields with a width of 50 cm, centered at each landmark, similar to the approach taken in ref. 64. We performed cross-validated linear regression with weights $w_1...w_5$, holding out one VR block out at a time, to predict the population activity of non-grid spatial cells in the heldout environment. Example predicted firing patterns across held-out environments for one cell are shown at right (Sim.) and the true firing patterns at far right (Cell 1). **b**. We identified "cue cells" as cells for which the predictivity of the linear regressor exceeded the 95th percentile of a shuffled distribution consisting of training linear regressors on randomly scrambled landmark positions. 56% of non-grid spatial cells met the definition of cue cell, and the distribution of predictivities was much higher than a random shuffle (two-sided KS test, p = 1.25e-34, cell n = 6,332; animal n = 6, session n = 32). **c**. Grid cell firing patterns, in contrast, were not well predicted by linear combinations of sensory stimuli (two-sided KS test, p = 1.65e-30; cell n = 6,332; animal n = 6, session n = 32). **d**. In addition to sensory stimuli, we included the grid cell attractor states (2 coordinates per module) as features in our regression model. We found that this significantly improved predictivity in heldout environments, indicating the possibility that non-grid spatial cells conjunctively code for sensory stimuli and the state of the grid cell attractor (two-sided KS test, p = 1.55e-7, cell n = 6,332; animal n = 6, session n = 32). **e**. To determine the extent to which cells in MEC were driven by individual visual features, we computed tuning curves (blue line) in each block of the random environment track (as illustrated in the top three rows). We then aligned these tuning curves to each of the landmarks, one by one. A cell visually driven by landmark A, when aligned to landmark A, should have a similar tuning curve across environments (as illustrated by the bottom row). **f**. Quantification of the similarity of neural activity for each landmark, analyzed as described in (a). Each graph represents the stability of neurons (n = 20,718) with respect to each landmark. Stability is defined as the average spatial correlation of landmark-aligned maps across all 12 random environment blocks. Note there were very few cells with a high similarity (that is, with similar responses to the same visual landmark across environments). The reward had the strongest "pinning", with the greatest number of cells exhibiting consistent tuning to the reward. Of the visual landmarks, Tower 2 had the largest number of tuned cells. **g**. Heatmap of tuning curves of all cells considered "stable" (Pearson r > 0.4) for Tower 2 (orange) and for the reward (purple), averaged across random environments.

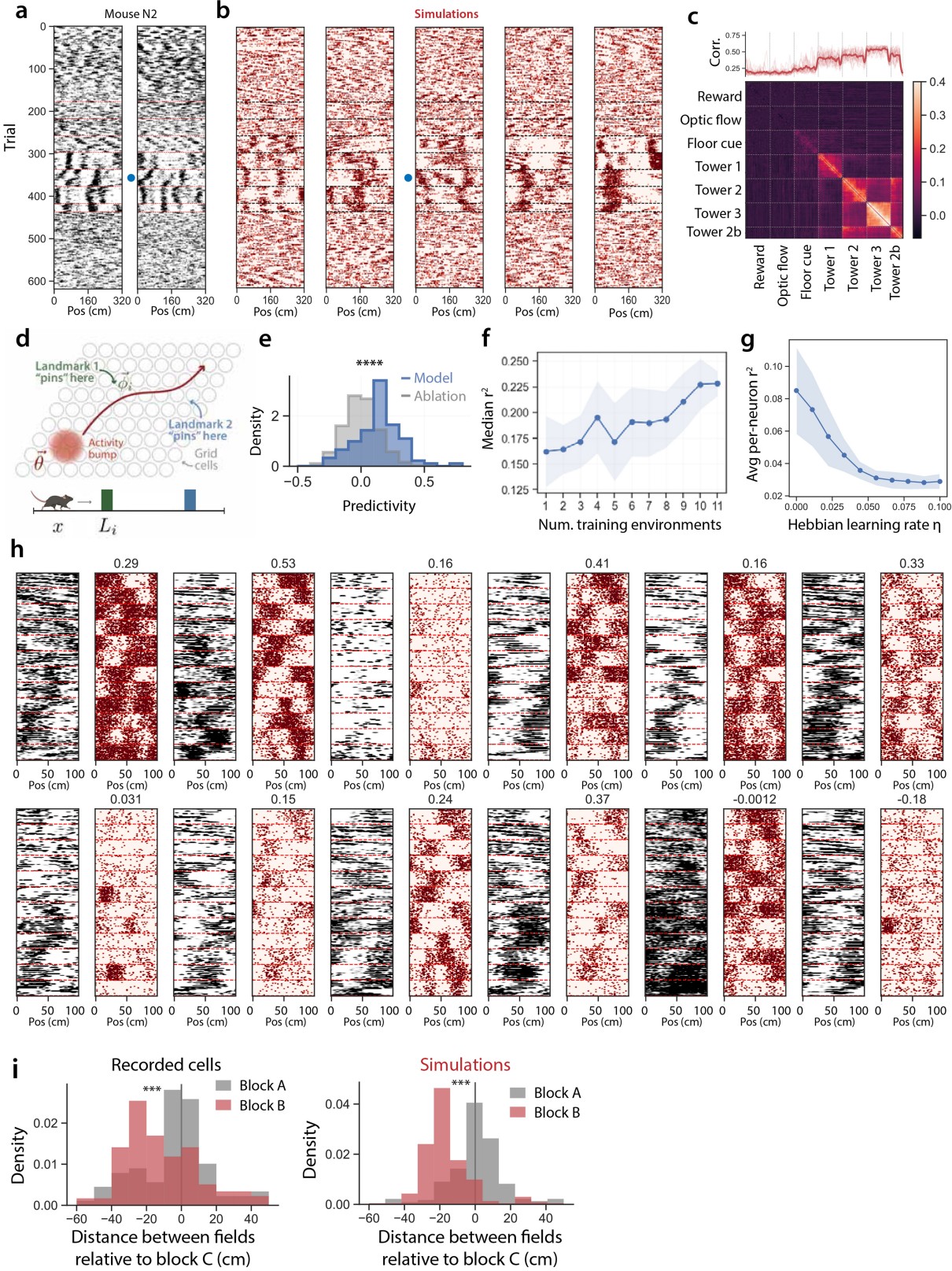

**Extended Data Fig. 8** | See next page for caption.

**Extended Data Fig. 8 | Model details and performance. a-c.** Model simulations in the build-up track (Fig. 2). **a.** A pair of simultaneously recorded grid cells from mouse N2. Black dots correspond to experimentally measured neural spikes and rasters are plotted as in Fig. 2b, with horizontal lines corresponding to different blocks of the build-up track. **b.** Results of 5 independent simulations using random pinning phases for each landmark. Red dots correspond to simulated neural spikes. **c.** Similar to Fig. 2f, simulated grid cell activity rapidly stabilizes within each trial block, but remaps between blocks. **d.** Schematic of 2D dynamical model of grid cell population code. **e.** In addition to comparing to a random shuffle in Fig. 3f, we performed an ablation to confirm that the model uses the spatial locations of the landmarks to predict the grid cells' activity patterns in a novel environment, and does not simply reproduce their average firing properties. To verify this, we fed randomly shuffled landmark locations, rather than the true landmark locations, to the model in each heldout environment. This ablation disrupted the performance of the model to no better than a random shuffle (Fig. 3h), indicating that knowledge of the landmark locations in the novel environment is crucial to predict grid cells' activity patterns (two-sided KS test, p = 3.58e-18, cell n = 91; animal n = 3, session n = 3, module n = 3). **f.** Model performance improves as we increase the number of random environments used to estimate the model parameters ("pinning phases", Methods). Dots represent average across three independent simulations, and shaded areas represent standard error on the mean. **g.** Model fit at different learning rates $\eta$, which defines to what extent the pinning phases $\rho_i$ evolve from trial to trial to minimize their distance to the bump of activity when the animal is near the corresponding landmark. Neural activity is better explained by a fixed, non-plastic circuit ($\eta = 0$) than a plastic circuit. See (Ocko et al.)[15] for details on this Hebbian learning rule. Dots represent average across three independent simulations, and shaded areas represent standard error on the mean. **h.** True grid cell spatial firing rate (black) and simulated grid cell responses for 12 randomly chosen cells within a single session (red). Pearson r between simulated and true tuning curves denoted at the top of each simulated raster. **i.** Quantification of grid cell distortions in the hidden reward task (Fig. 4), for recorded grid cells (*left*) and simulated cells from the model (*right*). Distances between grid cell firing fields became compressed as landmarks were shifted closer together from block A to block B. Because our data includes cells with multiple spatial scales, we normalized the per-cell distances between firing fields in blocks A and B by subtracting the per-cell distances between firing fields in block C (which differs from A only by an overall shift in its arrangement of landmarks). Since blocks A and C have similar distances between firing fields, the gray histogram is centered at zero, while the red histogram is shifted 20–30 cm left because distances between firing fields are compressed in block B. We observe significant distortions in both real grid cells (top, two-sided KS test, p = 5.82e-07, n = 86 blocks; cell n = 457, mouse n = 5, session n = 10), and in cells simulated by the model (two-sided KS test, p = 3.91e-07, n = 150 blocks, simulated cell n = 500, random simulated session n = 20).

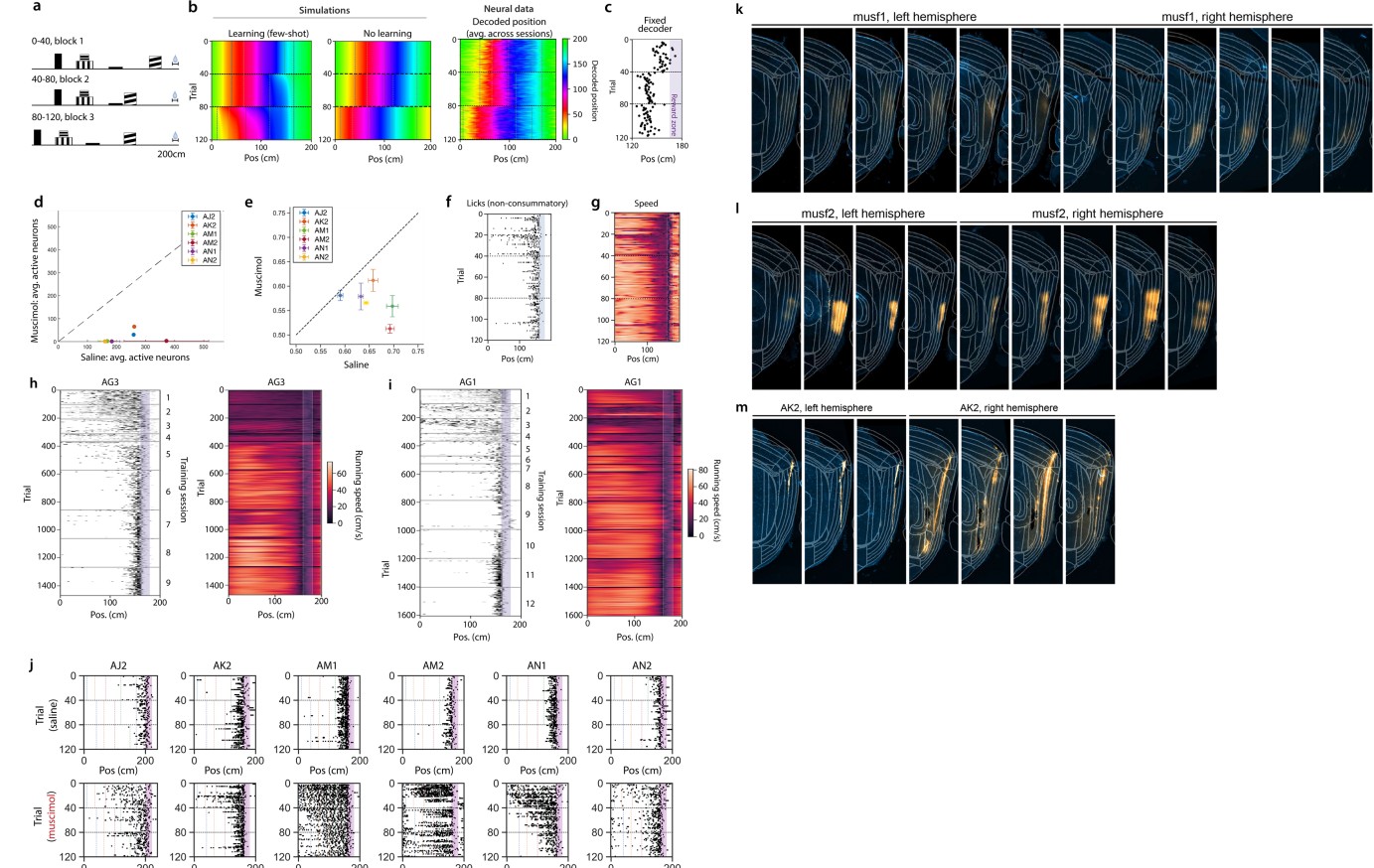

**Extended Data Fig. 9 | MEC coding and behavioral performance on the hidden reward task, and the effect of muscimol. a-c**. Neural activity in the "backward shift condition", analogous to what is shown in Fig. 4a,b,e. Behavior is shown in Fig. 4f,g. **a**. Schematic of the hidden reward task for the "backward shift condition". The unmarked reward zone is indicated by the water drop. The mouse traveled a variable distance from the start of each trial to the first landmark. **b**. *Left-Middle*: Decoded position from a population of simulated grid cells which exhibit few-shot learning (*left*) and no learning (*middle*) on the "backward shift condition". Horizontal black dotted lines correspond to the trial in which the location of the visual landmark shifts. Vertical lines indicate the location of the landmarks for each block of trials. *Right*: Decoded spatial position from a linear decoder trained on the population activity of simultaneously recorded grid cells in a tower-shift experiment, averaged over modules, sessions, and animals (animal n = 5, session n = 10, module n = 29, cell = 1325). **c**. Predicted lick locations for a fixed decoder trained on all three blocks in the "backward shift condition". Dots represent licks. **d-g**. Muscimol-induced disruption of neural activity is correlated with navigational impairments. **d**. Scatterplot of the average number of well-isolated, active neurons recorded in MEC following bilateral infusion of saline or muscimol. Each dot represents the mean for a single animal, with standard error of mean error bars where appropriate (>1 recording for a given condition). Dashed black line represents the unity line. Pooled across all sessions, muscimol significantly reduced the number of active units (two-tailed unpaired t-test, p < 1e-4, animal n = 7; saline sessions n = 11, muscimol sessions n = 8). Note that animal AK2, represented by the orange point, is a potential outlier, since the number of recorded active units following muscimol infusion was greater than those for other animals.

See panels (f-g and l) for related behavior and histology from the muscimol session for this animal. **e**. Scatterplot of average grid cell stability (quantified by average Pearson r correlation of grid cell population activity on one trial to other trials within a block) following bilateral infusion of saline or muscimol. Each dot represents the average Pearson r for a single animal, bars indicate standard error of mean error where appropriate (>1 recording for a given condition). Dashed black line represents the unity line. Compared to saline, muscimol significantly reduced grid map stability (Wilcoxon rank sum test, p < 1e-4, animal n = 6; saline sessions n = 10, muscimol sessions n = 8). **f-g**. Animal AK2's licking (f) and running (g) behavior on the hidden reward track following bilateral fluorescent muscimol injection. Consistent with the incomplete silencing of MEC activity in (d) and partial spread of fluorescent muscimol in (l), licking and running behavior were not as impaired as for other animals with more complete MEC silencing. **h-i**. Behavioral performance on the hidden reward task over the course of training for two mice. *Left*: Non-consummatory licks, dots indicate licks and the purple bar indicates reward zone. *Right*: Running speed. Horizontal lines indicate training sessions. Over the course of training, animals learn to slow down and lick in anticipation of the hidden reward zone (purple). For training details see Methods. **j**. Non-consummatory lick rasters for saline and muscimol sessions from 6 animals. Purple shaded region represents the reward zone. **k-m**. Example histology showing the extent of fluorescent muscimol (in orange) spread in three animals (musf1 (k), musf2 (l), and AK2 (**m**)). Slices from the left hemisphere are ordered from most lateral to most medial, and vice versa for those from the right hemisphere. Fluorescence is primarily localized to the MEC, but a small amount can be found in the parasubiculum for a few slices.

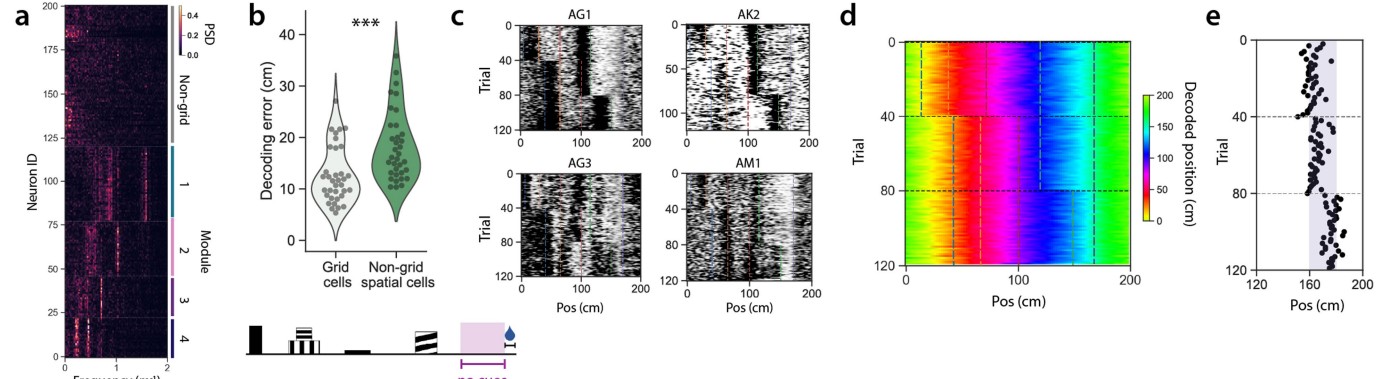

**Extended Data Fig. 10 | Non-grid spatial cells in the hidden reward task.**
**a**. Power spectra for the same session as in Fig. 1c, but including all simultaneously recorded MEC neurons, including non-grid cells (labeled by gray band at *right*), which show no clear periodic structure. Power spectra for each neuron color coded for maximum (white) and minimum (black) power in each frequency, sorted dorsal (top) to ventral (bottom). **b**. Absent any visual cues, in the dark 15 cm region preceding the reward zone (*pink, bottom*), grid cells provide more faithful spatial information than non-grid spatial cells. Grid cells are identified based on their firing properties in the dark; non-grid spatial cells are identified within each block as those cells whose tuning curves in the first half of the block (20 trials) have a spatial correlation greater than 0.2 with their tuning curve in the second half of the block. Linear circular decoders are trained using equal numbers of grid and non-grid spatial cells (chosen as the minimum of the two) to predict all locations along the track, and are evaluated only on the 15 cm preceding the reward zone, where the animal is in the dark. Decoding error is significantly smaller for a decoder trained on grid cell firing, across blocks, animals, and sessions (two-sided Wilcoxon signed-rank test, p = 9.35e-8, block n = 36; animal n = 4, session n = 6, cell n = 318). **c**. Example non-grid spatial cell rasters in the tower-shift task (animal n = 4). These cells exhibit firing fields closely tied to landmarks. **d**. Figure 4b repeated with a decoder trained on all MEC cells, rather than just putative grid cells. **e**. Figure 4e repeated with a decoder trained on all MEC cells, rather than just putative grid cells.

# Reporting Summary

## Statistics

For all statistical analyses, confirm that the following items are present in the figure legend, table legend, main text, or Methods section.

| n/a | Confirmed | |
|---|---|---|
| ☐ | ☒ | The exact sample size ($n$) for each experimental group/condition, given as a discrete number and unit of measurement |
| ☐ | ☒ | A statement on whether measurements were taken from distinct samples or whether the same sample was measured repeatedly |
| ☐ | ☒ | The statistical test(s) used AND whether they are one- or two-sided<br>*Only common tests should be described solely by name; describe more complex techniques in the Methods section.* |
| ☐ | ☒ | A description of all covariates tested |
| ☐ | ☒ | A description of any assumptions or corrections, such as tests of normality and adjustment for multiple comparisons |
| ☐ | ☒ | A full description of the statistical parameters including central tendency (e.g. means) or other basic estimates (e.g. regression coefficient) AND variation (e.g. standard deviation) or associated estimates of uncertainty (e.g. confidence intervals) |
| ☐ | ☒ | For null hypothesis testing, the test statistic (e.g. $F$, $t$, $r$) with confidence intervals, effect sizes, degrees of freedom and $P$ value noted<br>*Give P values as exact values whenever suitable.* |
| ☒ | ☐ | For Bayesian analysis, information on the choice of priors and Markov chain Monte Carlo settings |
| ☒ | ☐ | For hierarchical and complex designs, identification of the appropriate level for tests and full reporting of outcomes |
| ☐ | ☒ | Estimates of effect sizes (e.g. Cohen's $d$, Pearson's $r$), indicating how they were calculated |

*Our web collection on statistics for biologists contains articles on many of the points above.*

## Software and code

Policy information about availability of computer code

| | |
|---|---|
| Data collection | Custom code was written for Unity 5.6.3p2 for virtual reality experiments. Neuropixels data were collected using spikeGLX. |
| Data analysis | All custom code is available at the following website: https://github.com/GiocomoLab/mec-rapid-learning/tree/main.<br>Kilosort2 was used for spike sorting offline.<br>Phy version 1 was used to manually inspect and curate clusters after spike sorting.<br>SHARP-Track was used to reference histological slices to the Allen Brain Atlas and to infer probe locations from dyes.<br>DeepLabCut version 2.2.0.6 was used to identify animals' position and head direction from green and red LEDs on a headstage holder. |

For manuscripts utilizing custom algorithms or software that are central to the research but not yet described in published literature, software must be made available to editors and reviewers. We strongly encourage code deposition in a community repository (e.g. GitHub). See the Nature Portfolio guidelines for submitting code & software for further information.

## Data

Policy information about <u>availability of data</u>

All manuscripts must include a <u>data availability statement</u>. This statement should provide the following information, where applicable:
- Accession codes, unique identifiers, or web links for publicly available datasets
- A description of any restrictions on data availability
- For clinical datasets or third party data, please ensure that the statement adheres to our <u>policy</u>

All data required to reproduce the paper figures are available in two parts on Mendeley Data:

Wen, John; Sorscher, Ben; Giocomo, Lisa (2024), "One-shot entorhinal maps enable flexible navigation in novel environments - part 1", Mendeley Data, V1, doi: 10.17632/rgtk6jygjc.1

Wen, John; Sorscher, Ben; Giocomo, Lisa (2024), "One-shot entorhinal maps enable flexible navigation in novel environments - part 2", Mendeley Data, V1, doi: 10.17632/2n4t9bw3xz.1

A publicly available database used to register histological sections to a reference mouse brain atlas is available here: https://figshare.com/articles/dataset/ Modified_Allen_CCF_2017_for_cortex-lab_allenCCF/25365829

## Research involving human participants, their data, or biological material

Policy information about studies with <u>human participants or human data</u>. See also policy information about <u>sex, gender (identity/presentation), and sexual orientation</u> and <u>race, ethnicity and racism</u>.

| | |
|---|---|
| Reporting on sex and gender | No human data was used for this study. |
| Reporting on race, ethnicity, or other socially relevant groupings | No human data was used for this study. |
| Population characteristics | No human data was used for this study. |
| Recruitment | No human data was used for this study. |
| Ethics oversight | No human data was used for this study. |

Note that full information on the approval of the study protocol must also be provided in the manuscript.

# Field-specific reporting

Please select the one below that is the best fit for your research. If you are not sure, read the appropriate sections before making your selection.

☒ Life sciences  ☐ Behavioural & social sciences  ☐ Ecological, evolutionary & environmental sciences

For a reference copy of the document with all sections, see nature.com/documents/nr-reporting-summary-flat.pdf

# Life sciences study design

All studies must disclose on these points even when the disclosure is negative.

| | |
|---|---|
| Sample size | Sample sizes were determined in part by what is conventionally required in studies with mice. For each experiment, at least 4 different mice were used to ensure replication. |
| Data exclusions | Some analyses required a sufficient number of co-recorded grid cells from a module. These analyses are explicitly stated in the manuscript. Recording sessions that fail to meet this criterion were not used for these analyses. |
| Replication | Replication was verified by conducting the same experiment in different animals and across different days. 19 sessions were conducted across 4 animals on the build-up track. 38 sessions were conducted across 11 animals on the random environment track. 16 sessions were conducted across 7 animals on the hidden reward track. 3 sessions were conducted across 3 animals on the open field and 1D head-fixed dark runs. 10 muscimol sessions and 13 saline control sessions were conducted across 7 animals on the hidden reward track. Each of the seven animals injected with muscimol underwent at least one control saline session. For more details, see the associated supplementary table. |
| Randomization | Generally, randomization of subjects into different groups was not relevant in our study as most comparisons were made within animal, and all animals used within an experiment were subject to the same behavioral and electrophysiological data collection protocol. However, for our |

inactivation studies using muscimol and saline control, which substance was used on the first day of recording was randomized across subjects.

Blinding | For most experiments, blinding was not necessary as there were no "treatment" groups. The only treatment compared muscimol to saline injections. The data from those experiments were analyzed blind, but because of the drastic effects of muscimol, far fewer recorded neurons were present in the muscimol-injected cases.

# Reporting for specific materials, systems and methods

We require information from authors about some types of materials, experimental systems and methods used in many studies. Here, indicate whether each material, system or method listed is relevant to your study. If you are not sure if a list item applies to your research, read the appropriate section before selecting a response.

## Materials & experimental systems

| n/a | Involved in the study |
|---|---|
| ☒ | Antibodies |
| ☒ | Eukaryotic cell lines |
| ☒ | Palaeontology and archaeology |
| ☐ ☒ | Animals and other organisms |
| ☒ | Clinical data |
| ☒ | Dual use research of concern |
| ☒ | Plants |

## Methods

| n/a | Involved in the study |
|---|---|
| ☒ | ChIP-seq |
| ☒ | Flow cytometry |
| ☒ | MRI-based neuroimaging |

## Animals and other research organisms

Policy information about studies involving animals; ARRIVE guidelines recommended for reporting animal research, and Sex and Gender in Research

Laboratory animals | Mice: C57Bl/6, between 12-24 weeks of age.

Wild animals | This study did not involve wild animals.

Reporting on sex | Only females were used for the virtual reality experiments in this study. This was for the following reasons: 1. In our lab, we have found that female mice typically learn faster than male counterparts in on our head-fixed virtual reality setup. 2. Female mice are less aggressive, reducing the chance of headbar detachment when on the head-fixed virtual reality setup.  Three male mice were added during the revision stage to record the activity of freely moving mice. Male mice were used as they are larger and can better support the weight of the chronic implant.

Field-collected samples | This study did not involve samples collected from the field.

Ethics oversight | All procedures were approved by the Institutional Animal Care and Use Committee at Stanford University School of Medicine

Note that full information on the approval of the study protocol must also be provided in the manuscript.

