## [Peer Review File · Nature]

Manuscript Title: One-shot entorhinal maps enable flexible navigation in novel environments

Redactions – Third Party Material

Reviewer Comments & Author Rebuttals

Reviewer Reports on the Initial Version:

Referees' comments:

Referee #1 (Remarks to the Author):

Wen, Sorscher et al present analysis and interpretation from recordings of thousands of neurons in MEC during virtual reality tasks. The motivation for this work was to determine how new environment features are incorporated into the grid cell map to act as landmarks. The authors first describe their methods to track the attractor bump of activity across the grid cell torus manifold. When animals ran in a straight path in the dark on a treadmill, there was apparent coherence across grid cell modules, but drift in the trajectory of the bump around the torus as path integration errors accumulated. As the authors added rewards and towers to the linear track that the mouse was running along, the bump trajectory was more and more constrained to a particular path. The authors presented some analysis to show that the path was curved (not straight) in the presence of the tower landmarks. From the speed of change in the grid cell network and analysis of path curvatures, the authors conclude that the grid cell activity was weakly anchored to landmark inputs, but in a way not requiring any plasticity. This seems to have led to the conclusion of “one-shot” entorhinal maps. In a separate task the authors shifted visual tower cues with respect to a reward location and found that the grid cell map shifted with the cues, but the animal’s behavior was largely unchanged and quickly adapted to the manipulation. The main questions addressed here are interesting and timely; however, I was underwhelmed by the results, analysis and interpretation and largely feel that the conclusions are not strongly supported. I was left wondering what the solid conclusions were, and how they differ from previous work on grid cells. Furthermore, I found the manuscript to be extremely difficult to read, with leaps in logic that I did not follow and complicated analysis methods that were not well described. Even if these issues could be fixed, the findings and details presented here do not seem a good fit for a general audience. I elaborate on the issues below.

Major points:

1. The authors present a new method to classify grid cells and decode their location in the neural space/physical space while mice are head-fixed and running in the dark. This method has the potential to become a simpler way than previous methods to classify grid cells without having to record ephys/endoscopes in an open-field environment. However, to be accepted and adopted by the community the authors must validate the method first in an open field environment. Without a ground truth of open field recording, it is difficult to estimate the advantages and limitations of this method.
 - a. How robust is the method? For example, it is not explained how conjunctive cells (grid cells + head direction) are handled and how they may interfere with the computations performed (relatedly,

there is no discussion about what layers are targeted and how that may influence the results). Spectrograms and dynamics were computed only on 10 laps (~30 m) (Line 102). Why was this not computed for the full session, or computed on different sections to see how consistent the measures are?

b. The Authors compute the spectrogram for each neuron's spike train using a window/lap of 3-5 m (line 703-704) (the spatial frequency band in Figure 1 is 0 to 2 periods per meter). Why was this window chosen? Mouse grid cells can be as tight as 30-40 cm spacing, which would necessitate a maximum spatial frequency of at least 3 periods per meter. How does this method succeed/fail in classifying grid cells in this spatial frequency band? Without a ground truth of open field recording this cannot be answered.

c. The mice are running on a circular treadmill in the dark. It is necessary to show that the observed periodicity in the cells is not affected by any physical treadmill periodicity since grid cells could entrain to periodic sensory cues from the treadmill (imperfections in the surface, odors, etc.). The treadmill circumference is ~50 cm. The authors do not show the power spectrogram of the cells above 2 periods per meter (it appears 2 is excluded).

2. Measures of behavior seem to be missing from Figures 2-3. Is the animal engaged in the task, and are they actually paying attention to the visual cues? With animals exposed to so many different environments and manipulations in a single session, it seems that task engagement could play a major role in altering activity patterns in MEC (see for example Pettit et al, Nat Neuro 2022 for example effects on place cells). In fact, this is supported by the authors finding that the grid population remaps when the animal is exposed to the same environment across multiple days. Finally, I wonder if the animal interprets the landmarks in the way that the authors intended. For example, in Figure 3 the five towers are scrambled to different locations twelve different times. Even if the animals are paying attention to the towers, do they really perceive the towers as landmarks, or perhaps are they encoding the towers as mobile objects that are not reliable stable landmarks (possible with different encoding compared to stable landmarks)?

3. I do not follow the logic in 181-221 (Figure 3). It seems as though the authors are saying that, because plasticity in the fly system can take several minutes, then any stable change in the grid cell modules in mice that is faster than this must not require plasticity? There are many examples of rapid plasticity mechanisms in cortex and hippocampal systems, including the BTSP the authors cite, which seems to undermine this logic, if this is in fact what the authors intended. Additionally, since it is well known that maps form in the hippocampal system in single trials (for example, many place cells are present on the first traversal of a new environment) and grid cell rapid remapping has also been described (Fyhn 2007, Yoon K et al. 2013, Barry et al. 2012), then it is unclear what is new here. What is the novelty of the "one-shot" map finding? Also related to the logic and conclusions of 181-221 (Figure 3), I do not see evidence that the distorted bump trajectories are caused by the landmarks "tugging" on the bump. I see evidence (Figu 3C,D) that the bump trajectories are distorted, but not necessarily "tugged" by the landmarks. The predictive model analysis results related to this point (3H) do not appear very strong. Could the "distortions" around landmarks just be the animal sometimes paying attention to the cues (when they are close) and sometimes not (see point above)?

4. The data for "flexible adaptive behavior" do not strongly support the conclusions that are drawn

(line 248). There is minimal support for MEC grid cells endowing this system with “powerful computational properties” (lines 21-22). Certainly grid cells may be a critical part of the system, but the behavioral adaptation and flexibility is likely occurring somewhere else. If anything, the data presented is consistent with this view. The extensive computational and theoretical literature on how this could be implemented is largely ignored. Whittington et al – Nat Neuro 2022, for example, provides a cohesive way of considering grid cells in this system. The model of downstream BTSP doesn’t add much insight and if anything suggests that the main focus of the present manuscript is not where the interesting computation related to behavior is happening.

5. The premise/motivation presented in the manuscript oversimplifies what is known by trying to set-up the grid cell network as the primary and dominant source of spatial information during behavior. The contribution of the remaining ~75% of MEC cells, most HPC cells and all LEC cells, for example, are minimally considered.

Minor points:

128 -129: “Together, these results indicate that even in the absence of visual landmarks, the grid cell population is well described by low-dimensional attractor dynamics.” This has been already shown in RJ Gardner 2022 et al. where they show in an unsupervised method that the topology of torus preserved during sleep (no sensory cues). The authors’ method is supervised and assumes in advance the topology of the neural space. This is weaker and not sufficiently novel.

137-139: “In recordings that captured the activity of more than one module of grid cells, we observed that activity bumps from different modules were tightly coupled and drifted in concert, taking the same overall trajectories.” This was already shown in Waaga, T. et al. 2022. The authors put citation without explicitly saying that the result is consistent with this already published paper.

177-179: “This provides a remarkably simple understanding of remapping as just taking a new trajectory on a toroidal attractor, thereby demystifying the complex re-organization of individual grid cells during a remapping event, a process that can appear random aside from the preserved correlations between grid cells.” It is already known that at the single-cell level, the phase between cells is preserved during remapping, which corresponds to a different trajectory on the torus at the population level. The authors need to explain how the two findings differ. A citation of Fyhn et al 2007 is missing.

219-221: “We found that grid cells developed different maps in the two repeated environments, despite them being visually identical.” This result is inconsistent with papers of 2d open field environments where mice are switched to a novel environment and then returned to the original environment (A -> B -> A’). Grid cells were found to be stable between A and A' (Fyhn et al 2007). Could the differences observed in the present work potentially point to low attention to the VR environments (as noted above)?

Referee #2 (Remarks to the Author):

This is a very interesting paper. The authors have performed some excellent experiments, analysis, modelling, and theory work. The final result is a novel idea about how the entorhinal-hippocampal area functions. The manuscript is mostly clear, well-written and well-presented. This paper should have a substantial impact on the field. I do have a few clarifications/questions that are listed below.

1) The most obvious issue is that there is some serious data selection here (ie which neurons are to be included for further analysis). We go from 68k active neurons to 15k grid cells and then to hundreds and tens of co-modular grid cells (grid cells from one power spectra module) in the construction of network (bump) activity. Do the authors have a principled reason for not including (the majority of?) neurons that did not show grid cell like activity? How does inclusion of non-grid cells into the population activity pattern alter that activity? For that matter, what happens when the activity of multiple grid-cell modules are combined? I realize the authors are trying to do something analogous to dimensionality reduction to describe the activity of co-modular grid cells. But they are also trying to make some functional conclusions relating to a role for synaptic plasticity in network remapping of novel environments. Can plasticity among the non-grid cells be involved in the learning of new maps for altered environments? Are any new grid cells formed during the novel environments that perhaps did not meet the grid cell criteria because of untuned activity before the environmental change? Also given all of the neuronal activity from non-grid cells in the EC, how could only the activity of grid cells (particularly one module) be read out downstream? Some additional discussion concerning these issues would be helpful.

2) The fixed and weak landmark to grid map pinning system is an interesting idea. But the activity generated by the model did not strongly correlate with the actual activity recorded from the neurons used to produce the model. There is a large amount of overlap between the distribution of correlations for random conditions and even the best model (fig 3i; plots in Sfig 6). I couldn't find any discussion of the idea that perhaps a continuum exists with different attractor networks spanning the range from purely idiothetic to strong landmark pinning. Did the authors find any evidence that some modules appeared to be mostly idiothetic and others strongly pinned to landmarks? Perhaps the result presented reflects an average "level of pinning" that spans a large range? Some additional discussion of these issues and the parameter, α , could help here. Do the authors consider each grid cell module to inhabit a single, unique attractor network?

3) Finally, the idea that neuronal activity in a grid cell module is constrained to move around in a defined and continuous space when a mouse is moving in an environment is interesting. But what happens to the bump of activity within the attractor when the animal stops running? Does it stay at the same location in activity space (ie some sort of persistent activity)? Does it move around in the same activity space or a different space? If it moves around, how does it find its way back to the appropriate location in activity space? I suppose this is mostly relevant for when there are rewards present in the task. Does the animal stop and consume the reward for some variable amount of time? What happens to the neuronal activity during this standing period? If the activity here is not simply persistent can the weak pinning model handle the standing activity as well as the running?

4) Please add at least one additional figure for ease of visualization.

Referee #3 (Remarks to the Author):

This paper presents remarkable experiments analyzed in clever ways. It is hard to quibble about 68,484 recorded cells, although I will manage to do that below. As I understand it, the main points are: 1) The neural sheet that forms the basis of the attractor model of grid cells remains constant, at least to a good approximation, across all the manipulations reported. 2) In the absence of external landmarks, the angle that activity traverses across the neural sheet diffuses. 3) In the presence of landmarks, the angle of the trajectory along the neural sheet rotates in such a way that grid cell response are periodic over the length of a lap. 3) Landmarks are mapped to fixed points on the neural sheet. 4) Landmarks distort the path of activity across the neural sheet, resulting in curved trajectories. 5) Downstream learning can compensate for these distorted trajectories using BTSP. The paper is difficult to follow for reasons given below, and it has a number of puzzling aspects.

1) In the description above, I have used neural sheet in the way that the authors do, but they also use the term grid cell map. I am not sure what that is, and how it relates to the neural sheet. It would be good to include a clear definition of these terms and to use them consistently.

2) The grid cell system is built to describe motion in 2 dimensions. The authors act as if, on this linear track, the animal is representing the world as 1D and that the curvature of the path on the neural sheet is a distortion of a 1D world. What if it is a 2D trajectory? Shouldn't this possibility at least be considered? As I understand it, the introduction of landmarks induces small curvature in the neural sheet path that straighten out as more landmarks are added and the grid response become periodic. Couldn't this be interpreted literally as the animal thinking initially that it is not moving in a straight line, perhaps due to an imbalance in the placement of the landmarks, and later realizing that it is?

3) The authors assume that the landmarks they use are totally accepted by the animals, but that their effect on the neural sheet trajectory is weak. They then provide a fairly long, and to me mystifying, discussion of why this is a good idea as a general policy. The trajectory presumably reflects the best estimate of where the animal thinks it is. Maybe it considers the artificial landmarks used in this experiment to be weak evidence of position and is weighing them appropriately. Would the effect be stronger if more convincing evidence of position was provided? What about textures on the track or odors? It seems to me that the idea of weak evidence weighting should be eliminated before such a strong case for the optimality of a weak effect is made.

4) 68,484 cells is remarkable, but the paper is based on 15,000 of them. The others are dismissed as having little apparent spatial structure. Does this mean little apparent spatial modulation? I am not suggesting an analysis of the remaining cells, but perhaps they could be dismissed more informatively.

5) The authors argue that the MEC can do one-shot learning without plasticity, which is a roundabout way of saying nothing is happening in the MEC. Listing this as a virtue seems far-fetched. Evidence of position is being weighed outside the MEC, and MEC activity is being used to read out the results of that weighing, using the structure of grid cells to decipher the signal. Isn't that a more reasonable statement of what is going on?

5) A very nice result is the extraction of the diffusion constant for the angular drift. It is stated that this is consistent with other studies of animals getting lost in the dark, but a quantification of this comparison would be helpful.

6) Although this is point 6, it introduces what was, for me, the most frustrating aspect of the paper. The distortions the authors are discussing are associated with the animal having more information about where it is, but in the paper they are presented as something that needs to be compensated for, rather than as something carrying more information. It is as if the animal is more confused in the presence of landmarks, but that downstream circuits can compensate for this confused representation. A title states that "One-shot MEC maps enable flexible adaptive behavior despite small distortions". Almost nothing in this title makes sense to me. Why is it one-shot - it is fast but is it really one-shot? How can the MEC enable something despite distorting it? The paper does not present the distortions as anything positive or show that they increase accuracy, yet they are the result of introducing landmarks. This is, to me, the most confusing aspect of the paper.

7) The part of the paper about BTSP is interesting, but it is presented so briefly as to appear as an afterthought. It would be good to expand this presentation.

8) I found the discussion particularly confusing. The most intuitive idea would be what the authors call strong pinning - when the animal sees a landmark it places itself in that position. Is the problem with this that it would require a remapping? If so, what is wrong with that? The authors state that "The weak, fixed, landmark to grid cell pinning solution favored by MEC is thus a computationally powerful approach enabling one-shot learning of grid maps in new or dynamic environments, with small distortions, that can then subserve flexible adaptive behaviors through downstream plasticity". What does this mean? As stated above, the grid cells seem to be doing nothing other than responding in their usual way to inputs that have been modified by new evidence. Is this one-shot learning of a grid map or just an existing grid map responding to different input? Why would a system require downstream plasticity to decode more accurate information about where the animal is? Is a system that introduces a distortion and then compensates for it computationally powerful?

Minor points:

1) What is a twisted torus?

2) The abstract is particularly cryptic.

Author Rebuttals to Initial Comments:

We thank the reviewers for their helpful comments and constructive criticism. We have made significant changes to the analyses and figures in accordance with their suggestions. We feel these changes strengthen the paper's conclusions and broaden its novelty. Below, we provide a brief summary of the changes we made as well as a detailed response to each Reviewer suggestion. We would note that we significantly edited the text to improve the clarity of the findings and generated multiple new figure panels - many of which we have added to the Main or Extended Data Figures.

Summary of major changes in response to reviewers:

1. We have added new data in which we recorded, in the same mouse, medial entorhinal cortex neurons in both freely moving 2D environments and the head-fixed complete darkness condition (animal $n = 3$, cell $n = 504$). This allowed us to examine, with ground truth, how well the grid cell classification based on 1D dark recordings corresponds to grid cell classification based on 2D spatial firing rate patterns. We find very high overlap between these classification systems and have included this data in Figure 1 and new Extended Data Figure 3. 2D recordings also allow us to examine conjunctive head-direction coding.

2. We have extensively revised the abstract, results and discussion sections of the manuscript. We hope that this better highlights our findings, the novelty of the results and the relationship of our work to prior studies. We simplified Figures 1 and 3 by moving some panels to Extended Data. We have also included a new schematic to illustrate the competing hypotheses of rapid plasticity between landmark inputs and the grid cell network versus fixed connectivity between landmark inputs and the grid cell network (new Figure 2j-l).

3. We have added new Extended Data Figure panels regarding the anatomical location of our recordings (Extended Data Fig. 1b-c) and the behavior of the mice (e.g. licking accuracy and running speed, Extended Data Fig. 1d-l). We have also included behavior in the main Figures (Figure 3h).

4. We have added new analyses regarding the population coding of non-grid cells in MEC, and included two new figures summarizing these results (Extended Data Figs. 7 and 10). First, we build a predictive model of non-grid spatial cell firing patterns based on linear combinations of VR landmarks, which can be used to predict the firing patterns of non-grid spatial cells (but not grid cells) in held-out environments. Second, we extend our population decoding analyses to the entire population of MEC cells, not just grid cells. We find that these cells deform in a similar manner to grid cells when we shift VR landmarks, and offer no new spatial information not already captured by the grid cell code.

5. We have performed new analysis (see Extended Data Fig. 6) showing that the speed of the bump of activity on the neural sheet tracks the speed of the animal in the VR environment, and that the bump (noisily) tracks the animal's position even when the animal is stationary.

Please note that we have provided some figure panels to aid in the response to reviewer comments that are not currently part of the Main or Extended Data Figures. If the reviewers or editor feel that these would clarify the manuscript, we would be happy to incorporate them.

Referee #1 (Remarks to the Author):

Wen, Sorscher et al present analysis and interpretation from recordings of thousands of neurons in MEC during virtual reality tasks. The motivation for this work was to determine how new environment features are incorporated into the grid cell map to act as landmarks. The authors first describe their methods to track the attractor bump of activity across the grid cell torus manifold. When animals ran in a straight path in the dark on a treadmill, there was apparent coherence across grid cell modules, but drift in the trajectory of the bump around the torus as path integration errors accumulated. As the authors added rewards and towers to the linear track that the mouse was running along, the bump trajectory was more and more constrained to a particular path. The authors presented some analysis to show that the path was curved (not straight) in the presence of the tower landmarks. From the speed of change in the grid cell network and analysis of path curvatures, the authors conclude that the grid cell activity was weakly anchored to landmark inputs, but in a way not requiring any plasticity. This seems to have led to the conclusion of “one-shot” entorhinal maps. In a separate task the authors shifted visual tower cues with respect to a reward location and found that the grid cell map shifted with the cues, but the animal’s behavior was largely unchanged and quickly adapted to the manipulation. The main questions addressed here are interesting and timely; however, I was underwhelmed by the results, analysis and interpretation and largely feel that the conclusions are not strongly supported. I was left wondering what the solid conclusions were, and how they differ from previous work on grid cells. Furthermore, I found the manuscript to be extremely difficult to read, with leaps in logic that I did not follow and complicated analysis methods that were not well described. Even if these issues could be fixed, the findings and details presented here do not seem a good fit for a general audience. I elaborate on the issues below.

We thank the reviewer for their comments and constructive feedback. We have now included new experimental data to better map 1D to 2D grid cells, extensively clarified the text in a manner we hope better communicates the novelty and conclusions of the work, and performed a number of analyses, which we address in detail below.

First however, to address the issue of novelty raised near the end of the above paragraph, we do feel that one of the fundamentally new aspects of our work, is the ability to *quantitatively predict* the detailed distortions of grid cell firing patterns in a completely novel environment consisting of randomly located familiar landmarks, *without ever* using recordings from that novel environment, via an extremely simple dynamical system consisting of only 2 variables (a 2D attractor phase), using only 2 learned parameters per landmark (a fixed 2D attractor pinning phase that is the *same* across all prior environments and the new one) and 2 learned parameters per cell (the cell’s location on a toroidal neural sheet) (e.g. see Fig. 3g,i,j). To our knowledge, no prior work has been able to quantitatively predict such detailed grid cell distortions in a new environment without data from that environment, let alone with such a simple model. Furthermore, the predictive accuracy of our model, combined with the fixed nature of landmark inputs in the model, as well as the rapid single trial remapping of grid cells upon the inclusion of novel landmarks, together provides strong evidence that this single trial remapping can occur with fixed connectivity from landmarks to grid cells - a notion that was not clearly understood prior to our work.

Major points:

1. The authors present a new method to classify grid cells and decode their location in the neural space/physical space while mice are head-fixed and running in the dark. This method has the potential to become a simpler way than previous methods to classify grid cells without having to record ephys/endoscopes in an open-field environment. However, to be accepted and adopted by the community the authors must validate the method first in an open field environment. Without a ground truth of open field recording, it is difficult to estimate the advantages and limitations of this method.

The reviewer raises an important point. To address this concern, and provide ground truth for our new 1D method, we now include new data from 4 mice in which we chronically implanted Neuropixels 2.0 probes (New Extended Data Figure 3 - provided below for ease of reference). This approach allowed us to record from the same medial entorhinal neurons as head-fixed mice ran in complete darkness and in an open-field environment. Methods for these experiments are in the appropriate section and the results described in the main results section, Figure 1 and Extended Data Figure 3 (provided below for ease of reference). Briefly, for each chronically recorded mouse, we conducted a recording session in which mice ran head-fixed on a dark, featureless 1D track with no water rewards for at least 140 trials (200 cm per trial). Immediately afterwards, mice were removed from the head-fixed condition and allowed to freely explore a 75 cm open field with vanilla sprinkles randomly scattered throughout for 60 - 70 minutes. For chronic recordings, neural data streams from the head-fixed and freely moving conditions (for each session) were concatenated using CatGT (<https://github.com/billkarsh/CatGT>) before spike sorting. This allowed us to examine the same cells across head-fixed dark and freely-moving open arena conditions.

We then quantified the consistency between grid cell identification on the 1D VR track in the dark, and in 2D open environment. To identify grid cells in 2D, we used the method described in [Gardner et al., Nature, 2022]. Briefly, we performed UMAP on spatial autocorrelograms, and clustered the resulting 2D representations (Extended Data Figure 3d-f). In our data, we found this approach slightly less robust than described in Gardner et al., in that it was not always consistent in separating distinct grid modules, perhaps due to differences between rats and mice or because we have fewer grid cells in each module compared to the data set used in Gardner et al. Even so, we were able to reliably separate grid cells from non-grid cells in 2D (Extended Data Figure 3a-c versus k-m). Moreover, we observed good agreement between grid cell identification in 2D and 1D VR track in the dark (*Methods, average false negative rate: 10.3%, average false positive rate: 15.8%, cell $n = 504$, session $n = 3$, animal $n = 31e-5$*). We suspect that the false negative rate is overestimated, because our 1D VR detection method allows for the identification of grid cell modules too large to be resolved in the 75cm wide 2D environment (e.g. Module 2 in Extended Data Fig. 3a, inferred grid scale = 76cm, and Module 3 in Extended Data Fig. 3b, inferred grid scale = 65cm). Aside from these large scale modules, this quantitative perspective matched with the visual appearance of the tuning curves of cells in 2D, with nearly all cells identified in the dark exhibited tuning curves consistent with grid cells (multiple firing fields offset by 60 degrees) in 2D (Extended Data Figure 3a-c). We additionally plot grid score histograms (Extended Data Fig. 3h-j) for all cells (gray), and for cells identified as grid cells on the 1D VR track in the dark.

These new experiments are highly consistent with our observations using only grid cells identified in head-fixed dark conditions, namely that such cells exhibit cardinal properties of grid cells observed in freely moving animals (e.g. modular organization of grid scale, increasing grid scale from dorsal to ventral entorhinal cortex, inter-module coordination).

Extended Data Fig. 3: Grid cell identification in 2D open-field environments. **a-c.** 2D firing rate maps for putative grid cells identified based on their power spectra on the 1D VR track in the dark (Methods), for three different animals. Cells are separated into putative modules. We speculate that the largest modules in **a** (Module 2) and **b** (Module 3) did not show clear grid-like tuning in 2D because their inferred 2D grid scales, 76cm and 65cm, respectively, were too large to exhibit clear periodic firing in the 75cm wide environment. Color coded for minimum (blue) and maximum (red) values. **d-e.** Spatial autocorrelograms for all cells from each of the recordings in **a-c** are collected and projected into 2D via UMAP (Methods), as in ⁴⁸. This projection reveals a cluster of cells with periodic tuning spatial maps, and a cluster without. Putative grid cells identified based on their power spectra on the 1D track in the absence of visual cues are shown in red, and all cells in grey. The cluster of cells with periodic tuning, identified via KMeans clustering, had high agreement with grid cells identified based on their power spectra on the 1D track in the absence of visual cues (Methods, average false negative rate: 10.3%, average false positive rate: 15.8%, cell $n = 504$, session $n = 3$, animal $n = 3$). **h-j.** Grid scores distributions for the recordings in **a-c** (gray) and for putative grid cells identified based on their power spectra on the 1D track in the absence of visual cues (red) (Methods). **k-m.** All cells from the recordings in **a-c** not classified as putative grid cells based on their power spectra on the 1D track in the absence of visual cues.

a. How robust is the method? For example, it is not explained how conjunctive cells (grid cells + head direction) are handled and how they may interfere with the computations performed (relatedly, there is no discussion about what layers are targeted and how that may influence the results).

To address the robustness of the method, we have collected and included new data from mice running head-fixed in complete darkness and in an open-field environment (see comment above). In terms of conjunctive cells, we cannot identify this sub-population of neurons in head-fixed conditions as the mouse is constrained to run along a single axis. However, as we collected new experimental data in mice exploring both 1D and 2D environments, we were able to examine a subset of grid cells with head direction tuning across both conditions. We found a subset of neurons (first two rows in the image below) that were identified as grid cells in the open field and head-fixed dark condition showed some head direction tuning in the open field (mean vector length > 0.1, determined by the 99th percentile of mean vector lengths from a shuffled distribution). This analysis is shown below in Reviewer Figure 1. Thus, we believe that grid cells with directional tuning are still detectable using the head-fixed analysis in complete darkness. We cannot rule out that our method occasionally misclassifies a conjunctive grid cell as a non-grid cell, but this will be a challenge inherent to any tuning curve classification system (Hardcastle et al., Neuron, 2017).

In terms of histology, we would note that i) most of our recordings targeted the superficial layers of MEC and ii) the dye diffuses, making it difficult to differentiate which exact neurons came from layers II versus III, for example. However, prior work has shown that grid modules are coordinated across layers and extend across large medio-lateral distances (~1 mm) (Stensola et al., Nature, 2012; See also Stensola & Moser, Grid cells and spatial maps in the entorhinal cortex and the hippocampus, 2016 and Gu et al., Cell, 2018). As our method requires all neurons to be co-modular, or else we cannot reconstruct the neural sheet and perform the analyses used in the manuscript, it should be layer and mediolateral independent. However, we also agree with the reviewer that the inclusion of histological information is an important component of the work and thus, we have included new subpanels in Extended Data Figure 1 that better indicate the anatomical location of our recordings.

Reviewer Figure 1: Conjunctive grid-head direction cells identified in 1D VR environments in the absence of VR cues. We plot 2D spatial tuning curves for all cells identified as putative grid cells based on our method in the 1D VR environment in the absence of VR cues, along with their 1D head direction tuning curves. Some cells

exhibit head direction tuning, and the first two rows of cells are classified as conjunctive grid-head direction cells based on the mean vector length of their head direction tuning (threshold mean vector length > 0.1, determined by the 99th percentile of mean vector lengths from a shuffled distribution). Head direction tuning curves are annotated with their mean vector length, and cells are sorted from top to bottom based on mean vector length.

Spectrograms and dynamics were computed only on 10 laps (~30 m) (Line 102). Why was this not computed for the full session, or computed on different sections to see how consistent the measures are?

We apologize that this was not clear in the previous version of the manuscript. Since grid cells accumulate error and drift in complete darkness, we extract spectrogram peaks and putative grid cell phases on sections of 10-20 laps at a time. As the reviewer suggests, we verified that the extracted phases are consistent across independent sections in Extended Data Fig. 2e. We have provided additional text to clarify this point in the results and methods sections.

b. The Authors compute the spectrogram for each neuron's spike train using a window/lap of 3-5 m (line 703-704) (the spatial frequency band in Figure 1 is 0 to 2 periods per meter). Why was this window chosen? Mouse grid cells can be as tight as 30-40 cm spacing, which would necessitate a maximum spatial frequency of at least 3 periods per meter.

We thank the reviewer for this important clarification, and have added text to the Methods to expand on this point. The window of 3-5m was chosen to balance resolution in the frequency domain (to achieve good resolution in inferred peak locations) with resolution in the time domain (since the location of the three peaks drifts over many trials in the dark, Fig. 1h). The reviewer is correct that we compute spectrograms using a window of 3-5 m, and using spike trains sampled every 2 cm. With this window, the lowest frequency we can resolve is the frequency resolution, $1/\text{window_size} = 1/300 \text{ cm}$, and the highest frequency we can resolve (given that we bin spikes into 2cm bins) is the Nyquist frequency, $\frac{1}{2} * 1/\text{sampling_rate} = \frac{1}{4} \text{ cm}^{-1}$. Hence our method captures periodicities in the range 4 cm to 300 cm. We believe this dynamic range is wide enough to encapsulate the full range of grid cell frequencies observed in prior work. As the reviewer notes, mouse grid cells can reach as small a periodicity as 30cm, but this is still much larger than the minimum periodicity we can resolve at our sampling rate (4cm). However, it is important to note that the peak frequencies in the spectrogram may be much smaller or larger than the overall grid frequency, depending on the angle of the 1D slice through the 2D lattice. That is, for a module with grid spacing 1/30 cm, the three frequencies observed in 1D are constrained to obey $(f_1^2 + f_2^2 + f_3^2)/4 = (1/30 \text{ cm})^2$, but the individual frequencies f_1, f_2, f_3 can vary widely. In Reviewer Fig. 2, below, we extend the x-axis of Fig. 1 out to 5 periods per meter, to show that our method captures much larger frequencies (we simply do not observe power in these frequencies, in this particular recording).

How does this method succeed/fail in classifying grid cells in this spatial frequency band? Without a ground truth of open field recording this cannot be answered.

First, note our response to the first concern, as we have now provided data from mice that performed both head-fixed running in complete darkness and random foraging in an open-field environment. Using recordings in 2D we verify that even the highest frequencies observed in 2D open field can be resolved by our 1D analysis. See Extended Data Figure 3.

Second, we considered up to 5 periods but, for example, in the recording shown in Figure 1, we did not observe anything in that frequency band, thus we plotted only up to 2 periods. See Reviewer Figure 2 below, in which we plot the data from Figure 1 out to 5 periods. Moreover, in general we find that grid spacing tends to be

larger in VR compared to real-world coordinates, consistent with observations in previous work (e.g. Campbell et al., 2018).

Reviewer Figure 2. Power spectral densities (PSDs) for four different grid cells from four simultaneously recorded modules over multiple trials in the dark, computed over a window size of 10 laps (same session as in main text Fig. 1). The x-axis is extended to 5 m^{-1} .

c. The mice are running on a circular treadmill in the dark. It is necessary to show that the observed periodicity in the cells is not affected by any physical treadmill periodicity since grid cells could entrain to periodic sensory cues from the treadmill (imperfections in the surface, odors, etc.). The treadmill circumference is $\sim 50 \text{ cm}$. The authors do not show the power spectrogram of the cells above 2 periods per meter (it appears 2 is excluded).

The reviewer notes an important control. To address this, we overlaid the periodicity of the circular treadmill in the dark (green dotted line) relative to spectrograms computed across modules, animals, and sessions (shown below in Reviewer Figure 3). Note that the spectrograms reveal no clear peak at the periodicity of the treadmill (or at integer multiples of the periodicity of the treadmill).

Reviewer Figure 3. Average PSDs across all neurons from 14 animals and 89 sessions from the dark block of the random environments task, with the frequencies corresponding to integer multiples of the treadmill length overlaid (green dashed lines). We find no clear evidence for excess power at the frequencies corresponding to the treadmill length (cell $n = 20,718$, animal $n = 14$, session $n = 89$).

2. Measures of behavior seem to be missing from Figures 2-3. Is the animal engaged in the task, and are they actually paying attention to the visual cues? With animals exposed to so many different

environments and manipulations in a single session, it seems that task engagement could play a major role in altering activity patterns in MEC (see for example Pettit et al, Nat Neuro 2022 for example effects on place cells). In fact, this is supported by the authors finding that the grid population remaps when the animal is exposed to the same environment across multiple days. Finally, I wonder if the animal interprets the landmarks in the way that the authors intended. For example, in Figure 3 the five towers are scrambled to different locations twelve different times. Even if the animals are paying attention to the towers, do they really perceive the towers as landmarks, or perhaps are they encoding the towers as mobile objects that are not reliable stable landmarks (possible with different encoding compared to stable landmarks)?

The reviewer makes a set of interesting points. First, to demonstrate that animals are engaged in the task, we have added an example of licking behavior to Figure 3h, as well as licking and running speed behavior from individual sessions on the build-up and the random rearrangement tracks in Extended Data Figure 1. In addition, we have included quantifications of behavioral task engagement for individual sessions and averaged across sessions in Extended Data Figure 1. As mentioned by the reviewer, Pettit et al., Nat Neuro 2022, among others, measured task engagement through anticipatory licking, which reflects the extent of licking near a reward zone relative to other portions of a track. Similarly, we have now included in Extended Data Figure 1 quantification of anticipatory licking, as well as anticipatory slowing, for both the build-up and random rearrangement tracks. Finally, we have also included a measure of licking accuracy for the random rearrangement track. In brief, we find that for both the build-up and random rearrangement tracks, animals show task engagement through anticipatory licking and slowing across blocks.

For the build-up track, anticipatory licking and slowing increase as additional landmarks are added to the track. Importantly, this shows that our animals did not disengage with the task like in Pettit et al, Nat Neuro 2022. As mentioned in that paper, their animals typically disengaged towards the end of a session, after nearly 1 mL of rewards had been consumed. In our VR setup, we have carefully titrated the amount of water delivered per reward (~1.5-2 μ l) so that animals can typically complete at least 240 trials without becoming satiated. However, we should note that while measures of anticipatory licking and slowing increase across blocks, we don't believe this indicates a disengagement with the task on early blocks of the build-up track. Rather, anticipatory behavior likely improves because additional landmarks allow for better estimates of position. This is consistent with our results in Figure 2f, where we find that grid cell population activity becomes more correlated with the addition of landmarks. Similarly for the random rearrangement track, anticipatory licking and slowing remained high across blocks, indicating sustained engagement with the task. In addition to anticipatory licking and slowing, we also have a measure of accuracy for the random rearrangement track. Here, accuracy refers to the proportion of trials where the animal successfully requested a reward within the reward zone before an automatic reward was given. More specifically, the 20 cm reward zone is divided in two halves of 10 cm each. If an animal licks within the first half of the reward zone, they can trigger a reward. If they fail to do so, they will receive an automatic reward 10 cm from the start of the reward zone. Only one reward is given per trial. Similar to anticipatory licking and slowing, accuracy remains high across all 12 blocks, indicating reliable engagement throughout the task.

Finally, we would note that here, we are considering a VR feature a "landmark" if it is a distinct visual feature that could be used as a point of reference across more than one trial. We would note that whether the animals consider these as immutable landmarks (such as a building wall that never moves) or less reliable landmarks (such as the location of a trash bin that moves on collection day) is a question that falls outside the scope of the current work. Although we agree that this is an interesting question, a different set of experiments would be required to parametrically explore the full scope of how the predictability, certainty or valence of landmark input controls the bump trajectory.

3. I do not follow the logic in 181-221 (Figure 3). It seems as though the authors are saying that, because plasticity in the fly system can take several minutes, then any stable change in the grid cell modules in

mice that is faster than this must not require plasticity? There are many examples of rapid plasticity mechanisms in cortex and hippocampal systems, including the BTSP the authors cite, which seems to undermine this logic, if this is in fact what the authors intended. Additionally, since it is well known that maps form in the hippocampal system in single trials (for example, many place cells are present on the first traversal of a new environment) and grid cell rapid remapping has also been described (Fyhn 2007, Yoon K et al. 2013, Barry et al. 2012), then it is unclear what is new here. What is the novelty of the “one-shot” map finding?

We thank the reviewer for raising these important points. First, in terms of the logic underlying Figure 3, we have added a new schematic to Figure 2 that we hope better illustrates the two hypotheses we aim to test in Figure 3: that in novel environments or changing landmark conditions there is rapid plasticity between landmark inputs and the grid cell network (akin to what is seen in the fly) *or* that there is fixed connectivity between landmark inputs and the grid cell network (an idea more consistent with our data). Second, we completely agree that remapping has been observed and carefully considered in important prior works. We have made significant changes to the results (particularly in reference to Figure 1) that we hope better capture how our results are consistent with prior work. Finally, we have made changes to the abstract, introduction, results and discussion that we hope better frame the novelty of our findings, which we consider to be: 1) establishment of a method for tracking the moment-by-moment (i.e. real-time) state of the grid cell attractor network, a method we feel is now more rigorously established with the inclusion of data from mice that performed both freely-moving random foraging in 2D environments and 1D navigation in complete darkness; 2) providing a simple yet quantitative predictive framework for understanding the precise dynamics by which external landmark inputs and internal grid cell position coordinates interact - we believe the advances in our understanding of this framework can also broadly inform our understanding of how external sensory input interfaces with internal neural activity patterns to drive behavior; 3) insight regarding a potentially broader neural principle - that by allocating fixed and plastic connectivity across different networks, the brain can solve problems requiring both rapidity and representational accuracy.

Also related to the logic and conclusions of 181-221 (Figure 3), I do not see evidence that the distorted bump trajectories are caused by the landmarks “tugging” on the bump. I see evidence (Figure 3C,D) that the bump trajectories are distorted, but not necessarily “tugged” by the landmarks. The predictive model analysis results related to this point (3H) do not appear very strong. Could the “distortions” around landmarks just be the animal sometimes paying attention to the cues (when they are close) and sometimes not (see point above)?

The reviewer makes a set of important points. First, to provide a concrete example of what we mean by landmarks “tugging” the bump of activity, we show in Reviewer Fig. 4 the firing patterns of grid cells recorded during a behaviorally engaged session of the tower shift task. This figure reveals that local manipulations to the VR environment generate local distortions in the grid cell firing patterns, compressing or expanding the distance between firing fields. At a population level, this corresponds to the bump of activity being “tugged” forwards or backwards depending on the positions of the landmarks (we quantify the population level distortions in Fig. 4b,d, and include more examples of single-cell firing fields in Fig. 4c). We have edited the text extensively to try and better communicate what we mean by ‘tugging’. In the results section, we now describe our results as below:

We refer to this ‘tugging’ as ‘weak pinning’, given that the influence of visual landmarks was strong enough to entrain the bump to periodic trajectories (Fig. 2e, Extended Data Fig. 7) but not strong enough to fully ‘pin’ the bump to a particular position on the neural sheet each time the animal passed the corresponding visual landmark across the 12 environments (Fig. 3e-g).

Second, in relation to the animal paying attention or not on certain blocks, we now include behavior from the random rearrangement track (new Figure 3h, new Extended Data Fig. 1h-l) that illustrates how mice slow down and lick in the appropriate reward location across different landmark re-arrangements, which points to the mouse as engaged in the task (see also our response to comment 2, above). In addition, in the examples of firing field distortions during the tower-shift task (Reviewer Figure 4, Fig. 4b-d) we know the animal is also engaged across trials because it successfully licked in the hidden reward location within each block. This supports the idea that ‘tugging’ or ‘distortions’ are not reflective of the animal no longer paying attention to the landmarks or features of the environment.

Reviewer Figure 4. Visual landmarks distort grid cell firing patterns. (left, center) Two grid cells recorded during the tower shift task, each column is one cell. Colored bars indicate the location of landmarks (top and dotted lines). Note that when some towers are shifted, only those fields near the shifted towers move, while the fields near a stationary tower remain in place. The result is the distance between grid fields is compressed or expanded. (right) Cartoon of how the 2D maps might look under such a manipulation.

Finally, in relation to the model predictivity not appearing very strong, we wish to emphasize that predicting the firing of grid cells in a novel environment *before the animal enters that environment* is a challenging task for several reasons described below. We are careful to include strong baselines, and while the absolute correlations are not very high, we believe the fact that our correlations exceed the baselines is a surprising and novel result which demonstrates evidence of a structure underlying grid cell distortions which can be captured by our simple model.

1. As described in the text, the trajectory of the bump (and thus the subsequent pattern of activity a given neuron exhibits) is a function of both the influence of landmark inputs (which we refer to as ‘tugging’ or ‘weak pinning’) and the history of the bump trajectory. Interestingly, our model shows the same history dependence, since within a given environment there exist multiple stable limit cycles (trajectories) the bump can take. Which trajectory the bump takes depends sensitively on the initial conditions. When we

sample multiple initial conditions and take the best fitting limit cycle, we find noticeably higher average correlations (Reviewer Figure 5).

2. In many heldout blocks, some grid cells exhibit low firing or unstable firing patterns with no clear firing fields. Interestingly, our model often reproduces these unstable firing patterns (see Extended Data Fig. 8), likely due to the bump trajectory not consistently passing over that particular grid cell in that environment. However, if both the true and predicted firing patterns are unstable within a held-out block, the correlation between their tuning curves will be low in that held-out block. For this reason, in addition to Fig. 3f, which shows correlation *averaged* across held-out blocks, we include Fig. 3g, which shows correlation within each held-out block. We see that predictivity is higher in blocks that are more stable.
3. We are using a simple model, in which the landmark input ‘pins’ to a single phase on the grid cell neural sheet. It is very possible, as we describe in the discussion, that landmark inputs pin to more than a single phase or show connectivity dynamics not captured by the current model. Even so, we find it compelling that a very simple model can still predict the grid cell firing pattern for an arbitrary arrangement of landmarks to a very high degree when there is strong spatial stability (Fig. 3g).

Reviewer Figure 5. We repeat the analysis in Fig 3f, Extended Data Figure 8b, but in order to more fully sample the space of stable limit cycles, we initialize from 20 random initial conditions per heldout block, and keep the limit cycle which best predicts neural activity. As in Fig. 8b, the ablation consists of scrambling landmark locations. Note that for both ablation and model we sample 20 random initial conditions and take the best one for each held-out block. The model achieves significantly higher predictivity (KS test, $p < 1e-5$, animal $n = 3$, session $n = 3$, module $n = 3$, cell $n = 91$).

4. The data for “flexible adaptive behavior” do not strongly support the conclusions that are drawn (line 248). There is minimal support for MEC grid cells endowing this system with “powerful computational properties” (lines 21-22). Certainly grid cells may be a critical part of the system, but the behavioral adaptation and flexibility is likely occurring somewhere else. If anything, the data presented is consistent with this view. The extensive computational and theoretical literature on how this could be implemented is largely ignored. Whittington et al – Nat Neuro 2022, for example, provides a cohesive way of considering grid cells in this system. The model of downstream BTSP doesn’t add much insight and if anything suggests that the main focus of the present manuscript is not where the interesting computation related to behavior is happening.

We fully agree with the reviewer that there is a wide body of literature on downstream learning mechanisms which would be compatible with such flexible adaptation. We chose to highlight BTSP as a candidate downstream learning rule because of intriguing recent experimental evidence of MEC-directed learning in hippocampus on rapid timescales rapid enough to support the flexible learning we observe in our experiments [Bowler & Losonczy, Neuron, 2023; Grienberger, Magee, Nature, 2022; Bittner et al., Science 2017]. However,

it is certainly true that other downstream learning mechanisms are possible, including Hebbian learning in an associative network, the mechanism described in [Whittington et al, Nat Neuro 2022] along with many works before it [Manns & Eichenbaum, Hippocampus, 2006]. We have expanded our discussion of downstream learning circuits to include these other mechanisms.

However, neither do we feel that a cohesive understanding of flexible adaptation in the MEC-hippocampal circuit has already been achieved. We discussed our experimental results with the authors of [Whittington et al., Cell, 2020]. A central point in their work is the advantage of *factorized* codes (i.e. where the grid cell code is undistorted by external cues). That work, along with [Whittington et al., Cell, 2020], describes a learning mechanism which will "disentangle" the grid cell code from external cues, similar to the prediction in Fig. 4b (*left*). The authors were surprised and intrigued to learn that our experimental results were more consistent with Fig. 4b (*right*) – that is, that flexible behavior occurs in the "entangled" (distorted, in our language) regime, rather than the "factorized" regime – and we discussed modifications of their model which could account for this observation. Thus, we feel that this experimental observation may be highly valuable for future theoretical models of adaptive learning in the hippocampal-entorhinal circuit.

Finally, we would note that we have extensively revised the text itself, and we no longer use some of the language referred to here (e.g. 'powerful computational properties'). We hope the way in which the text has been revised better communicates the work and the conclusions that can be made from the work.

Extended Data Fig. 7a-d. Spatial coding properties of non-grid cells. In the random environment track, we identified non-grid spatial cells as cells that 1) showed no periodic structure in the dark and 2) had an average spatial Pearson correlation >0.2 between the first and second half of each VR block (Methods). We found that many of these cells were driven by linear combinations of environmental features. **a.** To quantify this effect, we defined template maps (colored heatmaps) for each VR landmark as gaussian firing fields with a width of 50 cm, centered at each landmark, similar to the approach taken in [Kinkhabwala et al., Elife, 2020]. We performed cross-validated linear regression with weights $w_1 \dots w_5$, holding out one VR block out at a time, to predict the population activity of non-grid spatial cells in the heldout environment. Example predicted firing patterns across held-out environments for one cell are

shown at right (Sim.) and the true firing patterns at far right (Cell 1). **b.** We identified 'cue cells' as cells for which the predictivity of the linear regressor exceeded the 95th percentile of a shuffled distribution consisting of training linear regressors on randomly scrambled landmark positions. 56% of non-grid spatial cells met the definition of cue cell, and the distribution of predictivities was much higher than a random shuffle ($p < 1e-5$, cell $n = 6,332$, animal $n = 6$, session $n = 32$). **c.** Grid cell firing patterns, in contrast, were not well predicted by linear combinations of sensory stimuli ($p < 1e-5$, cell $n = 6,332$, animal $n = 6$, session $n = 32$). **d.** In addition to sensory stimuli, we included the grid cell attractor states (2 coordinates per module) as features in our regression model. We found that this significantly improved predictivity in heldout environments, indicating the possibility that non-grid spatial cells conjunctively code for sensory stimuli and the state of the grid cell attractor ($p < 1e-5$, cell $n = 6,332$, animal $n = 6$, session $n = 32$).

5. The premise/motivation presented in the manuscript oversimplifies what is known by trying to set-up the grid cell network as the primary and dominant source of spatial information during behavior. The contribution of the remaining ~75% of MEC cells, most HPC cells and all LEC cells, for example, are minimally considered.

We apologize that the language used in the manuscript overemphasized the role of the grid cell network. We chose to focus on grid cells for this work as they are believed to generate an internal map of space and underlie path integration. We sought to characterize precisely how landmarks in novel and changing conditions are incorporated into this map of space on near instantaneous timescales. We also recognize the reviewer's concern and have performed new analyses to investigate the spatial coding of non-grid cells in MEC, and have made several changes we hope better explain both why we focus on grid cells and the remaining medial entorhinal neurons. First, we have significantly revised the text (abstract, introduction and discussion) in a manner we hope better communicates our central findings and does not overstate or mis-state our interpretations. Second, we have performed new analyses on non-grid cells, which has been added as new panels in Extended Data Figure 7 (panels a-d) and new Extended Data Figure 10 (Shown below for ease of reference).

1. We investigated the coding properties of non-grid spatial cells in MEC. We found that the majority of these cells were driven by linear combinations of sensory stimuli (though few cells were correlated with a *single* VR feature, Extended Data Fig. 7e-g). Taking advantage of the unique experimental design of the random environment track, we were able to learn each non-grid spatial cell's preferred combination of sensory stimuli, and thereby predict their firing patterns in held-out environments significantly better than chance (Extended Data Fig. 7b). Interestingly, grid cell firing patterns were *not* well predicted by linear combinations of sensory stimuli (Extended Data Fig. 7c), and indeed were better predicted by our dynamical model (Fig. 3). Moreover, we found evidence that non-grid spatial cells' firing patterns may depend on both sensory stimuli and the instantaneous attractor state of the grid cells (Extended Data Fig. 7d).
2. We extended our analysis decoding position from the population activity of grid cells in the tower-shift task (Fig. 4) to include the population activity of *all* MEC cells, not just grid cells (Extended Data Fig. 10). We found that non-grid cells did not provide new spatial information, not already captured by the grid cells (Extended Data Fig. 10a), and that non-grid spatial cells shifted with the VR landmarks similar to the grid cells (Extended Data Fig. 10b), yielding the same distortions in decoded position (Extended Data Fig. 10c,d).

This analysis reveals that our central conclusions regarding the ‘fixed’ nature (i.e. fixed relationship between landmark inputs and the entorhinal neural network) of entorhinal dynamics do not change when we include all medial entorhinal neurons, not just grid cells.

Extended Data Fig. 10. Non-grid spatial cells in the tower shift task. **a.** Absent any visual cues, in the dark 20cm region preceding the reward zone (pink, bottom), grid cells provide more faithful spatial information than non-grid spatial cells. Grid cells are identified based on their firing properties in the dark; non-grid spatial cells are identified within each block as those cells whose tuning curves in the first half of the block (20 trials) have a spatial correlation greater than 0.2 with their tuning curve in the second half of the block. Linear circular decoders are trained using equal numbers of grid and non-grid spatial cells (chosen as the minimum of the two) to predict all locations along the track, and are evaluated only on the 20cm preceding the reward zone, where the animal is in the dark. Decoding error is significantly smaller for a decoder trained on grid cell firing, across blocks, animals, and sessions (Wilcoxon signed-rank test, $p < 1e-4$, animal $n = 4$, session $n = 6$, block $n = 36$, cell $n = 318$). **b.** Example non-grid spatial cell rasters in the tower-shift task (animal $n = 4$). These cells exhibit firing fields closely tied to landmarks. **c.** Fig. 4b repeated with a decoder trained on all MEC cells, rather than just putative grid cells. **d.** Fig. 4d repeated with a decoder trained on all MEC cells, rather than just putative grid cells.

Minor points:

128 -129: “Together, these results indicate that even in the absence of visual landmarks, the grid cell population is well described by low-dimensional attractor dynamics.” This has been already shown in RJ Gardner 2022 et al. where they show in an unsupervised method that the topology of torus preserved during sleep (no sensory cues). The authors’ method is supervised and assumes in advance the topology of the neural space. This is weaker and not sufficiently novel.

We agree with the reviewer and apologize, as this was not intended to be an indication of novelty but rather an indication that the method we use to identify grid cells in 1D does indeed identify cells with key, known cardinal properties of grid cells. We have amended this part of the text extensively, as Figure 1 no longer includes some of the replication results (they are now in Extended Data Figure 2) and we have included 1D to 2D comparisons in Figure 1. We have also included text to try and emphasize that the novelty of the method lies in 1) being able to examine grid cells in 1D and, 2) in extracting the instantaneous attractor state of the network.

137-139: “In recordings that captured the activity of more than one module of grid cells, we observed that activity bumps from different modules were tightly coupled and drifted in concert, taking the same overall trajectories.” This was already shown in Waaga, T. et al. 2022. The authors put citation without explicitly saying that the result is consistent with this already published paper.

We thank the reviewer for drawing our attention to this. As noted above, we have extensively edited the text pertaining to Figure 1 (including this sentence) and hope it better reflects the prior work, how our findings are consistent with this prior work and the novelty of the current work.

177-179: “This provides a remarkably simple understanding of remapping as just taking a new trajectory on a toroidal attractor, thereby demystifying the complex re-organization of individual grid cells during a remapping event, a process that can appear random aside from the preserved correlations between grid cells.” It is already known that at the single-cell level, the phase between cells is preserved during remapping, which corresponds to a different trajectory on the torus at the population level. The authors need to explain how the two findings differ. A citation of Fyhn et al 2007 is missing.

We have added a citation to Fyhn et al., 2007 and how our observations are consistent with this prior observation. We would also note that in our work, we offer a more explicit observation of this phenomenon at the population level and that this is observed rapidly after relatively small changes to landmark features.

219-221: “We found that grid cells developed different maps in the two repeated environments, despite them being visually identical.” This result is inconsistent with papers of 2d open field environments where mice are switched to a novel environment and then returned to the original environment (A -> B -> A’). Grid cells were found to be stable between A and A' (Fyhn et al 2007). Could the differences observed in the present work potentially point to low attention to the VR environments (as noted above)?

First, we would note that we only observed this in a subset of experiments (~50%), which we have ensured is clearly indicated now in the results.

Second, regarding the concern that low attention to the VR environments may contribute to the observation of different maps in two repeated environments, we have included new behavioral analyses, as mentioned above. Our quantifications of anticipatory licking and slowing suggest that our animals continue to engage with the build-up track even during the latter blocks (See Extended Data Fig. 1). To directly test whether there’s a difference in task engagement between the two visually identical blocks, we compared anticipatory licking and slowing between blocks 6 and 8 (Tower 2 and Tower 2b conditions). We found that between the two blocks, task engagement was not significantly different by measures of anticipatory licking (paired t-test, $p = 0.66$, $n = 19$ sessions) and anticipatory slowing (paired t-test, $p = 0.30$, $n = 19$ sessions).

Third, regarding the prior Fyhn et al., 2007 work. In the Fyhn et al., paper, tetrodes were lowered over the course of 5-10 days before data collection, indicating that animals had at a minimum, 5-10 days of experience in the open-field environment before data collection. This timeline would be consistent with a role for experience in stabilizing grid maps, a perspective highly consistent with our work. In addition, remapping studies in work such as Fyhn et al., 2007 had some other notable differences - in that environments were explored for long (tens of minutes) periods of time over multiple days. In contrast, in the current manuscript, environments are often experienced for only a few minutes and over many fewer days (often we are considering the very first exposure to a given environment). Thus, our work is focused on *rapid* map dynamics, rather than longer term map dynamics, a question interesting in its own right but beyond the scope of the current manuscript.

Referee #2 (Remarks to the Author):

This is a very interesting paper. The authors have performed some excellent experiments, analysis, modelling, and theory work. The final result is a novel idea about how the entorhinal-hippocampal area functions. The manuscript is mostly clear, well-written and well-presented. This paper should have a substantial impact on the field. I do have a few clarifications/questions that are listed below.

We thank the reviewer for their constructive feedback on the paper. Below, we provide new analyses that we hope will clarify the overall results of the manuscript.

1) The most obvious issue is that there is some serious data selection here (ie which neurons are to be included for further analysis). We go from 68k active neurons to 15k grid cells and then to hundreds and tens of co-modular grid cells (grid cells from one power spectra module) in the construction of network (bump) activity. Do the authors have a principled reason for not including (the majority of?) neurons that did not show grid cell like activity? How does inclusion of non-grid cells into the population activity pattern alter that activity?

We completely agree with the reviewer that our analysis centers on grid cells, as our work focuses on understanding how novel and changing landmarks are incorporated into the grid cell attractor network. We also agree with the reviewer that understanding how non-grid cells contribute to the spatial code in MEC is an important question, and we have taken steps toward understanding it with new analyses in our response to reviewers.

First, we wish to clarify that we exclude non-grid cells from our neural sheet analysis because these cells, as far as we can tell, do not participate in the toroidal attractor dynamics in the dark; that is, they exhibit no underlying periodic structure. We demonstrate this in Reviewer Figure 6, left, where we show PSDs for *all* cells from an example recording (the same session as in Fig. 1) in complete darkness. We see clear periodic structure (vertical bands) for grid cells across four modules, but no clear structure for non-grid cells. This not

Reviewer Figure 6. Spectral properties of grid and non-grid cells in the dark. We plot power spectra for the same session as in Fig. 1c, but including all simultaneously recorded MEC neurons, including non-grid cells (labeled by gray band at right), which show no clear periodic structure. Power spectra for each neuron color coded for maximum (white) and minimum (black) power in each frequency, sorted dorsal (top) to ventral (bottom). only prevents us from identifying preferred phases for the non-grid cells, and therefore sorting them onto the neural sheet, but also suggests that they do not participate in the attractor dynamics in the absence of visual cues.

Second, the reviewer raises the important questions: what does the population activity of non-grid cells look like, and how does it relate to the population activity of grid cells? We address these questions through new analyses

and two new figures, Extended Data Figs. 7 and 10, which we describe below (reproduced from response to reviewer 1):

1. We investigated the coding properties of non-grid spatial cells in MEC. We found that the majority of these cells were driven by linear combinations of sensory stimuli (though few cells were correlated with a *single* VR feature, Extended Data Fig. 7e-g). Taking advantage of the unique experimental design of the random environments task, we were able to learn each non-grid spatial cell's preferred combination of sensory stimuli, and thereby predict their firing patterns in held-out environments significantly better than chance (Extended Data Fig. 7b). Interestingly, grid cell firing patterns were *not* well predicted by linear combinations of sensory stimuli (Extended Data Fig. 7c), and indeed were better predicted by our dynamical model (Fig. 3). Moreover, we found evidence that non-grid spatial cells' firing patterns may depend on both sensory stimuli and the instantaneous attractor state of the grid cells (Extended Data Fig. 7d).
2. We extended our analysis decoding position from the population activity of grid cells in the tower-shift task (Fig. 4) to include the population activity of *all* MEC cells, not just grid cells (Extended Data Fig. 10). We found that non-grid cells did not provide new spatial information, not already captured by the grid cells (Extended Data Fig. 10a), and that non-grid spatial cells shifted with the VR landmarks similar to the grid cells (Extended Data Fig. 10b), yielding the same distortions in decoded position (Extended Data Fig. 10cd).

For that matter, what happens when the activity of multiple grid-cell modules are combined? I realize the authors are trying to do something analogous to dimensionality reduction to describe the activity of co-modular grid cells. But they are also trying to make some functional conclusions relating to a role for synaptic plasticity in network remapping of novel environments. Can plasticity among the non-grid cells be involved in the learning of new maps for altered environments?

This is an important comment. First, we wish to clarify that in our decoder analyses related to learning in novel environments, we always decode from the *full population of grid cells*, not just a single module (Fig. 4). Second, we absolutely agree that one possibility is that non-grid cell activity could be involved in learning of new maps of altered environments. To consider this possibility, we examined the non-grid spatial cells in the tower-shift task (new Extended Data Figure 10). We found that these cells shifted with the VR landmarks in much the same way grid cells did, and never recovered from these shifts (new Extended Data Figure 10b). Furthermore, when we included these cells in our decoding analysis, we observed the same distortions as we observed when decoding from grid cells (new Extended Data Figure 10c,d), and no evidence of recovery from these distortions. These analyses revealed that non-grid cells contained no new spatial information beyond that encoded in the grid cell population activity.

Are any new grid cells formed during the novel environments that perhaps did not meet the grid cell criteria because of untuned activity before the environmental change?

This is an interesting point. To investigate this, we identified grid cells separately in the dark block at the start of each session (block A) and again during the dark block at the end of each session (block B). An average of ~50 cells were identified as grid cells in each block. Of those, 8.6 +/- 2.0 cells were classified as grid cells in block A but not block B, and 10.3 +/- 2.5 cells were classified as grid cells in block B but not block A (Reviewer Figure 7, shown below). While the number of cells at the end of the session was on average greater, the difference was not statistically significant, and cells identified as grid cells in one block but not the other are likely due to

measurement noise, the grid cell being close to the edge of the neural sheet or transient instability in grid cell firing patterns. Thus, overall, we do not see that there is strong evidence that new grid cells are recruited during the novel environment experience.

Reviewer Figure 7. Grid cells identified in the dark block at the beginning but not the end of the session (blue), and grid cells identified in the dark block at the end of the session but not the beginning (green). Bar height represents average across sessions, and error bars represent standard error on the mean (session $N = 14$, mouse $N = 4$, cell $N = 2,424$).

Also given all of the neuronal activity from non-grid cells in the EC, how could only the activity of grid cells (particularly one module) be read out downstream? Some additional discussion concerning these issues would be helpful.

We agree with the reviewer that downstream circuits likely read out from all of MEC, not just grid cells. In the new Extended Data Figure 10c,d, we repeat the decoder analysis in Fig. 4b,d, using all recorded cells in MEC. We find that a decoder trained on all of MEC exhibits the same behavior as a decoder trained on grid cells only. Additionally, in the new Extended Data Figure 10a, we train a decoder on only the non-grid spatial cells, and evaluate its ability to predict the animal's position in the dark region just before the hidden reward zone. We find that the decoder trained on non-grid spatial cells performs worse than a decoder trained on grid cells, suggesting that in the behaviorally relevant region, absent any visual cues, the grid cells offer more reliable spatial information than the non-grid spatial cells.

2) The fixed and weak landmark to grid map pinning system is an interesting idea. But the activity generated by the model did not strongly correlate with the actual activity recorded from the neurons used to produce the model. There is a large amount of overlap between the distribution of correlations for random conditions and even the best model (fig 3i; plots in Sfig 6). I couldn't find any discussion of the idea that perhaps a continuum exists with different attractor networks spanning the range from purely idiothetic to strong landmark pinning. Did the authors find any evidence that some modules appeared to be mostly idiothetic and others strongly pinned to landmarks? Perhaps the result presented reflects an average "level of pinning" that spans a large range? Some additional discussion of these issues and the parameter, α , could help here. Do the authors consider each grid cell module to inhabit a single, unique attractor network?

The reviewer raises a number of interesting points. First, we wish to re-emphasize the challenging nature of predicting the firing of grid cells in a novel environment *before the animal enters that environment*. We reproduce below three points from our response to reviewer 1 on this topic. We are careful to include strong baselines, and while the absolute correlations are not always high, we believe the fact that our correlations exceed the baselines

is a surprising and novel result which demonstrates evidence of a structure underlying grid cell distortions which can be captured by our simple model.

1. As described in the text, the trajectory of the bump (and thus the subsequent pattern of activity a given neuron exhibits) is a function of both the influence of landmark inputs (which we refer to as 'tugging' or 'weak pinning') and the history of the bump trajectory. Interestingly, our model shows the same history dependence, since within a given environment there exist multiple stable limit cycles (trajectories) the bump can take. Which trajectory the bump takes depends sensitively on the initial conditions. When we sample multiple initial conditions and take the best fitting limit cycle, we find noticeably higher average correlations (Reviewer Fig 5).
2. In many heldout blocks, some grid cells exhibit low firing or unstable firing patterns with no clear firing fields. Interestingly, our model often reproduces these unstable firing patterns (see Extended Data Fig. 8), likely due to the bump trajectory not consistently passing over that particular grid cell in that environment. However, if both the true and predicted firing patterns are unstable within a held-out block, the correlation between their tuning curves will be low in that held-out block. For this reason, in addition to Fig. 3f, which shows correlation *averaged* across held-out blocks, we include Fig. 3g, which shows correlation within each heldout block. We see that predictivity is higher in blocks that are more stable.
3. We are using a simple model in which the landmark input 'pins' to a single phase on the grid cell neural sheet. It is very possible, as we describe in the discussion, that landmark inputs pin to more than a single phase or show connectivity dynamics not captured by the current model. Even so, we find it compelling that a very simple model can still predict the grid cell firing pattern for an arbitrary arrangement of landmarks to a very high degree when there is strong spatial stability (Fig. 3g).

Second, we find the idea the reviewer suggests regarding different modules having different average 'levels of pinning' highly intriguing. For the build up track, we had a number of recordings that spanned modules. Thus, to consider this question we estimated the pinning strength by considering the average stability of the grid code within a given block (e.g. 2 tower block or 3 tower block). If the pinning strength is very weak, the correlation should be lower, as the trajectory the bump of activity takes on any given trial will vary slightly. We observed a slight trend in which larger grid modules showed lower correlation values, which may indicate that pinning strength decreases along the dorsal to ventral axis of MEC. We have included this panel in Extended Data Figure 4 and included discussion text on how different modules may show different levels of pinning. The text now reads:

Another possibility is that grid cells in different modules are more or less prone to distortions (Extended Data Fig. 4e). Prior work has shown a dorsal to ventral gradient in the degree to which grid cell firing patterns distort after a change in environmental geometry. In addition, there are dorsal to ventral MEC gradients in gene expression, and whether such gradients could play a role in how strongly different grid cells modules are influenced by landmark inputs remains unknown.

Extended Data Figure 4e: Grid scale vs estimated pinning strength in the build-up track. Pinning strength is measured for a given module as the average stability of the grid cell code within VR blocks (Pearson r). We observe a small effect where estimated pinning strength is weaker for larger modules ($p=0.005$, module $N = 49$, animal $N = 4$, session $N = 14$).

3) Finally, the idea that neuronal activity in a grid cell module is constrained to move around in a defined and continuous space when a mouse is moving in an environment is interesting. But what happens to the bump of activity within the attractor when the animal stops running? Does it stay at the same location in activity space (ie some sort of persistent activity)? Does it move around in the same activity space or a different space? If it moves around, how does it find its way back to the appropriate location in activity space? I suppose this is mostly relevant for when there are rewards present in the task. Does the animal stop and consume the reward for some variable amount of time? What happens to the neuronal activity during this standing period? If the activity here is not simply persistent can the weak pinning model handle the standing activity as well as the running?

The reviewer raises a very interesting point which we have investigated through further analyses. These new analyses are described in the results section and shown in new Extended Data Figure 6. In Extended Figure 6, we plot both the location of the animal on the circular track (bottom, black), as well as the 2D location of the bump on the attractor (middle and top, orange and blue), for the duration of an entire build up track session (which includes both dark and visual landmark feature blocks), spanning a little over an hour. For most of the session the animal is moving quickly, however we observed a few periods spanning many seconds-minutes over which the animal is stationary. We observe that the bump also appears to stop with the animal during these periods, with a small amount of diffusion around a central tendency. This diffusion spans roughly one or two radians, which given the frequency of this module corresponds to a spatial error of roughly 20-30cm. It is not clear whether this diffusion is due to noise in our ability to estimate the bump's location (i.e. the bump would be exactly stationary if we had perfect measurement fidelity), or represents a more fundamental diffusion in the neural dynamics [Burak & Fiete, Plos Comp. Bio., 2009]. In panel b we zoom in on a period of 100 seconds over which the animal is engaged, and find that the speed of the bump on the sheet closely tracks variability in the animal's running speed on the treadmill. In panel c we visualize the variability in the inferred bump location on the sheet during a period where the animal is stationary (left) and a period where the animal is moving (right).

Extended Data Fig. 6: Bump trajectory and animal behavior on an example buildup track session (~1 hr). **a.** top, middle, Inferred 2D bump coordinates on the toroidal attractor, with both phases plotted as a function of time (blue and orange panels) over the course of the entire session. bottom, Animal's position on the periodic track throughout the entire duration of the experiment. We observe a few periods of many seconds-minutes where the animal is stationary on the treadmill. During these periods, the inferred position of the activity bump on the attractor is also roughly stationary, though drifts slightly (within one revolution). Note that this drift appears magnified, since one lap of the animal on the track typically corresponds to 3-4 revolutions of the activity bump around the attractor. **b** Zoomed in view of a stationary period of ~90 seconds. We compare the (z-scored) speed of the animal on the treadmill (gray line) to the (z-scored) speed of the activity bump on the attractor (red line). The activity bump's speed closely tracks the animal's speed on the treadmill. **c.** The drift in the inferred position of the activity bump on the attractor during a period when the animal is stationary (left), compared to a period where the animal is running (right).

4) Please add at least one additional figure for ease of visualization.

We agree with the reviewer that the manuscript needed to be improved in terms of clarity. To this end, we have modified Figure 1 to include open field 2D and VR 1D recordings, which we hope will provide more intuition for the underlying analyses we use to examine the moment-by-moment state of the grid cell attractor network. In addition, we have added a new schematic to Figure 2j-l (included below for ease of reference), which illustrates the two hypotheses for how landmark inputs project to the grid cell network (a fixed landmark to grid cell relationship or a plastic landmark to grid cell relationship).

Fig. 2: j-l. Cartoon models of the influence of landmarks on the grid cell attractor state, represented as a bump of activity (red) on a sheet of grid cells (gray circles), which follows the red trajectory as the animal (bottom) traverses the VR track. **j**, A single landmark (green) weakly pins the attractor bump, slightly deflecting its trajectory when the animal approaches the landmark. **k**, The addition of a new landmark (blue) can deflect the bump onto a new trajectory, as seen in **i**. **l**, Plasticity could hypothetically allow landmarks to update their pinning phases on the neural sheet, to provide a consistent relationship between landmarks in the environment and attractor states.

Referee #3 (Remarks to the Author):

This paper presents remarkable experiments analyzed in clever ways. It is hard to quibble about 68,484 recorded cells, although I will manage to do that below. As I understand it, the main points are: 1) The neural sheet that forms the basis of the attractor model of grid cells remains constant, at least to a good approximation, across all the manipulations reported. 2) In the absence of external landmarks, the angle that activity traverses across the neural sheet diffuses. 3) In the presence of landmarks, the angle of the trajectory along the neural sheet rotates in such a way that grid cell response are periodic over the length of a lap. 3) Landmarks are mapped to fixed points on the neural sheet. 4) Landmarks distort the path of activity across the neural sheet, resulting in curved trajectories. 5) Downstream learning can compensate for these distorted trajectories using BTSP. The paper is difficult to follow for reasons given below, and it has a number of puzzling aspects.

1) In the description above, I have used neural sheet in the way that the authors do, but they also use the term grid cell map. I am not sure what that is, and how it relates to the neural sheet. It would be good to include a clear definition of these terms and to use them consistently.

We thank the reviewer for raising this point. By 'grid map', we meant the collective activity of all the grid cells in a given module. However, we recognize that we used this term inconsistently and that it may add unnecessary confusion. Thus, we have edited the paper extensively to try and limit the terminology. 'Grid map' has been removed and we have tried to only use a few key terms that are well defined (e.g. neural sheet, activity bump).

2) The grid cell system is built to describe motion in 2 dimensions. The authors act as if, on this linear track, the animal is representing the world as 1D and that the curvature of the path on the neural sheet is a distortion of a 1D world. What if it is a 2D trajectory? Shouldn't this possibility at least be considered? As I understand it, the introduction of landmarks induces small curvature in the neural sheet path that straighten out as more landmarks are added and the grid response become periodic. Couldn't this be interpreted literally as the animal thinking initially that it is not moving in a straight line, perhaps due to an imbalance in the placement of the landmarks, and later realizing that it is?

We thank the reviewer for this important clarification, and we believe our interpretation of the results is broadly consistent with the reviewer's. We first note that our findings related to Figures 1 and 2 suggest that, despite the environment being 1D, the neural representation of the environment is indeed better described by a 2D trajectory on a toroidal attractor than a 1D trajectory on a ring (i.e. requires 2 coordinates: the two coordinates on the neural sheet). Though the bump takes straight-line trajectories on the neural sheet over single trials, the angle of these trajectories drifts over many trials. One interpretation of this observation is that the animal may not realize it is traveling in a straight line, especially in the absence of visual cues. Once visual cues are added to the track, the bump trajectory becomes straighter over long horizons (Fig. 2d), consistent with these cues providing more sensory feedback about the shape of the straight corridor the animal traverses.

As the reviewer notes, the addition of visual landmarks also introduces local distortions, which manifest both as stretching and compression of the activity bump's straight-line trajectory (anisometry, in Fig. 3d, Methods), and local curvature in the bump's trajectory (geodesic curvature, Fig. 3c). We call these transformations distortions because the bump's position on the neural sheet, which typically tracks the animal's position faithfully, becomes dissociated from the animal's position on the track in these regions of local distortion. Whether the animal "thinks" it is at the position represented by the bump on the attractor, or "thinks" it is at its true location on the VR track is an interesting question with behavioral consequences, which motivates our tower-shift experiments in Fig. 4. The fact that, for one to a few trials, the animal licks late when we shift the last tower to appear later on the track, and licks early when we shift the tower to appear early on the track, is consistent

with the reviewer's observation that the animal "thinks" it is taking the trajectory represented by the bump on the neural sheet. While our experiments do not conclusively reveal whether the animal "thinks" it is departing from a straight trajectory, this might be an interesting direction for future investigation.

3) The authors assume that the landmarks they use are totally accepted by the animals, but that their effect on the neural sheet trajectory is weak. They then provide a fairly long, and to me mystifying, discussion of why this is a good idea as a general policy. The trajectory presumably reflects the best estimate of where the animal thinks it is. Maybe it considers the artificial landmarks used in this experiment to be weak evidence of position and is weighing them appropriately. Would the effect be stronger if more convincing evidence of position was provided? What about textures on the track or odors? It seems to me that the idea of weak evidence weighting should be eliminated before such a strong case for the optimality of a weak effect is made.

The reviewer raises a set of interesting points. First, we believe that the animal is using the visual landmarks, as we have performed new behavioral analysis (See Extended Data Figure 1) that indicates that mice are engaged in the various tasks and lick in anticipation of the reward location even when landmarks are moving across blocks of trials. Second, we have significantly amended the text, which we hope better emphasizes our findings in a way that does not rely on the idea of 'weak pinning' as a good general policy. Finally, we completely agree that a rich sensory environment would be a highly interesting future direction for this work to go. However, we felt that adding somatosensory or olfactory cues to an already complex set of behavioral tasks was beyond the scope of the current manuscript and would be best considered in future work that could fully parameterize and explore the space of multi-sensory landmark features.

4) 68,484 cells is remarkable, but the paper is based on 15,000 of them. The others are dismissed as having little apparent spatial structure. Does this mean little apparent spatial modulation? I am not suggesting an analysis of the remaining cells, but perhaps they could be dismissed more informatively.

We thank the reviewer for this important concern, which we hope to clarify and address below. We identify putative grid cells as those cells which exhibit periodic spatial structure in the absence of visual cues. This intrinsic periodic structure is revealed by computing the power spectra of cells while the animal is running in the dark (see Reviewer Fig. 5). We find that some cells exhibit clear periodic structure, while others do not, and those that do fall neatly into discrete clusters (putative grid modules) of increasing spatial frequency.

The remaining non-grid cells certainly do exhibit spatial structure, but only in the presence of visual cues and rewards. In response to the reviewers' comments, we have added new analyses and two new figures to investigate the population coding of these non-grid spatial cells (Extended Data Figs. 7 and 10), which we describe in the comments reproduced below:

1. We investigated the coding properties of non-grid spatial cells in MEC. We found that the majority of these cells were driven by linear combinations of sensory stimuli (though few cells were correlated with a *single* VR feature, Extended Data Fig. 7e-g). Taking advantage of the unique experimental design of the random environments task, we were able to learn each non-grid spatial cell's preferred combination of sensory stimuli, and thereby predict their firing patterns in held-out environments significantly better than chance (Extended Data Fig. 7b). Interestingly, grid cell firing patterns were *not* well predicted by linear combinations of sensory stimuli (Extended Data Fig. 7c), and indeed were better predicted by our dynamical model (Fig. 3). Moreover, we found

evidence that non-grid spatial cells' firing patterns may depend on both sensory stimuli and the instantaneous attractor state of the grid cells (Extended Data Fig. 7d).

2. We extended our analysis decoding position from the population activity of grid cells in the tower-shift task (Fig. 4) to include the population activity of *all* MEC cells, not just grid cells (Extended Data Fig. 10). We found that non-grid cells did not provide new spatial information, not already captured by the grid cells (Extended Data Fig. 10a), and that non-grid spatial cells shifted with the VR landmarks similar to the grid cells (Extended Data Fig. 10b), yielding the same distortions in decoded position (Extended Data Fig. 10c,d).

5) The authors argue that the MEC can do one-shot learning without plasticity, which is a roundabout way of saying nothing is happening in the MEC. Listing this as a virtue seems far-fetched. Evidence of position is being weighed outside the MEC, and MEC activity is being used to read out the results of that weighing, using the structure of grid cells to decipher the signal. Isn't that a more reasonable statement of what is going on?

We agree with the reviewer and have re-structured the text significantly in a manner that we believe better communicates our take home conclusions. We would also note that we have now included text that references a recently proposed hypothesis - that distortions in the MEC could serve a useful purpose (e.g. they could signal to the hippocampus when something has changed about the environment). In addition, we believe something *is* happening in MEC, as muscimol injections only into the MEC impair behavior. Even so, we have significantly reduced language regarding the virtues/drawbacks of our observations regarding MEC coding and more straightforwardly report what our experiments reveal in terms of how the circuit operates.

5) A very nice result is the extraction of the diffusion constant for the angular drift. It is stated that this is consistent with other studies of animals getting lost in the dark, but a quantification of this comparison would be helpful.

The reviewer raises a very interesting point which we agree was under-explored in our original manuscript. To make a quantitative comparison, we turned to a study in which blindfolded humans were asked to walk in a straight line (Souman et al., *Curr. Bio.*, 2019). The study found that blindfolded human participants took meandering paths, often walking in circles. The distribution of changes in walking direction per second for 15 human participants are quantified in their Fig. 2e, reproduced below (Reviewer Figure 8). The authors write, "The fact that participants walked in circles instead of following a random zigzag path suggests that the veering from straight ahead was caused by a change in their subjective sense of straight ahead rather than by random noise in either sensory input or the motor output".

For comparison, we calculated the distribution of changes in inferred grid orientation per second for 5 mice navigating a 1D treadmill in the dark, across a total of 20 sessions (Reviewer Figure 8b). Similar to the authors of Souman et al., *Curr. Bio.*, 2019, we find no overall bias evident in the direction of orientation drift, and the magnitude of grid orientation changes per second is fairly comparable to the magnitude of walking direction changes per second. It is important to note that Reviewer Figure 8a is a behavioral measurement, while Reviewer Figure 8b is a measurement of the neural representation of heading by inferred grid orientation. However, if grid orientation represents the animal's "subjective sense of straight ahead", these experimental results suggest that mice would take similarly meandering, looping trajectories in the absence of visual cues.

a

[REDACTED]

b

Reviewer Figure 8. Accumulation of angular drift in humans and mice. **a.** [REDACTED] Reproduced from Souman et al., *Curr. Bio.*, 2019. Individual distributions of the change in walking direction for 15 blindfolded participants tasked with walking in a straight line. Participants walked without vision in five trials of 10 min each (MV, JS, PS, JH, SK, and KB) or in ten trials of 5 min each (other participants), with 1–2 min breaks with vision in between. Box-plot whiskers represent 1.5 times the interquartile range; red crosses indicate outliers. Participants are shown sorted according to their median direction change. **b.** Individual distributions of the change in inferred 2D grid orientation for 5 mice navigating the 1D treadmill in the dark, across a total of 20 sessions. Box-plot whiskers represent 1.5 times the interquartile range; red crosses indicate outliers. Mice are shown sorted according to their median direction change.

6) Although this is point 6, it introduces what was, for me, the most frustrating aspect of the paper. The distortions the authors are discussion are associated with the animal having more information about where it is, but in the paper they are presented as something that needs to be compensated for, rather than as something carrying more information. It is as if the animal is more confused in the presence of landmarks, but that downstream circuits can compensate for this confused representation. A title states that "One-shot MEC maps enable flexible adaptive behavior despite small distortions". Almost nothing in this title makes sense to me. Why is it one-shot - it is fast but is it really one-shot? How can the MEC enable something despite distorting it? The paper does not present the distortions as anything positive or show that they increase accuracy, yet they are the result of introducing landmarks. This is, to me, the most confusing aspect of the paper.

First, we agree with the reviewer that the discussion made it unclear whether the distortions are something to be compensated for or something that actually carries more information. We have significantly edited the abstract, introduction and discussion in a way we hope better communicates that our data shows the map is distorted but this has multiple interpretations (e.g. it may be useful in the sense that it can convey information about important changes to landmark features). We also hope this editing has made the text and results clearer. Second, we apologize that our reference to one-shot remapping was not clear. We base this, in part, on the illustration of remapping shown in Figure 2i. The very first trial in which the animal experiences the 3 tower condition (after the 2 tower condition) is denoted in bold. Note that the trajectory of the neural activity bump shifts nearly instantly upon the animal encountering the 3rd tower.

7) The part of the paper about BTSP is interesting, but it is presented so briefly as to appear as an afterthought. It would be good to expand this presentation.

We have expanded our discussion to include more details about BTSP as a rapid learning rule for a flexible downstream decoder, along with discussion of other potential downstream learning rules (e.g. Hebbian learning in an associative network). Of note is the experimental observation in recent work that MEC provides, via BTSP, a spatial template for the formation of new place cells in the hippocampus in novel/changing environments. We have added discussion of how our investigations into the structure of the MEC code in novel/changing environments demonstrate that it provides a suitable spatial template for behaviorally adaptive learning downstream, of the kind that BTSP could afford.

8) I found the discussion particularly confusing. The most intuitive idea would be what the authors call strong pinning - when the animal sees a landmark it places itself in that position. Is the problem with this that it would require a remapping? If so, what is wrong with that? The authors state that "The weak, fixed, landmark to grid cell pinning solution favored by MEC is thus a computationally powerful approach enabling one-shot learning of grid maps in new or dynamic environments, with small distortions, that can then subserve flexible adaptive behaviors through downstream plasticity". What does this mean? As stated above, the grid cells seem to be doing nothing other than responding in their usual way to inputs that have been modified by new evidence. Is this one-shot learning of a grid map or just an existing grid map responding to different input? Why would a system require downstream plasticity to decode more accurate information about where the animal is? Is a system that introduces a distortion and then compensates for it computationally powerful?

We thank the reviewer for this important clarification. We agree that our discussion was confusing, and that it is an overall subtle point. We have reworked the discussion and included new schematic panels to Fig. 2 in hopes of clarifying this point.

To respond briefly here, downstream plasticity is required because if a landmark is moved or a new landmark appears, the prior correspondence between position on the attractor and position on the physical virtual track is now changed to a new correspondence. For example, in Fig. 4, when towers are shifted between blocks, the position on the track immediately preceding the reward zone is now represented by a new position on the attractor (Fig. 4d, *left*). A downstream decoder must learn this new correspondence if the animal is to lick in the correct location (note also our new Extended Data Fig. 10, which shows that MEC encodes no new spatial information in this region beyond what is encoded by the attractor state). Hence pinning prevents a consistent mapping between locations on the attractor and positions on the track. The advantage of pinning is that it stabilizes the grid cell code, so that it does not drift from trial to trial. Thus paired with a flexible downstream decoder, the animal can successfully navigate across changing environments.

Thus to summarize:

1. Weak, fixed landmark pinning has two positive effects:
 - a. It stabilizes the attractor trajectory, thereby preventing drift/diffusion from trial to trial, or over time in a fixed environment.
 - b. It can immediately stabilize the attractor trajectory in a new environment, or when landmark locations are changed in an existing environment, without any need for plasticity from landmark cells to grid cells.
2. However, weak fixed landmark pinning has one potentially negative effect:
 - a. It changes the relationship between physical position and neural position on the attractor as new landmarks appear or old landmarks change position.
 - b. Because of this effect, there is likely an upper bound on the strength of the pinning so that these distortions do not become too large.

3. But this latter negative effect can be compensated for by downstream plasticity that mediates the transformation from MEC attractor state to position dependent behavior.
4. Thus the strength of pinning is likely carefully tuned
 - a. It cannot be so weak that it does not realize the positive effect of stabilizing drift/diffusion in any fixed environment
 - b. It cannot be so strong that small changes in an environment completely distort the relationship between spatial location and neural location, which would be a negative effect.

We agree with the reviewer that the language of advantages and disadvantages is confusing in our prior submission, and we have rewritten the results and discussion to omit this language, and to present our experimental observations as clearly and faithfully as possible, consistent with the above summary.

Minor points:

1) What is a twisted torus?

We agree this is a confusing term that is not essential to the Results, and have removed it except where necessary to explain the Methods. Grid cells have a toroidal population activity structure because their firing fields are periodic patterns in two dimensions. The "twist" in the twisted torus is due to the *hexagonal* lattice pattern of their firing fields. If grid cells fired instead in a square lattice, their population activity would lie along a regular torus

To explain this further, we reproduce Fig. 1 from [Guanella et al, Int J. Neural Syst., 2007] below (Reviewer Figure 9). Traversing a unit cell of the lattice pattern (a, dark grey) in the left-to-right direction corresponds to a revolution along the left-to-right axis of the torus. Traversing a unit cell in the bottom-to-top direction, however, corresponds to one revolution along the bottom-to-top direction *and* a half revolution along the left-to-right direction. Hence the activity of grid cells corresponds to a torus where the left and right sides of the neural sheet (b) are identified, and the bottom and top sides of the neural sheet are identified after a half twist.

This is the reason we have to transform the three coordinates in Extended Data Fig. 2a-d to two coordinates on the neural sheet using the transformation in Methods. We also rely on this hexagonal periodic structure when simulating the activity of grid cells in our model (Methods).

Fig. 1. (a) Repetitive rectangular structure (gray filled rectangle) of the subfields (gray circles) of grid cells defining a regular triangular tessellation of space. **(b)** Matrix of a population of 10×9 grid cells. Neighboring relationships between cells on the side of the structure are represented by gray arrows. For instance, neurons at two opposite vertical sides are neighbors.

Reviewer Figure 9. Fig. 1 reproduced from [Guanella et al, Int J. Neural Syst., 2007].

2) The abstract is particularly cryptic.

We have significantly edited the abstract, as well as other portions of the text. We hope that this editing has resulted in a clearer presentation of the results.

Reviewer Reports on the First Revision:

Referees' comments:

Referee #1 (Remarks to the Author):

The authors have substantially revised the manuscript and the writing, story and potential impact of the research is now much clearer. In particular, the Introduction, Abstract and sections of the Results are much improved. Also, the addition of the 2d data to validate the head fixed classifier is a very nice addition. Overall the clarity is much better, and I can finally understand the results and implications. I now agree with the authors that their findings are highly interesting. However, now that I understand the story better, there are several issues that I am quite concerned about, and more clarity is still required in several sections:

Major Concerns

1. The significance of the population model is unclear. While the model clearly provides good fits to some cells based on the examples in Figure 3i,j, the overall r-squared and correlations seem much lower in contrast, and the methods describing how the model is fit are unclear. Additionally, deeper analysis of the simulated activity in other task variants would likely strengthen the claims of the paper. Some more specific points and questions:

- a. The model was determined to significantly fit the data better than chance through a KS test between two distributions, one of which is shuffled (Fig. 3f). Is this a single shuffle of the spikes/tuning curves which is then compared over all blocks? How consistent is this result over repeated shuffles? A more convincing method would be to repeatedly shuffle the data, then compute the average error, then compare this distribution to the true average of unshuffled data.
- b. While the cells in Figure 3i,j are convincingly replicated by the simulated neurons, they seem somewhat misleading, given the relatively low correlations in 3f, and low r-squared values in Extended Data Figure 8c,d. In the main figure (3i,j) and extended figure (8e), it would be good to indicate the correlation between the model and the cell in each raster to indicate whether the examples shown are representative.
- c. The methods for matching the simulated grid cell position on the sheet are unclear. The authors mention that the fits of individual grid cells could be poor so they perform a grid search for the “phase sorting” of each cell within the sheet, which seems reasonable, but is not entirely clear from the methods. Is this procedure done for all cells or just the ones with poor correlations? Is every combination of the position for each cell in the sheet attempted (this seems computationally impractical)? The authors don't report the impact of this additional procedure on the predictive power over the “vanilla” model... what was the r^2 of the model without this step?

2. Further details are needed pertaining to the spectral analysis (as in Figure 1). For example, the power spectral density is either performed over 32 m segments, 10 laps at a time (which seems to change between 200-320 cm), or 3-5 m depending on which portion of the text I read. While the text refers to spectrograms, I do not see any for individual neurons (the only spectrograms are Ext. Data Fig. 2c and 4). It would be very helpful to see how the spectrum changes over the session and would also help justify the authors' choice of bin window. Along those same lines, Ext. Data Fig. 2f-h are not labeled well: Are pairs of thumbnails and line plots two simultaneously recorded neurons or the

same neuron across the session? Are the PSDs taken over the entire session or for a 10-lap section (which section?) or the average over all the 10-lap sections? In parallel to clarifying all these points, the Methods for this analysis should be written more clearly and with equations when needed: Was there a threshold to exclude periods when the mouse was running slowly? How were discrete spikes binned into 2 cm (which is roughly 40 msec of running at 50 cm/s)? Was any smoothing applied? The motivation for these questions is that the spectra shown are impressive and the method itself is a potentially powerful way of unifying the vast body of work on open field grid cells with the expanding body of work using head-fixed mice on linear treadmills. Thus it could be a major strength of the paper if the descriptions, methods, and figures are improved.

3. While the paper focuses on one-shot map formation, there are several fascinating aspects of the data that seem to be underexplored in the current manuscript. We outline a few of these fascinating aspects:

a. MEC grid cells somehow entrain perfectly when periodicity is added to the track (by adding objects), such that the population activity bump completes whole numbers of revolutions per lap. However, this observation is not given a satisfactory workup in the current manuscript. An alternative explanation is that the bump just jumps or resets near landmarks, but the data and description do not support this idea. A possibly related hint of the phenomenon appears in Ext. Data Fig. 5. In many conditions (most prominently in the dark), the bump trajectory does not randomly wander as implied by “an angular diffusion process” as stated in the Results. Rather, the trajectories appear constrained on a grid. This observation also appears remarkable and, as far as I can tell, ignored. At the very least a discussion is warranted but this could also be critical to unlocking something more fundamental about how the grid cell modules are operating.

b. Some of the main findings of this paper are that grid cells rapidly remap when landmarks change in random environments (Figure 3), and that the spatial representation is distorted around landmarks when their relative positions change (Figure 4). The fact that the grid-cell model has “significant” predictive power for the experiments in Figure 3 supports the rapid remapping result, but does the model also capture the firing field distortions observed when landmarks are shifted relative to one another (experiment in Figure 4)? To explicitly show that the model can account for both the remapping shown in Figure 3 and the distortion shown in Figure 4 would strengthen the claims being made here.

4. The hypotheses in Figure 2j-l are poorly described, despite them serving as a crucial transition laying out some of the main points in the paper. Figure 2j is never referenced and Figure 2k,l are described out of order. This section should be revised to more clearly outline these important hypotheses.

5. The data, analysis and modeling in Figure 4 seem like a weak part of the manuscript, and (as noted above) there are far more interesting features in the author’s data that seem more interesting/important to focus on. I am not convinced that the task in Figure 4 is MEC dependent. Among the arguments listed by the authors, the one that could have been convincing was the muscimol injection, but I find the behavioral effect to be mild at best. I agree that the licking is a bit more spread out across the track after muscimol injection, but the data in Figure 4i shows that the animal is still performing the task quite well. The authors should quantify anticipatory licking change after muscimol, rather than the odd metric of licking SD in 4k (see Pettit et al Nature 2022 for a clear

behavioral effector muscimol inactivation during a VR navigation task). This suggests that MEC possibly biases/influences task performance (perhaps the mice have trouble knowing where they are wrt the towers?), but it is certainly not MEC dependent. The mice could solve the task instead with reward centric coding in LEC, see for example Issa et al (Nat Neuro 2024). This pathway might send reward location information that could be tied to a sensory cue, like a landmark in track, in the hippocampus. This seems just as likely a scenario as the BTSP based position error correction model that the authors propose. While this BTSP hippocampal model provides one possible explanation for the problem posed by the authors, it seems highly speculative in my opinion and (with so many other interesting features to focus on, and other plausible explanations) it's not clear to me why it's included.

6. "one shot" in the title is misleading, and seems incorrect to me. I don't think the title implies to readers what the authors are intending. To me it implies rapid (one shot) plasticity, which is not what the authors are saying here; in fact they are saying the opposite, that there is little or no plasticity and the circuit connectivity is fixed. Similarly, the idea that remapping is "one shot" (single trial) is not new; global remapping has been known to be induced in one trial for example and I don't think this is described as "one shot". Shouldn't the title be something more like "Fixed connections between landmarks and grid cells enable rapid and stable spatial maps"?

Minor Concerns:

1. KS test results for Fig 3c are reported in the text, but not for Fig 3d.
2. Figure 3f: If predictivity is simply the Pearson's correlation between the model and the data, it seems better to simply label the axis as such, rather than as "predictivity."
3. Fig 3i-j: why do real spikes look stretched compared to simulated spikes in the rasters? Are they bursty or smoothed in some way that the simulated data is not?
4. Figure 4h: How is the correlation between behavioral error and grid cell stability calculated? Why 2099 cells and not 2630 as in Fig 4c? Why does SEM collapse at the lowest stability value? It would be reassuring and more direct to see this plot with the individual trials overlaid as dots, particularly because this is what the statistical tests were computed over (while the detailed n of sessions, mice, modules, cells is appreciated, the relevant value would be trials used for the test).
5. Extended Data Figure 8e is mislabelled as f in the caption.
6. For most statistics, the authors report n for animals, sessions, modules, cells, etc. While the detail is appreciated, it is not clear for many of the reported tests what level the test was actually computed over and what the true degrees of freedom of the tests were, which is essential for evaluating effect size.
7. Is the diagram in 4a wrong with reward location? I could not tell because the description of reward locations and distances in text does not seem to match what is in methods.

Referee #2 (Remarks to the Author):

This is an excellent paper and now suitable for publication. One small issue on the new EDFig 6. Part B does not appear to be a "view of a stationary period". And I am confused by the values on the different plots of theta. The theta values range from 0 to ~ 1 in part A but from -3 to 3 in part C. Please explain why these ranges are different and what a value of plus/minus 3 would mean for theta.

Referee #3 (Remarks to the Author):

The authors have done an excellent job of responding to previous criticisms and, most importantly, the paper is now much clearer. This is an important piece of work, skillfully done and intelligently analyzed.

Author Rebuttals to First Revision:

We thank the reviewers for their helpful comments and constructive feedback. We have now included new analyses, figure panels, and made changes to the text in accordance with the suggestions. We feel these changes further clarify the paper's conclusions and broaden its impact. Below, we provide a detailed response to each suggestion. Note that several new figure panels were added to the Extended Data - we have also supplied these figure panels in the response below for ease of reference. Some of the panels are also only presented in the response to reviewer comments (which we note below where appropriate). However, if either the reviewer or editor feels that any of these figures should be added to the manuscript, we are more than happy to do so. Note that major changes to the manuscript are denoted by gray highlighting in the main text and extended data figure legends.

Referee #1:

The authors have substantially revised the manuscript and the writing, story and potential impact of the research is now much clearer. In particular, the Introduction, Abstract and sections of the Results are much improved. Also, the addition of the 2d data to validate the head fixed classifier is a very nice addition. Overall the clarity is much better, and I can finally understand the results and implications. I now agree with the authors that their findings are highly interesting.

We thank the reviewer for their feedback on both this and the prior round of review, as we feel it has significantly clarified the results and improved the overall manuscript. Below, we have addressed the concerns raised with new analyses, figures and modifications to the text.

However, now that I understand the story better, there are several issues that I am quite concerned about, and more clarity is still required in several sections:

Major Concerns

1. The significance of the population model is unclear. While the model clearly provides good fits to some cells based on the examples in Figure 3i,j, the overall r-squared and correlations seem much lower in contrast, and the methods describing how the model is fit are unclear.

First, in terms of the methods describing the model, we have significantly expanded the methods section to provide more detail on how the model is fit. Second, in terms of the significance of the population model. At a high level, we feel the population model provides a concise, mechanistic explanation for most of the results in the paper (and see below for new model simulations of the build-up track from Figure 2 and the hidden-reward task from Figure 4). Without the model, it is more challenging to interpret seemingly disparate remapping responses to different environmental changes. However, a single model framework, as presented here, can conceptually answer: i) what enables grid cells to form rapid maps of space in unfamiliar environments, ii) what allows these rapidly formed maps to remain stable and iii) how these maps are affected by the addition, removal or displacement of external landmark cues. These connections are bolstered by a new series of simulations, at the reviewer's suggestion, in the build-up track and hidden reward task. While we agree with the reviewer that the model does not always perfectly capture the neural data, we nonetheless feel it has significant explanatory power over the dynamics that govern MEC grid cell firing patterns, and opens the door for future work to further understand how other biologically plausible constraints on the model might even further improve its predictive power (e.g. see discussion paragraph 2 regarding the nature of landmark inputs to the grid cell network).

Additionally, deeper analysis of the simulated activity in other task variants would likely strengthen the claims of the paper.

We agree with the reviewer that this is an important and interesting extension of the model. Thus, we have included below and in main Figure 4 and Extended Data Figure 8 simulations of the build-up track (Extended Data Figure 8) and the hidden reward task (Figure 4 and Extended Data Figure 8). Very importantly, note that these model comparisons can only be performed at the level of firing pattern features - for example - distortions, stability, and remapping across environments. This is because in the build-up track and hidden reward task, we do not have a sufficient number of blocks to quantify the pinning phase of the landmark inputs to the grid cell neural sheet. The predictive power of the random environment task lies in the fact that we have 12 blocks using the same landmarks but in different spatial locations, which allows us to quantify the pinning phase of the landmark inputs to the grid cell neural sheet, holding one block out at a time.

First, in Reviewer Fig. 1, we show 15 simulations for the build-up track. Note that a subset of these panels have been added to Extended Data Fig. 8. On the build-up track, we observed that the model reproduced many of the features we observed in experimentally measured grid cell firing patterns. These features include:

- Drift in the absence of visual cues (see trials 0-100 in Reviewer Fig. 1, below). The Moiré-like patterns in the dark blocks which drift from trial to trial in both simulated and experimentally measured data indicate the rotation of the underlying 2D spatial map. In other words, as the slice angle rotates, the spatial frequencies of grid cell firing increase or decrease from trial to trial, producing this pattern of firing activity.
- Increasing spatial stability as more cues are added to the track. We quantified this for the model simulations using the same analyses as in Figure 2f and noted that the underlying dynamics between the model and simulations are strikingly similar (compare panel e to f in the Reviewer Fig. 1).
- In particular, we noted rapid remapping in spatial maps between environments, with the simulated grid cell population maintaining high spatial correlation within, but not across, blocks containing at least one tower (Reviewer Fig. 1f).
- Multiple stable spatial maps, with occasional spontaneous remapping within a given environment. An example of this phenomena is illustrated by the blue dot for experimental data (Reviewer Fig. 1, panel a) and simulated data (Reviewer Figure 1, panel d, top row).

Second, in Reviewer Fig. 2, we show 10 simulations for the hidden reward task. Note that a subset of these panels have also been added to main Figure 4 and quantification of the distortions measured in real and simulated cells has been added to Extended Data Fig. 8. On the hidden reward task, we observed that the model reproduced the local shifts in firing field locations after the towers were moved, locally compressing or expanding the distance between firing fields, as in the experimentally recorded grid cell data shown in Figure 4.

Reviewer Fig. 1. Model simulations in the build-up track (Figure 2). **a-c**, Pairs of simultaneously recorded grid cells from three mice, N2 (a) N1 (b) and O3 (c). Firing rate maps are plotted as in Figure 2b, with horizontal lines corresponding to different blocks of the build-up track. **d**, Results of 15 independent simulations using random pinning phases for each landmark. Red dots correspond to simulated neural spikes. **e-f**, In both neural data (e) and model simulations (f), grid cell activity rapidly stabilizes within each trial block, but remaps between blocks. **e**, reproduced from Fig. 2f. **f**, Results from 30 randomly simulated populations of 100 grid cells each. *Note that a subset of these panels are now also in Extended Data Fig. 8

Reviewer Fig. 2. Model simulations in the hidden reward task (Figure 4). **a.** Example rasters for pairs of simultaneously recorded grid cells from four mice. Black dots correspond to experimentally measured neural spikes and rasters are plotted as in Figure 4c. **b.** Results of 10 independent simulations using random pinning phases for each landmark. The bottom right cell shows no firing fields because the bump did not pass over the randomly chosen location of this simulated cell on the neural sheet. **c.** Quantification of grid cell distortions. Distances between grid cell firing fields became compressed as landmarks were shifted closer together from block A to block B. Because our data includes cells with multiple spatial scales, we normalized the per-cell distances between firing fields in blocks A and B by subtracting the per-cell distances between firing fields in block C (which differs from A only by an overall shift in its arrangement of landmarks). Since blocks A and C have similar distances between firing fields, the gray histogram is centered at zero, while the red histogram is shifted 20-30cm left because distances between firing fields are compressed in block B. We observe significant distortions in both real grid cells (top, KS test, $p < 1e-5$, $n = 86$ blocks; cell $n = 457$, mouse $n = 5$, session $n = 10$), and in cells simulated by the model (KS test, $p < 1e-5$, $n = 150$ blocks, simulated cell $n = 500$, random simulated session $n = 20$). *Note that a subset of these panels are now also in Extended Data Fig. 8

Some more specific points and questions:

a. The model was determined to significantly fit the data better than chance through a KS test between two distributions, one of which is shuffled (Fig. 3f). Is this a single shuffle of the spikes/tuning curves which is then compared over all blocks? How consistent is this result over repeated shuffles? A more convincing method would be to repeatedly shuffle the data, then compute the average error, then compare this distribution to the true average of unshuffled data.

We apologize this was not clear in the manuscript. The shuffle we performed, if we understand the reviewer correctly, is exactly the one the reviewer describes here. For the shuffle, we: 1) repeatedly shifted the predicted tuning curves in each block, 2) computed the average error, 3) compared this distribution to the true average of the unshuffled data. The gray histogram in Fig. 3f represents the distribution of average errors over 200 such random shuffles for each cell. We have updated the manuscript and methods to clarify this procedure, which we have provided below for ease of reference. We would also note that this shuffle is much stronger than shuffling spikes, as it preserves the structure in the predicted tuning curves and shifts only the locations of their peaks.

Updated text in Fig. 3f caption:

"f. Histogram of model predictivity (Pearson's r between the predicted and true tuning curves in each block, averaged over blocks) across all cells, compared to a random shuffle. For the shuffle we repeatedly shift each cell's predicted tuning curves by a random amount, and report the distribution of single-cell predictivities over 200 such shuffles (KS test, $p < 1e-10$, animal $n = 3$, session $n = 3$, module $n = 3$, cell $n = 91$)."

Updated text in Methods:

"We evaluated the performance of the model by computing the Pearson correlation between the estimated firing rates $\hat{r}_i(x)$ and the true firing rates $r_i(x)$. We compared this to two shuffled conditions. In the first, we repeatedly shifted each cell's predicted firing rates by a random amount, $\hat{r}_i((x + \delta) \bmod L)$, $\delta \sim Unif(0, L)$, where L is the length of the track and $\bmod L$ accounts for the periodic nature of the track (Fig. 3f). This shuffle is stronger than simply shuffling spikes, as it preserves the spatial structure of the predicted tuning curves and shifts only the locations of their peaks. In the second, we fed a randomly shuffled set of landmark locations to the model (Extended Data Fig. 8e). This shuffle was designed to test whether the model uses information about the landmarks locations in the held-out environment to predict the grid cells' tuning curves, or simply reproduces their average firing properties."

b. While the cells in Figure 3i,j are convincingly replicated by the simulated neurons, they seem somewhat misleading, given the relatively low correlations in 3f, and low r-squared values in Extended Data Figure 8c,d. In the main figure (3i,j) and extended figure (8e), it would be good to indicate the correlation between the model and the cell in each raster to indicate whether the examples shown are representative.

We have added the correlation between the model and each cell in both main Figure 3 and Extended Data Figure 8. For both figures, the cells shown exhibit a range of correlation values. We also replaced the second cell in Figure 3 with a lower correlation example, to provide a more representative range of examples in Figure 3.

c. The methods for matching the simulated grid cell position on the sheet are unclear. The authors mention that the fits of individual grid cells could be poor so they perform a grid search for the "phase sorting" of each cell within the sheet, which seems reasonable, but is not entirely clear from the methods. Is this procedure done for all cells or just the ones with poor correlations? Is every combination of the position for each cell in the sheet attempted (this seems computationally impractical)? The authors don't report the impact of this additional procedure on the predictive power over the "vanilla" model... what was the r^2 of the model without this step?

We sincerely apologize for the confusion caused by this portion of the text. We in fact only report the results from the "vanilla" model, which does not involve any grid search or phase sorting. In a prior version of the manuscript, we did indeed perform phase shifting. However, while this phase shifting can sometimes improve the model fit for individual cells, we felt it was more parsimonious to simply consider the predictive power of the model without this phase sorting step. In addition, without the phase sorting step, the model actually does not get any information about the environment it must predict (as it's predicting the firing pattern for the held out environment), in contrast to phase sorting in which the model makes many predictions and only the best prediction is picked. We have deleted

this sentence from the Methods, as it was intended to explain why some observed correlations are low but we now realize reduces the clarity of the Methods.

2. Further details are needed pertaining to the spectral analysis (as in Figure 1). For example, the power spectral density is either performed over 32 m segments, 10 laps at a time (which seems to change between 200-320 cm), or 3-5 m depending on which portion of the text I read. While the text refers to spectrograms, I do not see any for individual neurons (the only spectrograms are Ext. Data Fig. 2c and 4). It would be very helpful to see how the spectrum changes over the session and would also help justify the authors' choice of bin window.

We agree with the reviewer that these are important points to clarify. We have added text to the Methods to make this more concrete:

"We identified putative grid cells by computing spectrograms for each neuron's spatial firing rate in the dark. We first z-scored each cell's spatial firing rates. Spectrograms were computed using `scipy.signal.spectrogram` with `nperseg = 1600` and `noverlap = 1400`, corresponding to segments of width 32 m (or 10 laps on the buildup track) with measurements every 4 m (example single-cell spectrograms in Extended Data Fig. 4). The segment width and overlap were chosen to balance resolution in the frequency domain..."

Below in Reviewer Fig. 3 we show single-neuron spectrograms for 30 simultaneously recorded grid cells across three different grid modules. Note that the frequencies drift significantly over timescales of order 100 trials. This drift is the primary reason we computed spectrograms on segments of 10 trials. Reviewer Fig. 4 highlights that the measured PSDs can be very different between two different segments of 10 trials. Using overlapping segments in our spectrograms allowed us to further improve the spatial resolution to obtain measurements every 4 m. A subset of these panels has been added to Extended Data Fig. 4.

Reviewer Fig. 3. Single-neuron spectrograms across an entire session, for 10 simultaneously recorded grid cells from 3 separate modules (30 grid cells total, 10 from each module). Color (dark low, white high) represents power in a given frequency f (y-axis). For this session example, one of the higher frequencies was more stable but note the drift in the lower frequencies, which indicates the slice angle is rotating (consistent with response to reviewer Figure 4). *Note that a subset of these panels have been added to Extended Data Fig. 4.

Along those same lines, Ext. Data Fig. 2f-h are not labeled well: Are pairs of thumbnails and line plots two simultaneously recorded neurons or the same neuron across the session? Are the PSDs taken over the entire session or for a 10-lap section (which section?) or the average over all the 10-lap sections?

We apologize this was not clear. Each pair represents two simultaneously recorded co-modular grid cells. PSDs are taken over a 10 lap section at the start of the session. To clarify this, we have modified the legend for Extended Data Fig. 2, provided below for ease of reference.

Updated legend for Extended Data Fig. 2f-h: "**f-h.** Data from three mice shown, with mouse labels above each column. Left: Example spike train raster plots for pairs of grid cells from two or more

different simultaneously recorded modules over 200 laps in the dark. Dots indicate spikes, color coded to match the modules identified on (right), sorted by overall grid scale $\lambda = \sqrt{2}/\sqrt{f_1^2 + f_2^2 + f_3^2}$. Right: Power spectral densities (PSDs) for each cell in (left), computed over the first 10 laps of the session (16 meters), revealed a prominent three-peaked structure within each module. PSDs are consistent for two simultaneously recorded cells within one module, but differ between modules."

Furthermore, to illustrate how the locations of the peaks in the spectrogram evolve over timescales of 100 laps, we compare spectra computed over laps 0-10 and laps 80-90 (Reviewer Fig. 4) for three simultaneously recorded cells from the session in Reviewer Fig. 3.

Reviewer Fig. 4. Power spectral densities (PSDs) evolve over timescales of 100 laps. **a.** Spiketrain rasters for three simultaneously recorded grid cells over 180 laps in the dark. **b.** Corresponding PSDs computed over 10-lap windows. Either laps 0-10 (black), or laps 80-90 (gold). Locations of three estimated peaks are indicated with vertical dashed lines.

In parallel to clarifying all these points, the Methods for this analysis should be written more clearly and with equations when needed: Was there a threshold to exclude periods when the mouse was running slowly? How were discrete spikes binned into 2 cm (which is roughly 40 msec of running at 50 cm/s)? Was any smoothing applied? The motivation for these questions is that the spectra shown are impressive and the method itself is a potentially powerful way of unifying the vast body of work on open field grid cells with the expanding body of work using head-fixed mice on linear treadmills. Thus it could be a major strength of the paper if the descriptions, methods, and figures are improved.

We've greatly expanded the Methods section with more mathematical detail and instructions for reproducing analyses, as well as how the data was processed. A few excerpts are reproduced below for ease of reference, but all changes to the Methods are noted in the appropriate section of the

manuscript. In addition, to clarify, for pre-processing data were not speed filtered, so no thresholds were used to exclude periods of slow running. A spike at time t was considered 'in' a given 2 cm bin defined by position edges x_1 and x_2 if at time t the position of the animal was greater than or equal to x_1 and less than x_2 . After binning spikes into spatial bins, spatial firing rates were calculated by dividing the number of spikes in a bin by the time occupancy in that bin. Smoothing was then applied using a 2 bin Gaussian kernel. The methods have been updated to reflect this calculation of the spatial firing rate.

Excerpts of new text from the Methods section:

"Data was not speed-filtered. To calculate spatial firing rates, spikes were first binned into 2 cm spatial bins and divided by time occupancy per spatial bin. Then, the firing rates were smoothed with a 2 bin Gaussian kernel."

"An idealized grid cell's hexagonal spatial map can be written as the sum of three plane waves oriented at 0° , 60° , and 120° ,

$$r_i(\vec{x}) = \sum_{a=1}^3 \cos(\vec{k}_a \cdot \vec{x} - \phi_a^i)$$

Where $r_i(\vec{x})$ is the activity of neuron i at 2D location \vec{x} , and $\vec{k}_{1,2,3}$ are wave vectors oriented at 0° , 60° , and 120° , which set the overall grid scale of the module. Hence each grid cell in a module is uniquely determined by its phases $\phi_{1,2,3}^i$. Our goal was to infer these phases from recordings while the animal is running on the 1D VR track. In 1D, the animal's trajectory follows a straight line at some angle γ , $\vec{x} = (x \cos(\gamma), x \sin(\gamma))$, and hence the 2D wave vectors are transformed to 1D scalar frequencies $f_{1,2,3} = \vec{k}_{1,2,3} \cdot (\cos(\gamma), \sin(\gamma))$,

$$r_i(x) = \sum_{a=1}^3 \cos(f_a x - \phi_a^i)$$

These three frequencies $f_{1,2,3}$ are the peaks we observe in the PSDs in Fig. 1. To estimate $f_{1,2,3}$, we used a standard peak-finding algorithm..."

"To extract the moment-by-moment attractor state $\vec{\theta}(x)$ of the grid cell population activity, we estimated the center of mass of the bump of activity on the sheet. We computed the average of the cells' phases on the neural sheet weighted by their (filtered) firing rates $r_i(\vec{x})$. Because the phases are periodic, we computed a circular average via the following formula,

$$\psi_{1,2,3}(x) = \text{atan2}\left(\sum_{i=1}^N \sin(\phi_{1,2,3}^i) r_i(x), \sum_{i=1}^N \cos(\phi_{1,2,3}^i) r_i(x)\right)$$

Where $\psi_{1,2,3}(x)$ represents the firing-rate-weighted average of each phase. If the population activity of grid cells is well described by the motions of a bump on a 2D sheet, then the three coordinates $\psi_{1,2,3}(x)$ represent only two degrees of freedom. In Extended Data Fig. 2 we show that $\psi_{1,2,3}(x)$ do not take on all possible values, but are indeed restricted to a 2D subspace. "

"The above procedure for extracting the moment-by-moment center of mass of the bump on the neural sheet has a geometrically equivalent interpretation as identifying the instantaneous location of the high-dimensional population activity vector on a toroidal attractor. To see this, note that computing the arguments of the circular average above,

$$\psi_{1,2,3}(x) = \text{atan2}\left(\sum_{i=1}^N \sin(\phi_{1,2,3}^i) r_i(x), \sum_{i=1}^N \cos(\phi_{1,2,3}^i) r_i(x)\right)$$

involves first projecting the N-dimensional neural activity vector $r_i(x)$ onto 3 pairs of axes spaces spanned by the vectors $u_{1,2,3}^i = \cos(\phi_{1,2,3}^i)$ and $v_{1,2,3}^i = \sin(\phi_{1,2,3}^i)$. Recall that an idealized grid cell's firing rate in 1D can be written as,

$$r_i(x) = \sum_{a=1}^3 \cos(f_a x - \phi_a^i)$$

Which we can rearrange to find,

$$\begin{aligned} r_i(x) &= \sum_{a=1}^3 \cos(\phi_a^i) \cos(f_a x) + \sin(\phi_a^i) \sin(f_a x) \\ &= \sum_{a=1}^3 u_a^i \cos(f_a x) + v_a^i \sin(f_a x) \end{aligned}$$

Hence the idealized grid cell population activity lives in a subspace spanned by the same three pairs of axes. Within each pair of axes the activity lies along a ring, and in the full subspace lies along a torus (Extended Data Fig 2a). Therefore, if we have reliably inferred the phases of our recorded grid cells, and they are well approximated by the idealized grid cells above, then projecting their population activity onto these three pairs of axes should reveal three rings. Indeed this is what we find for our recordings on the 1D VR track (Extended Data Fig 2bc), indicating that the population activity lies close to a toroidal attractor. Note that estimating the firing-rate-weighted average of each phase on the neural sheet $\psi_{1,2,3}(x)$ is mathematically equivalent to extracting the angle on each of the three rings. Hence the instantaneous center of mass of the bump of activity on the neural sheet is in one-to-one correspondence with the instantaneous location of the high-dimensional population activity on the toroidal attractor."

3. While the paper focuses on one-shot map formation, there are several fascinating aspects of the data that seem to be underexplored in the current manuscript. We outline a few of these fascinating aspects:

a. MEC grid cells somehow entrain perfectly when periodicity is added to the track (by adding objects), such that the population activity bump completes whole numbers of revolutions per lap. However, this observation is not given a satisfactory workup in the current manuscript. An alternative explanation is that the bump just jumps or resets near landmarks, but the data and description do not support this idea. A possibly related hint of the phenomenon appears in Ext. Data Fig. 5. In many conditions (most prominently in the dark), the bump trajectory does not randomly wander as implied by “an angular diffusion process” as stated in the Results. Rather, the trajectories appear constrained on a grid. This observation also appears remarkable and, as far as I can tell, ignored. At the very least a discussion is warranted but this could also be critical to unlocking something more fundamental about how the grid cell modules are operating.

We agree with the reviewer that the entrainment of grid cells to whole revolutions per lap after the addition of landmarks is one of the intriguing findings of the manuscript. First, we have added text to the discussion to highlight this further, which we provide below for ease of reference.

New text in the discussion: Another intriguing line of future study would be to understand how the certainty, regularity or sensory modality of landmarks can entrain the bump of activity to complete whole revolutions along each axis of the torus in concert with the completion of the animal on one lap down the track.

Second, we wish to clarify that the appearance of trajectories constrained on a grid is a function of the unwrapping process used in the previous version of Extended Data Fig. 5 to visualize bump trajectories over multiple copies of the neural sheet. In the dark, the bump does not show a preference for any one location on the neural sheet over another. The grid-like appearance of the unwrapped trajectories is an unfortunate and established consequence of the sensitivity of the unwrapping of angular phases, where small amounts of noise can cause independent trajectories to become dislocated by one period (or one copy of the neural sheet, leading to a grid-like pattern). We would note that this unwrapping was only used for visualization purposes, no analysis was run on using this process. Nonetheless, we devised a new procedure for mitigating these unwrapping errors which we include in an updated version of Extended Data Fig. 5 which results in much cleaner trajectories (reproduced below for ease of reference). We have updated the Methods with a detailed description of this procedure, reproduced below:

Since the neural sheet is periodic, we also occasionally unwrap θ_1, θ_2 to plot them as continuous trajectories across multiple copies of the neural sheet (Figs. 1g, 2d,i). Note that this unwrapping is used only for visualization and not used in data analyses. Unwrapping is sensitive to noise and often leads to unwrapping errors whereby the unwrapped trajectory becomes dislocated by one neural sheet length. To mitigate this issue, in Extended Data Fig. 5 we adopt the following approach. Due to the discontinuous nature of the unwrapping procedure, it is sensitive to noise only at the edge of the neural sheet. However, the location of the edge of the neural sheet is

arbitrary (we can experimentally measure only the relative phases between grid cells; the location of the origin is arbitrarily chosen). Moreover, different choices of the origin will lead to more or fewer unwrapping errors. Taking advantage of this freedom, we perform a grid search over 100 different choices of the origin for each window of five trials by applying a global shift to all cell's phases $\phi_{1,2,3}^i \rightarrow \phi_{1,2,3}^i + \eta_{1,2,3} \bmod 2\pi$, $\eta_{1,2,3} \in [0, 2\pi]$ and keep the choice of origin which leads to the most consistent trajectories over the five trial window (the underlying assumption is that trajectories do not drift much over windows of five trials, and that unwrapping errors will produce inconsistent trajectories). This procedure is somewhat cumbersome but leads to cleaner visualizations of the unwrapped bump trajectory for noisy trajectories.

We have replaced Extended Data Fig. 5 with this new method and provide the new figure below for ease of reference.

Extended Data Fig. 5: Additional activity bump trajectories and remapping statistics on the build-up track. **a-c.** Single-trial bump trajectories across all 9 blocks of the build-up track, for three different animals (Methods).

b. Some of the main findings of this paper are that grid cells rapidly remap when landmarks change in random environments (Figure 3), and that the spatial representation is distorted around landmarks when their relative positions change (Figure 4). The fact that the grid-cell model has “significant” predictive power for the experiments in Figure 3 supports the rapid remapping result, but does the model also capture the firing field distortions observed when landmarks are shifted relative to one another (experiment in Figure 4)? To explicitly show that the model can account for both the remapping shown in Figure 3 and the distortion shown in Figure 4 would strengthen the claims being made here.

We thank the reviewer for this excellent suggestion. We have performed additional simulations of our model for the tower-shift (hidden reward task), as shown in Reviewer Fig. 2 and Extended Data Fig. 8. We found that simulated cells in the model indeed distort in precisely the same way as experimentally recorded grid cells. Namely, their firing fields locally shifted with the shifts in the landmarks, stretching or compressing the distance between the firing fields.

4. The hypotheses in Figure 2j-l are poorly described, despite them serving as a crucial transition laying out some of the main points in the paper. Figure 2j is never referenced and Figure 2k,l are described out of order. This section should be revised to more clearly outline these important hypotheses.

We apologize that this was not clear. We have edited the text to reflect the correct figure panel order, which also now better encompasses panel j. For ease of reference, the text now reads:

How does the addition of visual landmarks to the build-up track (Fig. 2a) cause the activity bump trajectory to rapidly change yet then remain stable on subsequent trials (Fig. 2j-l)? One possibility is that landmark inputs do not shift where they project to on the neural sheet (Fig. 2j-k). In this case, landmarks would influence the trajectory of the activity bump, causing slight deflections in the trajectory of the activity bump when the animal approaches a landmark. This would result in stable but distorted maps, meaning that the distance traveled by the mouse in physical space would not retain a fixed proportion to the distance traveled by the activity bump on the neural sheet (Fig. 2j-k). Another possibility is that new landmark inputs rapidly shift to project to the appropriate location of the activity bump on the neural sheet (Fig. 2l). This would result in alignment between where landmark inputs project to on the neural sheet and neural coordinate estimates, as read out by the location of the activity bump on the neural sheet. While such learning between landmarks and neural coordinate estimates have been reported in the fly compass system^{18,19} and, under some conditions, in the mammalian head direction cell system^{57,58}, it can take on the order of minutes to accomplish.

5. The data, analysis and modeling in Figure 4 seem like a weak part of the manuscript, and (as noted above) there are far more interesting features in the author's data that seem more interesting/important to focus on. I am not convinced that the task in Figure 4 is MEC dependent. Among the arguments listed by the authors, the one that could have been convincing was the muscimol injection, but I find the behavioral effect to be mild at best. I agree that the licking is a bit more spread out across the track after muscimol injection, but the data in Figure 4i shows that the animal is still performing the task quite well. The authors should quantify anticipatory licking change after muscimol, rather than the odd metric of licking SD in 4k (see Pettit et al Nature 2022 for a clear behavioral effector muscimol inactivation during a VR navigation task). This suggests that MEC possibly biases/influences task performance (perhaps the mice have trouble knowing where they are wrt the towers?), but it is certainly not MEC dependent. The mice could solve the task instead with reward centric coding in LEC, see for example Issa et al (Nat Neuro 2024). This pathway might send reward location information that could be tied to a sensory cue, like a landmark in track, in the hippocampus. This seems just as likely a scenario as the BTSP based position error correction model

that the authors propose. While this BTSP hippocampal model provides one possible explanation for the problem posed by the authors, it seems highly speculative in my opinion and (with so many other interesting features to focus on, and other plausible explanations) it's not clear to me why it's included.

First, we agree with the reviewer that there are mechanisms outside of BTSP that could play a role in supporting task performance. Thus, we have added text to the discussion on BTSP that proposes lateral entorhinal cortex is a candidate region supporting flexible behavior.

Second, in terms of the specific comments regarding whether the task is dependent on the MEC. We agree with the reviewer that anticipatory licking, for example, is a clearer metric to use. We have thus performed additional analyses to quantify behavior in saline and muscimol sessions, including the licking selectivity metric (Pettit et al., Nature, 2022) recommended by the reviewer. This new analysis is now included in main Figure 4, and provided below as Reviewer Fig. 5 for ease of reference. We hope this new behavioral data better illustrates the lack of accurate performance in mice after muscimol injection. We would note that the task is specifically designed to force animals to integrate visual cues with self-motion cues by withholding their licks even after the last visual landmark appears on the track. In the region beyond the last visual landmark, the animals must rely on path integration to correctly lick in anticipation of the reward zone. We note that in the muscimol condition, the animals do not withhold their licks, and therefore could be relying on a visual-only strategy (for instance starting to lick when they see the last tower, and continuing to lick until they receive a reward, Reviewer Fig. 5). Note that animal AK2 below does appear to have licking selectivity spared after muscimol injection. Consistent with the spared behavior, we found the spread of muscimol was not as complete as in other sessions based on the spread of fluorescent muscimol observed in histology as well as the number of active units recorded (Extended Data Fig. 9). In the saline condition, in contrast, the animals withhold their licks even after they have passed the last visual landmark, and their lick distributions do not vary from block to block, indicating that they are successfully integrating visual cues with path integration.

Reviewer Fig. 5. a. Non-consummatory lick rasters for saline and muscimol sessions from 6 animals. Purple shaded region represents the reward zone. **b.** Licking selectivity for saline and muscimol sessions. Licking selectivity is defined as in [Pettit et al Nature '22], as the difference in number of licks within a 10-centimeter region immediately preceding the reward zone, and a 10-centimeter region earlier on the track not associated with the reward, divided by the sum of licks in both regions. Connected dots correspond to pairs of sessions from individual mice ($p < 0.001$, two sided permutation test with 1,000 shuffles, session pairs $n = 8$; mouse $n = 6$). **c.** Histology showing the extent of fluorescent muscimol (in orange) from AK2. Note that in the left hemisphere, the extent of muscimol spread is only partial, consistent with the partially spared behavioral performance of the animal as well as some extant neural activity (see manuscript, Extended Data Fig. 9).

6. “one shot” in the title is misleading, and seems incorrect to me. I don’t think the title implies to readers what the authors are intending. To me it implies rapid (one shot) plasticity, which is not what the authors are saying here; in fact they are saying the opposite, that there is little or no plasticity and the circuit connectivity is fixed. Similarly, the idea that remapping is “one shot” (single trial) is not new; global remapping has been known to be induced in one trial for example and I don’t think this is described as “one shot”. Shouldn’t the title be something more like “Fixed connections between landmarks and grid cells enable rapid and stable spatial maps”?

We agree with the reviewer and have changed the title.

Minor Concerns:

1. KS test results for Fig 3c are reported in the text, but not for Fig 3d.

This has been added. We apologize for the omission.

2. Figure 3f: If predictivity is simply the Pearson's correlation between the model and the data, it seems better to simply label the axis as such, rather than as "predictivity."

We agree this is simpler and have updated the axis labels.

3. Fig 3i-j: why do real spikes look stretched compared to simulated spikes in the rasters? Are they bursty or smoothed in some way that the simulated data is not?

Both spike rasters and simulations were smoothed for analyses. As all analyses used the smoothed spike rasters, we included these in the visualization, while the simulations - for visualization purposes only - are not smoothed (Reviewer Fig. 6). This is because to generate the visualization for model cells, we draw spikes using a Poisson process from the smoothed rate maps that are used to generate the correlation values. We have added text to the methods and figure legends to clarify these points.

Reviewer Fig. 6. Example smoothed (black, left) and unsmoothed (black, right) spatial firing rate maps for the random environments task. Smoothing is performed with a 2 spatial bin (4 cm) Gaussian kernel. Note that correlations were computed between smoothed grid cell maps and smooth underlying rate maps for simulated grid cells. Poisson spikes are drawn for simulated cell visualizations (red rate maps).

4. Figure 4h: How is the correlation between behavioral error and grid cell stability calculated? Why 2099 cells and not 2630 as in Fig 4c? Why does SEM collapse at the lowest stability value? It would be reassuring and more direct to see this plot with the individual trials overlaid as dots, particularly because this is what the statistical tests were computed over (while the detailed n of sessions, mice, modules, cells is appreciated, the relevant value would be trials used for the test).

We thank the reviewer for this important clarification. We have updated Fig. 4i with individual trials overlaid. Because the number of points is quite large and it is difficult to make out their density, we have included for completeness a 2D histogram in Reviewer Fig. 7b. The analysis includes 2099 cells

because we include only grid cells, while the decoding analysis using 2630 cells includes all MEC cells. However, for the decoding analysis we restrict to sessions where the animal is licking successfully and the MEC code is stable, while here we include both stable and unstable sessions to investigate the correlation of stability with behavior. We have also switched to a bootstrapped 95% confidence interval and updated the binning so that the SEM does not collapse. We apologize as the correlation previously reported referred to a prior version of the manuscript which measured block-by-block correlation. We have updated the correlation to trial-by-trial correlation which despite a lower correlation we believe captures more interesting variations in spontaneous behavior. We have updated the text to provide more detail on how this correlation was calculated.

Reviewer Fig. 7. a. Licking error (average distance of non-consummatory licks from reward zone) was large on trials where the grid cell population code was unstable (quantified by the trial-by-trial Pearson r correlation of population activity within a block, gray points represent individual trials, Pearson $r = -0.20$, $p < 1e-10$, trial $n = 1668$; animal $n = 8$, session $n = 17$, module $n = 50$, cell $n = 2099$). Licking error decreased on trials where the grid cell code was more stable (black dots indicate average licking error within bins of grid cell stability, shaded area represents bootstrapped 95% confidence interval across trials, trial $n = 1668$ animal $n = 8$, session $n = 17$, module $n = 50$, cell $n = 2099$). **b.** 2d histogram of the same gray points in **a**.

5. Extended Data Figure 8e is mislabelled as f in the caption.

This has been fixed, and Extended Data Fig. 8 has been significantly modified.

6. For most statistics, the authors report n for animals, sessions, modules, cells, etc. While the detail is appreciated, it is not clear for many of the reported tests what level the test was actually computed over and what the true degrees of freedom of the tests were, which is essential for evaluating effect size.

We agree with the reviewer that this is an important clarification. We have re-formatted the statistics throughout the manuscript to make it clear which degrees of freedom were involved in measuring the

effect size of each statistical test. We report these degrees of freedom first, followed by a semicolon, followed by the remaining experimental details

7. Is the diagram in 4a wrong with reward location? I could not tell because the description of reward locations and distances in text does not seem to match what is in methods.

We apologize for this error. The description in the methods is correct and we have amended the main text to reflect 35 cm (instead of 30 cm). Note also that only the forward shift condition is shown in 4a.

Referee #2 (Remarks to the Author):

This is an excellent paper and now suitable for publication. One small issue on the new ED Fig 6. Part B does not appear to be a "view of a stationary period". And I am confused by the values on the different plots of theta. The theta values range from 0 to ~1 in part A but from -3 to 3 in part C. Please explain why these ranges are different and what a value of plus/minus 3 would mean for theta.

We thank the reviewer for the encouraging comments and for catching these important typos. Panel **b** of Extended Data Fig. 6 is indeed meant to be a period in which the animal is engaged, not stationary. We have fixed this in the manuscript. Additionally, the theta values in panel **a** are correct, but those in panel **c** had not yet been transformed to 2D toroidal coordinates (Methods) which should be plotted on a rhomboidal sheet. We have fixed this and updated the figure panels (reproduced below). We have also updated panel **c** (right) so that it represents a period of equal duration to the period shown in panel **c** (left).

Extended Data Fig. 6: Bump trajectory and animal behavior on an example buildup track session (~1 hr). **a. top, middle,** Inferred 2D bump coordinates on the toroidal attractor, with both phases plotted as a function of time (blue and orange panels) over the course of the entire session. **bottom,** Animal's position on the periodic track throughout the entire duration of the experiment. We observe a few periods of many seconds-minutes where the animal is stationary on the treadmill. During these periods, the inferred position of the activity bump on the attractor is also roughly stationary, though drifts slightly (within one revolution). Note that this drift appears magnified, since one lap of the animal on the track typically corresponds to 3-4 revolutions of the activity bump around the attractor. **b.** Zoomed in view of a stationary period of ~90 seconds where the animal is engaged. We compare the (z-scored) speed of the animal on the treadmill (gray line) to the (z-scored) speed of the activity bump on the attractor (red line). The activity bump's speed closely tracks the animal's speed on the treadmill. **c.** The drift in the inferred position of the activity bump on the attractor during a period when the animal is stationary (*left*), compared to a period of equal duration where the animal is running (*right*). The variability in bump position during the stationary period spans roughly $\frac{1}{5}$ of a period, which, for this module of inferred grid scale 25cm, corresponds to a spatial error of ~5cm.

Referee #3 (Remarks to the Author):

The authors have done an excellent job of responding to previous criticisms and, most importantly, the paper is now much clearer. This is an important piece of work, skillfully done and intelligently analyzed.

We thank the reviewer for their helpful feedback in the prior round of reviews, which we believe substantially improved the manuscript.

Reviewer Reports on the Second Revision:

Referees' comments:

Referee #1 (Remarks to the Author):

I thank the authors for thoroughly addressing my concerns and I feel this work is novel in two main ways:

The paper has a novel way to classify grid cells when animals run in the dark. This can be applied to many other questions and bridges the literature of VR 1d and open field 2d.

This is the first paper to address in a computational model the effect of landmarks inputs on the mailman grid cells system. The main finding is that weak pinning of landmarks on the grid cells system can predict many known phenomena of grid cell firing.

My only suggestions:

Include some of Reviewer Figure 4 in a supplemental figure as I found it illuminating, maybe with Extended Data Figure 4a.

Include all of the animal licking plots from Reviewer Fig 5a in Extended Data Figure 9.

Author Rebuttals to Second Revision:

Reviewer 1:

Include some of Reviewer Figure 4 in a supplemental figure as I found it illuminating, maybe with Extended Data Figure 4a.

We have added this Reviewer Figure to Extended Data Figure 4.

Include all of the animal licking plots from Reviewer Fig 5a in Extended Data Figure 9.

We have added all animal licking plots from Reviewer Figure 5a to Extended Data Figure 9.